# A UNIFYING VIEW OF COVERAGE IN LINEAR OFF-POLICY EVALUATION

**Philip Amortila**
University of California, Berkeley
p.amortila@berkeley.edu

**Audrey Huang**
University of Illinois Urbana-Champaign
audreyh5@illinois.edu

**Akshay Krishnamurthy**
Microsoft Research, NYC
akshaykr@microsoft.com

**Nan Jiang**
University of Illinois Urbana-Champaign
nanjiang@illinois.edu

## ABSTRACT

Off-policy evaluation (OPE) is a fundamental task in reinforcement learning (RL). In the classic setting of *linear OPE*, finite-sample guarantees often take the form

$$\text{Evaluation error} \leq \text{poly}(C^\pi, d, 1/n, \log(1/\delta)),$$

where $d$ is the dimension of the features and $C^\pi$ is a ***coverage parameter*** that characterizes the degree to which the visited features lie in the span of the data distribution. While such guarantees are well-understood for several popular algorithms under stronger assumptions (e.g. Bellman completeness), the understanding is lacking and fragmented in the minimal setting where the target value function is linearly realizable in the features. Despite recent interest in tight characterizations of the statistical rate in this setting, the right notion of coverage remains unclear, and candidate definitions from prior analyses have undesirable properties and are starkly disconnected from more standard definitions in the literature.

We provide a novel finite-sample analysis of a canonical algorithm for this setting, LSTDQ. Inspired by an instrumental-variable view, we develop error bounds that depend on a novel coverage parameter, the **feature-dynamics coverage**, which can be interpreted as linear coverage in an induced dynamical system for feature evolution. With further assumptions—such as Bellman-completeness—our definition successfully recovers the coverage parameters specialized to those settings, finally yielding a unified understanding for coverage in linear OPE.

## 1 INTRODUCTION

*Coverage* is a foundational concept in reinforcement learning (RL) theory. In off-policy evaluation (OPE), the task of evaluating a target policy based on data collected from a different behavior policy, coverage characterizes the degree to which the data distribution contains relevant information about the target policy. The relevance of coverage extends beyond OPE, and the concept plays important roles in offline policy learning (Jin et al., 2021; Xie et al., 2021), *online* RL (Xie et al., 2023; Amortila et al., 2024b;c), or even statistical-computational trade-offs in LLMs (Foster et al., 2025). Coverage provides a mathematical characterization of distribution shift which is a central challenge in RL.

Mathematically, coverage is manifested as *coverage parameters* in finite-sample guarantees: for example, standard OPE guarantees often take the form

$$\text{Evaluation error} \leq \text{poly}(C^\pi, d, 1/n, \log(1/\delta)), \tag{1}$$

where $n$ is sample size, $\delta$ is the failure probability, $d$ is the statistical dimension of the function class, $\delta$ is the failure probability, and $C^\pi$ is the coverage parameter. The definition of $C^\pi$ can take many different forms depending on the algorithm and the assumptions, as well as how the proof handles *error propagation* through the dynamics of the MDP (Farahmand et al., 2010). The most naïve definition is $\|\mu^\pi/\mu^D\|_\infty$, the boundedness of density ratio between the target policy's discounted occupancy $\mu^\pi$ and the data distribution $\mu^D$. More refined definitions often take advantage of the

structure of the underlying MDP or the function-approximation scheme. Comparisons between these definitions offer connections and unified understanding across different learning settings, such as policy evaluation vs. optimization (Duan & Wang, 2020; Jin et al., 2021), offline vs. online (Xie et al., 2023), tabular vs. function approximation (Yin & Wang, 2021), Markovian vs. partially observed (Zhang & Jiang, 2024), and single-agent vs. multi-agent RL (Cui & Du, 2022; Zhang et al., 2023).

While a unified understanding of RL through the lens of coverage is emerging, one of the most fundamental settings—where the target value function is linearly realizable in a given feature map—eludes such understanding and remains starkly disconnected from the rest of the literature. The most natural algorithm for this setting is arguably LSTD(Q) (Boyan, 1999; Lagoudakis & Parr, 2003). Despite recent statistical results for this method (Duan et al., 2021; Mou et al., 2022a; Perdomo et al., 2023), the error bounds often come with obscure conditions and are hard to interpret, with little or no understanding of which quantities play the role of coverage and how they connect to coverage parameters in related settings and algorithms.

On the other hand, simpler analyses do provide more interpretable candidates, such as $1/\sigma_{\min}(A)$ with $A = \mathbb{E}_{\mu^D}[\phi(s,a)(\phi(s,a)^\top - \gamma\phi(s',\pi)^\top)]$ being the key matrix estimated in LSTDQ (Section 2.1). Given that LSTDQ approximates $Q^\pi \approx \phi^\top\theta$ by solving a linear equation in the form of $A\theta = b$, $1/\sigma_{\min}(A)$ is a very natural candidate as it determines the invertibility of $A$ and the solution's numerical stability. While bounds in the form of Eq.(1) can be established with $C^\pi = 1/\sigma_{\min}(A)$, the quantity $1/\sigma_{\min}(A)$ is unsatisfactory in many aspects as a coverage parameter:

1. **Lacking scale invariance.** The value of $\sigma_{\min}(A)$ can change arbitrarily if we simply redefine the features as $\phi_{\text{new}} = c\phi$.[1] While seemingly unrelated, this issue is mathematically tied to the fact that $\sigma_{\min}(A)$ as a coverage parameter has no concern over the initial state distribution of the MDP which should play an important role in the definition of coverage.

2. **Lacking off-policy characterization.** Coverage parameters provide important understanding for when data contains relevant information about the target policy. For $1/\sigma_{\min}(A)$, however, the only known characterization to our knowledge is its boundedness in the restrictive on-policy case, and its magnitude is hard to interpret for general off-policy distributions.

3. **Lacking unification with other analyses.** State abstractions are a special case of linear function approximation, under which LSTDQ coincides with the model-based solution. Prior works have established *aggregated concentrability* (Jia et al., 2024) as the appropriate coverage parameter for this setting, which cannot be recovered by specializing $1/\sigma_{\min}(A)$. Moreover, both concepts differ significantly from standard definitions of coverage in linear OPE when analyzed under the Bellman-completeness assumption: standard definitions measure coverage by analyzing how errors propagate under the ***groundtruth dynamics***, whereas aggregated concentrability does so under the ***compressed dynamics*** determined by the abstraction scheme.

In this paper, we provide a novel finite-sample analysis of LSTDQ inspired by an instrument-variable (IV) view (Bradtke & Barto, 1996; Chen et al., 2022), which comes with a new coverage parameter that we call ***feature-dynamics coverage***, $C_\phi^\pi$. Feature-dynamics coverage replaces $1/\sigma_{\min}(A)$ and elegantly addresses the above problems. Furthermore, it corresponds to feature coverage in a linear dynamical system induced by the features (first studied by Parr et al. (2008)). The system is the transition dynamics of the true MDP compressed through the given features, and naturally subsumes the $\chi^2$ version of aggregated concentrability as a special case. Furthermore, given Bellman-completeness as an additional assumption, feature-dynamics coverage recovers the standard notion of linear coverage, successfully unifying the previously fragmented understanding.

## 2 PRELIMINARIES

**Markov Decision Process (MDP).** We consider the groundtruth environment modeled as an infinite-horizon discounted MDP $(\mathcal{S}, \mathcal{A}, P, R, \gamma, \mu_0)$, where $\mathcal{S}$ is the state space, $\mathcal{A}$ is the action space, $P : \mathcal{S} \times \mathcal{A} \to \Delta(\mathcal{S})$ is the transition dynamics ($\Delta(\cdot)$ is the probability simplex), $R : \mathcal{S} \times \mathcal{A} \to \Delta([0, R_{\max}])$ is the reward function, $\gamma \in [0, 1)$ is the discount factor, and $\mu_0 \in \Delta(\mathcal{S})$ is the initial state distribution. We assume $\mathcal{S}, \mathcal{A}$ are finite, but their cardinalities can be prohibitively large

---

[1] Perdomo et al. (2023) provide a bound that depends on a term related to our coverage parameter and is scale invariant. We will compare and connect to their results in Section 4.

and thus should not appear in sample-complexity guarantees. A policy $\pi : \mathcal{S} \to \Delta(\mathcal{A})$ induces a distribution over random trajectories, generated as $s_0 \sim \mu_0$, $a_t \sim \pi(\cdot|s_t)$ (or simply $a_t \sim \pi$), $r_t = R(s_t, a_t)$, $s_{t+1} \sim P(\cdot|s_t, a_t)$. Let $\mathbb{P}_\pi[\cdot]$ and $\mathbb{E}_\pi[\cdot]$ denote the probability and expectation under such a distribution. The expected return of a policy is $J(\pi) := \mathbb{E}_\pi[\sum_{t=0}^\infty \gamma^t r_t]$, which falls in the range of $[0, V_{\max}]$ with $V_{\max} := R_{\max}/(1-\gamma)$. The discounted occupancy of $\pi$ is defined as

$$\mu^\pi(s, a) = (1-\gamma)\sum_{t \geq 0}\gamma^t \mu_t^\pi(s, a) := (1-\gamma)\sum_{t \geq 0}\gamma^t \mathbb{P}_\pi[s_t = s, a_t = a]. \tag{2}$$

**Value Function and Bellman Operator.** The $Q$-function $Q^\pi \in \mathbb{R}^{\mathcal{S} \times \mathcal{A}}$ is the fixed point of Bellman operator $\mathcal{T}^\pi : \mathbb{R}^{\mathcal{S} \times \mathcal{A}} \to \mathbb{R}^{\mathcal{S} \times \mathcal{A}}$, i.e., $Q^\pi = \mathcal{T}^\pi Q^\pi$, where $\forall f \in \mathbb{R}^{\mathcal{S} \times \mathcal{A}}$, $(\mathcal{T}^\pi f)(s, a) := R(s, a) + \gamma \mathbb{E}_{s \sim P(\cdot|s,a)}[f(s', \pi)]$. Here $f(s', \pi)$ is a shorthand for $\mathbb{E}_{a' \sim \pi(\cdot|s')}[f(s', a')]$. Given any $f \in \mathbb{R}^{\mathcal{S} \times \mathcal{A}}$ as an approximation of $Q^\pi$, we can induce an estimate of $J(\pi)$ as:

$$J_f(\pi) := \mathbb{E}_{s_0 \sim \mu_0, a_0 \sim \pi}[f(s_0, a_0)], \tag{3}$$

since $J(\pi) = J_{Q^\pi}(\pi)$. Here we assume $\mu_0$ is known for simplicity, and our results extend straightforwardly to the case where $\mu_0$ is unknown and needs to be estimated from a separate dataset.

**Linear Off-policy Evaluation (OPE).** OPE is the task of estimating the performance of a given *target policy* $\pi$ based on an offline dataset $\mathcal{D}$ sampled from a *behavior policy* $\pi_b$. As a standard simplification, We assume that $\mathcal{D}$ consists of $n$ i.i.d. tuples $(s, a, r, s', a')$ generated as

$$(s, a) \sim \mu^D, r \sim R(s, a), s' \sim P^\star(\cdot|s, a), a' \sim \pi(\cdot \mid s').$$

We use $\mathbb{E}_{\mu^D}[\cdot]$ to denote the expectation of functions of $(s, a, r, s', a')$ under the data distribution, and $\mathbb{E}_{\mathcal{D}}[\cdot]$ denotes the empirical approximation from $\mathcal{D}$. For most of the paper we are concerned with *return* estimation via linear function approximation, i.e., estimating the scalar $J(\pi)$ as $J_{\widehat{Q}^\pi}(\pi)$ where $\widehat{Q}^\pi(s, a) = \phi(s, a)^\top \widehat{\theta}$ for some given feature map $\phi : \mathcal{S} \times \mathcal{A} \to \mathbb{R}^d$. We make the following standard assumptions throughout the paper:

**Assumption 1** (Feature boundedness and realizability). *We assume that there exists $\theta^\star \in \mathbb{R}^d$ such that $Q^\pi(s, a) = \phi(s, a)^\top \theta^\star$. Furthermore, assume that $\|\phi(s, a)\|_2 \leq B_\phi, \forall s, a$.*

**Mathematical Notation.** We use $\sigma_{\min}(\cdot)$ and $\lambda_{\min}(\cdot)$ to denote the smallest singular value of an asymmetric matrix and the smallest eigenvalue of a symmetric matrix, respectively. Let $\rho(\cdot)$ denote the spectral radius of a matrix. For functions over $\mathcal{S} \times \mathcal{A}$ such as $Q^\pi$ and $d^\pi$, we also view them interchangeably as vectors in $\mathbb{R}^{|\mathcal{S} \times \mathcal{A}|}$ whenever convenient. We use $a \lesssim b$ as a shorthand for $a = c \cdot b$ for some absolute constant $c$, and use big-Oh notation (e.g., $o(1/n)$ and $\mathcal{O}(1/n)$) to highlight dependence on $n$ while ignoring other problem-dependent quantities. Given two square and possibly asymmetric matrices $\Sigma$ and $\Sigma'$, $\Sigma \preceq \Sigma'$ means $v^\top(\Sigma - \Sigma')v \leq 0$ for all $v$. We let $\|v\|_\Sigma = \sqrt{v^\top \Sigma v}$ denote the Mahalanobis norm, and $[K] := \{1, \ldots, K\}$.

## 2.1 LSTDQ

The LSTDQ algorithm estimates the following moments from data:

$$\Sigma = \mathbb{E}_{\mu^D}\left[\phi(s, a)\phi(s, a)^\top\right], \qquad \Sigma_{\mathrm{cr}} = \mathbb{E}_{\mu^D}\left[\phi(s, a)\phi(s', a')^\top\right],$$
$$A = \Sigma - \gamma\Sigma_{\mathrm{cr}}, \qquad b = \mathbb{E}_{\mu^D}[\phi(s, a)r].$$

Throughout the paper, we assume:

**Assumption 2** (Invertibility). *$\Sigma$ and $A$ are invertible.*

These moments satisfy

$$A\theta^\star - b = \mathbb{E}_{\mu^D}[\phi(s, a)(\phi(s, a)^\top\theta^\star - r - \phi(s', a')^\top\theta^\star)]$$
$$= \mathbb{E}_{\mu^D}[\phi(s, a)(Q^\pi(s, a) - (\mathcal{T}^\pi Q^\pi)(s, a)] = \mathbf{0},$$

which implies $\theta^\star = A^{-1}b$ if $A$ is invertible, and LSTDQ is simply the plug-in estimate of this inverse: let $\widehat{\Sigma}, \widehat{\Sigma}_{\mathrm{cr}}, \widehat{A}, \widehat{b}$ be the empirical estimates from $\mathcal{D}$,[2] and

$$\widehat{\theta}_{\mathrm{lstd}} = \widehat{A}^{-1}\widehat{b}, \quad \widehat{Q}_{\mathrm{lstd}}(s, a) = \phi(s, a)^\top\widehat{\theta}_{\mathrm{lstd}}. \tag{4}$$

---

[2] When estimating $\widehat{\Sigma}_{\mathrm{cr}}$, either taking the average of $\phi(s, a)\phi(s', a')^\top$ or $\phi(s, a)\phi(s', \pi)^\top$ is fine and our results apply to both versions.

We do not explicitly assume the empirical matrices $\widehat{\Sigma}$ and $\widehat{A}$ are invertible, with the understanding that algorithms based on these quantities have degenerate behaviors when they are singular, and our guarantees hold under such convention.[3]

## 3 RELATED WORKS

**Coverage Parameters in Offline RL.** Early research in offline RL identifies the boundedness of density ratios, such as $\|\mu^\pi/\mu^D\|_\infty$, as a coverage parameter (Munos, 2007; Munos & Szepesvári, 2008; Antos et al., 2008). Later works point out that they can be tightened by leveraging the structure of the function class $\mathcal{F}$ used to approximate the value function, such as $\sup_{f \in \mathcal{F}} \frac{(\mathbb{E}_{\mu^\pi}[f - \mathcal{T}^\pi f])^2}{\mathbb{E}_{\mu^D}[(f - \mathcal{T}^\pi f)^2]}$. In our setting, the linear feature $\phi$ induces a linear $\mathcal{F}_\phi = \{\phi^\top \theta : \theta \in \mathbb{R}^d\}$, and these parameters often have simplified forms. Among them, the tightest known coverage parameter for off-policy return estimation is (Zanette et al., 2021; Yin et al., 2022; Gabbianelli et al., 2024; Jiang & Xie, 2024):

$$C_{\text{lin}}^\pi = (\phi^\pi)^\top \Sigma^{-1} \phi^\pi, \quad \text{where} \ \ \phi^\pi := \mathbb{E}_{(s,a) \sim \mu^\pi}[\phi(s,a)]. \tag{5}$$

In words, $C_{\text{lin}}^\pi$ only requires the *mean feature vector under the discounted occupancy of $\pi$* ($\mu^\pi$, cf. Eq. (2)) to lie in the span of data features. Looser alternatives have also been proposed, such as $\mathbb{E}_{\mu^\pi}[\|\phi\|_{\Sigma^{-1}}]$ require coverage of the distribution $\phi(s,a), (s,a) \sim \mu^\pi$ in a point-wise manner; see Jiang & Xie (2024) for further discussions. The majority of the study on coverage, however, crucially relies on the following assumption on $\mathcal{F}$ which is substantially stronger than realizability ($Q^\pi \in \mathcal{F}$), and we ***do not assume*** it in our main results unless stated otherwise explicitly.

**Assumption 3** (Bellman-completeness)**.** *Let $\mathcal{F} \subset (\mathcal{S} \times \mathcal{A} \to \mathbb{R})$ be a function class for approximating $Q^\pi$. We say that it is Bellman-complete if*

$$\mathcal{T}^\pi f \in \mathcal{F}, \forall f \in \mathcal{F}.$$

Indeed, a major potential of LSTDQ is that it is one of the few algorithms who enjoy theoretical guarantees under only realizability $Q^\pi \in \mathcal{F}$, which further enables its important role in the challenging problem of offline model selection (Xie & Jiang, 2020a; Liu et al., 2025). Unfortunately, coverage in the absence of Bellman-completeness is poorly understood even in the linear setting, as discussed in Section 1, which we address in this paper.

**LSTD(Q) Analyses.** LSTD methods are initially derived as the fixed point solution of TD methods (Sutton & Barto, 2018), and its closed-form nature separates it from typical dynamical-programming-style RL algorithms that often suffer from divergence. While we focus on LSTDQ, our results naturally extend to other variants such as LSTD (which uses state-feature to approximate $V^\pi$) or off-policy LSTD (up to handling importance sampling via concentration inequalities) (Nedić & Bertsekas, 2003; Bertsekas & Yu, 2009; Dann et al., 2014).

Early finite-sample analyses of LSTD are mostly in the on-policy setting and depend on quantities like $\sigma_{\min}(A)$ (Bertsekas, 2007; Lazaric et al., 2010; 2012). There is a recent surge of interest in providing tight statistical characterizations of LSTD, including in the off-policy setting (Amortila et al., 2020; Mou et al., 2022b; Amortila et al., 2023; Perdomo et al., 2023). These results, however, offer very little discussion on coverage, and sometimes require additional regularity assumptions. Perdomo et al. (2023) recently provides a sharp analysis that largely subsumes the earlier results, which we will compare to in Section 5.1.

## 4 FINITE-SAMPLE ANALYSIS OF LSTDQ

In this section we will present the finite-sample analysis of LSTDQ, which depends on our proposed coverage parameter.

---

[3]That is, either the high-probability event guarantees that the empirical matrix is invertible, or such matrices appears in the bound and make the guarantee vacuous (as in e.g., Eq. (12)).

**Special case of $\gamma = 0$.** It is instructive to start with the special case of $\gamma = 0$, where tight guarantees and the definition of coverage are well understood. Recall that $A\theta^\star = b$ can be rewritten as:

$$\mathbb{E}[ZX^\top]\theta^\star = \mathbb{E}[ZY], \tag{6}$$

where $Z = \phi(s,a) \in \mathbb{R}^d$, $X = \phi(s,a) - \gamma\phi(s',a') \in \mathbb{R}^d$, $Y = r \in \mathbb{R}$, and the expectation $\mathbb{E}[\cdot]$ is under $\mu^D$. Below we will go back and forth between the linear regression (LR) notation system $(X, Y, \mathbb{E}[XX^\top], \ldots)$ and the RL notation system $(\phi, r, \Sigma, \ldots)$, where we obtain concentration bounds from the LR literature and meaningful guarantees for OPE in the RL setting, respectively.

When $\gamma = 0$, we essentially face a contextual bandit problem with linear reward, and Eq. (6) becomes

$$\mathbb{E}[XX^\top]\theta^\star = \mathbb{E}[XY], \tag{7}$$

which is a classic linear regression problem, with $A = \Sigma = \mathbb{E}[XX^\top]$. In this special case, LSTDQ simply performs LR to fit the parameter $\theta^\star$. While parameter identification guarantees in LR (i.e., error bounds on $\|\widehat{\theta} - \theta^\star\|$) inevitably have to depend on $\sigma_{\min}(A) = \lambda_{\min}(\mathbb{E}[XX^\top])$, the key is in how we use $\widehat{\theta}_{\mathrm{lstd}}$ to form the final estimation in OPE:

$$J_{\widehat{Q}_{\mathrm{lstd}}}(\pi) = \mathbb{E}_{s_0 \sim \mu_0, a_0 \sim \pi}[\phi(s_0, a_0)^\top \widehat{\theta}_{\mathrm{lstd}}] = \phi_0^\top \widehat{\theta}_{\mathrm{lstd}},$$

where

$$\phi_0 := \mathbb{E}_{s_0 \sim \mu_0, a_0 \sim \pi}[\phi(s_0, a_0)]. \tag{8}$$

That is, for the purpose of estimating $J(\pi)$, we only need $\widehat{\theta}_{\mathrm{lstd}}$ to be accurate in the direction of $\phi_0$, and a high-probability bound can be established if $\phi_0$ is well *covered* by the distribution of $X = \phi(s,a)$ observed in data ($(s,a) \sim \mu^D$): informally, with probability at least $1 - \delta$,

$$|J_{\widehat{Q}_{\mathrm{lstd}}}(\pi) - J(\pi)| = |\phi_0^\top(\widehat{\theta}_{\mathrm{lstd}} - \theta^\star)| \lesssim \|\phi_0\|_{\widehat{\Sigma}^{-1}} \sqrt{\frac{\log(1/\delta)}{n}} V_{\max}. \tag{9}$$

Here $\|\phi_0\|_{\widehat{\Sigma}^{-1}}$ plays the role of coverage, characterizing how well the expected feature under the target policy $\pi$ is covered by the random features observed in the data (which determines $\widehat{\Sigma}$). Similar bounds with the population version of coverage $\|\phi_0\|_{\Sigma^{-1}}$ also hold under additional regularity assumptions (Hsu et al., 2011). The quantity is also consistent with the standard notion of linear coverage in MDPs (Eq. (5)) under Bellman completeness (Assumption 3), which is equivalent to realizability in the bandit setting ($\gamma = 0$).

For readers familiar with using LR bounds to calculate uncertainty bonuses in RL, this result might come as a surprise as it is ***dimension-free***, whereas the usual bound has $\sqrt{(d + \log(1/\delta))/n}$ which depends on $d$ (Abbasi-Yadkori et al., 2011; Jin et al., 2020; 2021). Roughly speaking, this is because the latter bound holds for all possible $\phi_0 \in \mathbb{R}^d$ simultaneously, but in OPE we only care about a single given $\phi_0$, and leveraging this saves the $d$ factor; we call this kind of results *directional bounds* and provide more detailed discussions in Appendix A.

**Extending to $\gamma > 0$ with Instrumental-Variable inspiration.** Given that the $\gamma = 0$ case is well-understood and does not suffer the issues mentioned in the introduction, we therefore seek to extend the above framework to $\gamma > 0$. When $\gamma > 0$, however, we have $Z \neq X$, and Eq. (6) is a form of Instrumental Variable (IV) problem induced by "error-in-the-variable" issues: it is known that

$$R(s,a) = \phi_{\mathrm{td}}(s,a)^\top \theta^\star, \quad \text{with } \phi_{\mathrm{td}}(s,a) := \phi(s,a) - \gamma\mathbb{E}_{\mu^D}[\phi(s',a')|s,a],$$

that is, the expected temporal-difference feature, $\phi_{\mathrm{td}}$, can linearly predict reward, which LSTDQ leverages to recover $\theta^\star$. However, in the data we do not observe the expected TD feature but its random realization, $X = \phi(s,a) - \gamma\phi(s',a')$, and $X - \phi_{\mathrm{td}}(s,a)$ is zero-mean (conditioned on $(s,a)$) noise. Given such "error in the variable", $\mathbb{E}[XX^\top]\theta^\star \neq \mathbb{E}[XY]$, so a straightforward linear regression from $X$ to $Y$ does not work. LSTDQ solves this problem by introducing $Z = \phi(s,a)$ as an *instrumental variable* (Bradtke & Barto, 1996; Chen et al., 2022), which is independent of the noise $X - \phi_{\mathrm{td}}(s,a)$ given $(s,a)$ and thus helps marginalizes out the said noise. Based on this view, we extend the LR analysis of $\gamma = 0$ to the $\gamma > 0$ case by consulting the IV literature (e.g., Xia et al., 2024; Della Vecchia & Basu, 2025), which leads to our main finite-sample error bounds (see Appendix B for the proof).

**Theorem 1** (Population Coverage Bound). *There exists $n_0$ such that when $n \geq n_0$, w.p. $\geq 1 - \delta$,*

$$\left| J_{\widehat{Q}_{\mathrm{lstd}}}(\pi) - J(\pi) \right| \lesssim \frac{V_{\max}}{1-\gamma} \sqrt{\frac{C_\phi^\pi \cdot \log(1/\delta)}{n}} + o(\sqrt{1/n}), \tag{10}$$

*where*

$$C_\phi^\pi := (1-\gamma)^2 \phi_0^\top A^{-1} \Sigma A^{-\top} \phi_0 \tag{11}$$

*and $n_0$ and the $o(1/\sqrt{n})$ term may depend on $d$ and $1/\sigma_{\min}(A)$.*

**Theorem 2** (Empirical Coverage Bound). *Under Assumptions 1 and 2, with probability at least $1 - \delta$,*

$$\left| J_{\widehat{Q}_{\mathrm{lstd}}}(\pi) - J(\pi) \right| \lesssim \frac{V_{\max}}{1-\gamma} \cdot \sqrt{\frac{\widehat{C}_\phi^\pi \cdot (d + \log(1/\delta))}{n}} \tag{12}$$

*where*

$$\widehat{C}_\phi^\pi := (1-\gamma)^2 \phi_0^\top \widehat{A}^{-1} \widehat{\Sigma} \widehat{A}^{-\top} \phi_0. \tag{13}$$

*We treat $\widehat{C}_\phi^\pi = +\infty$ if $\widehat{A}$ is not invertible.*

**Coverage Parameter and Tightness.** Our main results consist of two guarantees where $\widehat{C}_\phi^\pi$ and $C_\phi^\pi$ play the role of the coverage parameters. The $(1-\gamma)^2$ are normalization constants, whose roles will become clear in Section 5. Eq. (10) provides a bound that largely recovers that of linear regression in Eq. (9): when $\gamma = 0$, we have $A = \Sigma$, and therefore $C_\phi^\pi = \phi_0^\top \Sigma^{-1} \phi_0 = \|\phi_0\|_{\Sigma^{-1}}^2$, which is the (often more desirable) population version of the $\|\phi_0\|_{\widehat{\Sigma}^{-1}}$ term in Eq. (9). Since Eq. (9) is well-established for linear regression (and already tighter than commonly used LR bound in terms of $d$ dependence), this demonstrates the tightness of our bound in the $\gamma = 0$ regime up to the lower-order term. When $\gamma > 0$, our bound is also tight when compared to existing OPE guarantees for general function approximation analyzed under Bellman completeness as we will see in Section 5.3.

A caveat of Eq. (10) is that it still depends on quantities like $\sigma_{\min}(A)$ albeit in the lower-order term ($o(1/\sqrt{n})$) and the burn-in condition ($n \geq n_0$), raising the suspicion that $\sigma_{\min}(A)$ is still relevant to coverage. To address this, we provide Eq. (12) that depends on the empirical coverage parameter $\widehat{C}_\phi^\pi$ similar to Eq. (9), with no lower-order terms whatsoever, showing that $\sigma_{\min}(A)$ can be completely removed from the bound. Eq. (12) is looser than Eq. (10) in $d$ dependence, as it provides a guarantee that holds for all possible initial distribution $\mu_0$ simultaneously which can be used to establish function-estimation guarantees (Appendix D); see the discussion of directional bounds below Eq. (9) and in Appendix A. In Appendix E, we provide yet another result that eliminates the dependence on $1/\sigma_{\min}(A)$ in the population $C_\phi^\pi$ bound (Theorem 1), in exchange for dependence on standard quantities relevant to the analysis of LR under random design (namely $1/\lambda_{\min}(\Sigma)$), which we expect can be further sharpened to leverage-score-type conditions (Hsu et al., 2011; Perdomo et al., 2023)).

## 5 UNDERSTANDING THE COVERAGE PARAMETER

In this section we provide interpretations of $C_\phi^\pi$ as a coverage parameter and discuss how it addresses the issues mentioned in Section 1. First, it is clear that $C_\phi^\pi$ is invariant to feature rescaling, thanks to the introduction of $\phi_0$. That said, the expression $C_\phi^\pi = (1-\gamma)^2 \phi_0^\top A^{-1} \Sigma A^{-\top} \phi_0$ does not lend itself to easy intuition, let alone how it connects to and unifies existing results.

**Warm-up: the tabular case.** We start with the tabular setting and show that $C_\phi^\pi$ becomes something familiar, offering some basic intuitions as well as assurance that $C_\phi^\pi$, a quantity that falls out of the IV concentration analyses, holds meaningful interpretations in RL. The key is to rewrite $C_\phi^\pi/(1-\gamma)^2$ as

$$\phi_0^\top A^{-1} \Sigma A^{-\top} \phi_0 = \phi_0^\top (I - \gamma B^\pi)^{-\top} \Sigma^{-1} (I - \gamma B^\pi)^{-1} \phi_0, \quad \text{where } B^\pi := (\Sigma^{-1} \Sigma_{\mathrm{cr}})^\top. \tag{14}$$

The tabular setting can be viewed as a special case of linear function approximation with $d = |\mathcal{S} \times \mathcal{A}|$, and $\phi(s, a) = \mathbf{e}_{s,a}$ is the unit vector with the $(s, a)$-th coordinate being 1 and all other coordinates being 0. In this case, $\phi_0$ is simply the vector representation of the initial state-action distribution $\mu_0^\pi$, where $(s_0, a_0) \sim \mu_0^\pi \Leftrightarrow s_0 \sim \mu_0, a_0 \sim \pi$; $B^\pi$ is an $|\mathcal{S} \times \mathcal{A}| \times |\mathcal{S} \times \mathcal{A}|$ matrix with $[B^\pi]_{(s',a'),(s,a)} = P^\pi(s', a'|s, a) = P(s'|s, a)\pi(a'|s')$, i.e., the transition kernel of the Markov chain over $\mathcal{S} \times \mathcal{A}$ induced by policy $\pi$.[4] Put together, we have the textbook identity

$$(1 - \gamma)(I - \gamma B^\pi)^{-1}\phi_0 = \mu^\pi,$$

where we recall the definition of the discounted occupancy $\mu^\pi$ from Eq. (2). Plugging it into $C_\phi^\pi$,

$$C_\phi^\pi = (\mu^\pi)^\top \Sigma^{-1} \mu^\pi = \sum_{s,a} \mu^\pi(s, a)^2/\mu^D(s, a) = \mathbb{E}_{\mu^D}[(\mu^\pi/\mu^D)^2],$$

which is the $\chi^2$-divergence between $\mu^\pi$ and $\mu^D$ up to a constant shift and has appeared as a tight coverage parameter (especially when compared to $\|\mu^\pi/\mu^D\|_\infty$; Xie & Jiang, 2020b) when coverage is measured based on density ratios.

## 5.1 GENERAL INTERPRETATION

We now offer the interpretation for the general setting. Note that $B^\pi$ can be viewed as the multi-variate linear-regression solution of the regression problem $\phi(s, a) \mapsto \phi(s', \pi)$, thus

$$\mathbb{E}_{s' \sim P(\cdot|s,a)}[\phi(s', \pi)] \approx B^\pi \phi(s, a). \tag{15}$$

In general, the above relationship is only approximate (in Section 5.3 we will see that it becomes *exact* under an additional assumption), although $B^\pi$ is the best linear predictor. This leads to the following interpretation of $C_\phi^\pi$ (see Appendix C.1 for the proof):

**Proposition 1.** *Define a deterministic linear dynamical system $\{\mu_{\phi,t}^\pi\}_{t \geq 0}$, with $\mu_{\phi,0}^\pi := \phi_0$, and $\forall t \geq 0$,*

$$\mu_{\phi,t+1}^\pi = B^\pi \mu_{\phi,t}^\pi.$$

*When $\rho(B^\pi) < 1/\gamma$,[5] define the feature occupancy in $B^\pi$ as $\mu_\phi^\pi := (1 - \gamma)\sum_{t \geq 0} \gamma^t \mu_{\phi,t}^\pi$, then*

$$C_\phi^\pi = (\mu_\phi^\pi)^\top \Sigma^{-1} \mu_\phi^\pi.$$

The proposition rewrites $C_\phi^\pi$ in a form that closely resembles the standard notion of linear coverage in the literature (Eq. (5)), where we see the expected feature occupancy under the target policy ($\mu_\phi^\pi$ here) measured under the data-covariance norm $\Sigma^{-1}$; see Section 5.3. Accordingly, we call $C_\phi^\pi$ the ***feature-dynamics coverage***. The difference is that here the feature occupancy is defined in a deterministic dynamical system $B^\pi$ instead of the true MDP. Furthermore, while the latter, $\phi^\pi := \mathbb{E}_{(s,a) \sim \mu^\pi}[\phi(s, a)]$, is always bounded, $\mu_\phi^\pi$, on the other hand, may not be bounded in general and $\{\mu_{\phi,t}^\pi\}_{t \geq 0}$ may actually diverge. The connection between LSTD and the linear dynamical system $B^\pi$ was first identified by Parr et al. (2008) (see also Duan & Wang (2020)), though they focused on the algebraic equivalence between LSTD and the model-based solution in $B^\pi$, and did not perform finite-sample analyses or connect this to the notion of coverage.

**Comparison to Perdomo et al. (2023) and subsuming known tractable conditions.** Our bound shows that linear OPE under realizability is tractable as long as $C_\phi^\pi$ is small, which sharpens and generalizes existing understanding of the tractability of linear OPE. In particular, we compare to Perdomo et al. (2023), whose analysis was shown to be sharp and subsume many prior conditions known in the literature, including on-policy sampling (Tsitsiklis & Van Roy, 1997), Bellman completeness (c.f. Section 5.3), low distribution shift (Wang et al., 2021), symmetric stability (Mou et al., 2022a), and contractivity (Kolter, 2011) (see the discussion in Perdomo et al. for formal definitions). Furthermore, we can leverage the interpretation of our coverage coefficient to provide a novel and weaker on-policy-type condition that enables tractability.

**Proposition 2.** *Assume $\phi$ contains a bias term, that is, there exists $\theta_0 \in \mathbb{R}^d$ such that $\phi(s, a)^\top \theta_0 \equiv 1 \forall s, a$. If $\rho(B^\pi) < 1/\gamma$ and $\mathbb{E}_{\mu^D}[\phi(s, a)] = \mathbb{E}_{\mu^D}[\phi(s', \pi)] = \phi_0$, then $C_\phi^\pi \leq 1$.*

---

[4]For cleanliness of notation, we are taking the convention that current state is on the column, and next state is on the row (as standard in stochastic processes). In RL it is often the other way around.

[5]Recall that $\rho$ denotes the spectral radius of a matrix.

Unlike previous on-policy-type conditions that relate the *distributions* of $(s, a)$ and $(s', a')$ (see Appendix C.2 for more discussion), Proposition 2 only requires their means to be identical. The intuition is that while $B^\pi$ may not be accurate in next-feature predictions on individual $(s, a)$, its population prediction for $\mathbb{E}_{\mu^D}[\phi(s', a')]$ can be correct when $\phi$ contains a bias term. As a consequence, when $(s, a)$ and $(s', a')$ share the same mean feature $\mathbb{E}_{\mu^D}[\phi(s, a)]$ and this coincides with the starting feature $\phi_0$, we can establish that all future $\mu^\pi_{\phi, t}$ are identical to $\mathbb{E}_{\mu^D}[\phi(s, a)]$, leading to coverage in the compressed feature dynamics.

In terms of quantitative rates, Perdomo et al. establish that, under some regularity assumptions, $\|\Sigma^{1/2}(\theta^\star - \widehat{\theta})\|_2 \lesssim \frac{1}{\sigma_{\min}(I - \gamma\Sigma^{-1/2}\Sigma_{\mathrm{cr}}\Sigma^{-1/2})} \cdot \varepsilon_{\mathrm{stat}}$, for some $\varepsilon_{\mathrm{stat}}$ which is polynomial in $d, 1/n, \log(1/\delta)$, and spectral properties of $\Sigma$. While they only show function-estimation guarantee on $\mu^D$ (c.f. Appendix D), this intermediate result immediately implies a return-estimation guarantee comparable to ours:

$$\left| J_{\widehat{Q}_{\mathrm{lstd}}}(\pi) - J(\pi) \right| \leq \frac{\|\phi_0\|_{\Sigma^{-1}}}{\sigma_{\min}(I - \gamma\Sigma^{-1/2}\Sigma_{\mathrm{cr}}\Sigma^{-1/2})} \cdot \varepsilon_{\mathrm{stat}}.$$

As we have already shown that our statistical rate is tight, it suffices to compare our $C^\pi_\phi$ to their multiplicative factor in front of $\varepsilon_{\mathrm{stat}}$. In particular, we establish the following relationship (see Appendix C.3 for the proof).

**Proposition 3.** *We have that:*

$$\sqrt{C^\pi_\phi} = (1 - \gamma)\|(I - \gamma\Sigma^{-1/2}\Sigma_{\mathrm{cr}}\Sigma^{-1/2})^{-\top}\Sigma^{-1/2}\phi_0\|_2 \leq (1 - \gamma)\frac{\|\phi_0\|_{\Sigma^{-1}}}{\sigma_{\min}(I - \gamma\Sigma^{-1/2}\Sigma_{\mathrm{cr}}\Sigma^{-1/2})}.$$

*Furthermore, the gap between the LHS and the RHS may be arbitrarily large.*

This demonstrates that our coverage parameter, in addition to subsuming known tractability conditions for linear OPE, may lead to return-estimation guarantees that are sharper than those of Perdomo et al. (2023) (in an instance-dependent sense). This is due to our sensitivity on $\phi_0$ and its relationship to the spectrum of the matrix $(I - \gamma\Sigma^{-1/2}\Sigma_{\mathrm{cr}}\Sigma^{-1/2})$.

## 5.2 RECOVERING AGGREGATED CONCENTRABILITY

State abstractions are a special case of linear function approximation, where each state $s$ is mapped to one of the $K$ abstract states, $\psi(s) \in [K] := \{1, \ldots, K\}$, effectively treating states with the same $\psi(s)$ as aggregated and equivalent to reduce the size of the state space. Under the abstraction scheme, the natural model-based solution coincides with LSTDQ with $\phi(s, a) = \mathbf{e}_{\psi(s), a}$. When we only assume realizability (Assumption 1), the abstraction is $Q^\pi$-*irrelevant* (Li et al., 2006), and has been analyzed by Xie & Jiang (2020a); Zhang & Jiang (2021); Jia et al. (2024) with the following notion of *aggregated concentrability* as its coverage parameter (see Appendix C.4 for details):

**Definition 1** (Aggregated concentrability). *Given* $\psi : \mathcal{S} \to [K]$, *define the abstract MDP* $M_\psi = (\mathcal{S}_\psi, \mathcal{A}, P_\psi, R_\psi, \gamma, \psi_0)$ *where*[6] $\mathcal{S}_\psi = [K]$, *the initial distribution is* $\psi_0(k) = \sum_{s:\psi(s)=k} \mu_0(s)$, *and*

$$P_\psi(k'|k, a) = \frac{\mu^D(s, a) \cdot \sum_{s:\psi(s)=k}\left(\sum_{s':\psi(s')=k'} P(s'|s, a)\right)}{\sum_{s:\psi(s)=k} \mu^D(s, a)}.$$

*Consider* $\pi$ *whose action distribution depends on* $s$ *only through* $\psi(s)$, *with a slight abuse of notation we write* $\pi(\cdot|s) = \pi(\cdot|k), \forall k = \psi(s)$. Aggregated concentrability *refers to the size of* $\mu^\pi_{M_\psi}/\phi^D$, *either measured in* $\|\cdot\|_\infty$ *or* $\chi^2$, *where* $\mu^\pi_{M_\psi}$ *is discounted occupancy in MDP* $M_\psi$, *and* $\phi^D(k, a) = \sum_{s:\psi(s)=k} \mu^D(s, a) = \mathbb{E}_{\mu^D}[\phi(s, a)]$.

In this definition, $P_\psi$ is the dynamics over the abstract state space, and it is easy to see that the transition kernel of abstract-state pairs under $\pi$, $P^\pi_\psi(k', a'|k, a) = P_\psi(k'|k, a)\pi(a'|k')$, coincides with the definition of $B^\pi$ (see Proposition 1) specialized to this setting, and $\Sigma = \mathrm{diag}(\phi^D)$. As a result, $C^\pi_\phi$ recovers the $\chi^2$ version of aggregated concentrability (see Appendix C.5 for the proof):

---

[6]The definition of $R_\psi$ is irrelevant here and thus omitted.

Figure 1: Illustration of the evolution of occupancies under the true dynamics $P^\pi$ (top row) and that of features under the compressed dynamics $B^\pi$ (bottom row). Under Bellman completeness, the dashed blue arrows hold and two routes ($\rightarrow \ldots \rightarrow \downarrow$ vs. $\downarrow \rightarrow \ldots \rightarrow$) yield the same expected feature vectors, but they are generally different without such an assumption.

**Proposition 4.** *When $\phi(s,a) = \mathbf{e}_{\psi(s),a}$ is induced by a state abstraction $\psi$ and $\pi$ depends on $s$ only through $\psi(s)$, recall that $\phi^D = \mathbb{E}_{\mu^D}[\phi]$ is a distribution over $(k,a)$ pairs, and we have*

$$C_\phi^\pi = \mathbb{E}_{(k,a) \sim \phi^D}[(\mu_{M_\psi}^\pi / \phi^D)^2].$$

### 5.3 Recovering Standard Linear Coverage under Bellman-Completeness

Prior results on abstractions leave an intriguing question open: they measure coverage by analyzing error propagation in $M_\psi$, which a lower-dimensional and approximate model **compressed** from $M$ by $\psi$, as evidenced by $\mu_{M_\psi}^\pi$ in the definition of aggregated concentrability; this is also consistent with our results in Section 5.1 where occupancy is measured in the compressed linear dynamical system $B^\pi$. On the other hand, the mainstream notion of coverage in linear OPE, obtained under the Bellman-completeness assumption, is $C_{\text{lin}}^\pi = (\phi^\pi)^\top \Sigma^{-1} \phi^\pi$ (Eq. (5)), which is concerned with error propagation in the **true dynamics** $M$ since $\phi^\pi$ is defined w.r.t. the occupancy $\mu^\pi$ in $M$. While anecdotally it has been the general perception from the community that error propagation in compressed models (as in $Q^\pi$-irrelevant abstractions) was an exception to the typical error propagation analyses, our results below show that results that are seemingly disconnected with each other can be elegantly unified through the following proposition:

**Proposition 5.** *Let $\mathcal{F}_\phi := \{\phi^\top \theta : \theta \in \mathbb{R}^d\}$ be the space of functions linear in $\phi$. Assume $\mathcal{F}_\phi$ satisfies Bellman-completeness (Assumption 3). Then, (1) $B^\pi$ becomes an exact model for next-feature prediction, i.e., $\mathbb{E}_{s' \sim P(\cdot|s,a)}[\phi(s',\pi)] = B^\pi \phi(s,a)$, (2) $\mu_\phi^\pi = \phi^\pi$, (3) $\rho(B^\pi) \le 1$, and (4)*

$$C_\phi^\pi = C_{\text{lin}}^\pi = (\phi^\pi)^\top \Sigma^{-1} \phi^\pi.$$

The essence of the proposition is illustrated in Figure 1, showing that the expected features produced by the groundtruth dynamics ($\phi^\pi$) and the compressed dynamics ($\mu_\phi^\pi = (1-\gamma)\sum_t \gamma^t \mu_{\phi,t}^\pi$) coincide under Bellman-completeness, thus demonstrating that **error propagation through true dynamics is a special case of and thus unified with error propagation in the compressed dynamics.**

**Comparison to Yin & Wang (2020); Duan & Wang (2020) in Tabular and Linear MDPs.** In the special case of tabular MDPs, LSTDQ coincides with model-based OPE, and Bellman-completeness holds automatically. In this setting, our Eq. (12) matches the dimension-free guarantees (in $1/\sqrt{n}$ term) given by Yin & Wang (2020).[7] Duan & Wang (2020) also provide similar dimension-free guarantees for FQE in linear MDPs. In comparison, our analyses hold in arbitrary MDPs beyond these structural models.

**Connection to Bellman Residual Minimization (BRM).** Many (if not most) algorithms for learning $Q^\pi$ with general function approximation coincide with LSTDQ under linear function approximation (Antos et al., 2008; Xie et al., 2021; Uehara et al., 2020), and this fact allows us to compare our bound to the more general analyses in the literature. Among those algorithms, BRM

---

[7]They analyze the finite-horizon setting but the $1/\sqrt{n}$ terms in the bounds match under standard translation. Also, they use the variance of value function w.r.t. transition randomness in place of our $V_{\max}$, which we could also incorporate and choose not to for readability.

is a well-investigated example, which approximates $Q^\pi$ by solving the following minimax problem (Antos et al., 2008):

$$\widehat{f}^\pi = \arg\min_{f\in\mathcal{F}} \sup_{f'\in\mathcal{F}} \left( \mathbb{E}_\mathcal{D}[(f(s,a) - r - \gamma f(s',\pi))^2] - \mathbb{E}_\mathcal{D}[(f'(s,a) - r - \gamma f(s',\pi))^2] \right), \quad (16)$$

whose finite-sample guarantee can be established under Bellman completeness (Assumption 3). Antos et al. (2008); Xie et al. (2021) show that when $\mathcal{F}$ is linear, the solution coincides with LSTDQ, so we can compare the guarantee of BRM under linear $\mathcal{F}$ with our Theorem 2. Jiang & Xie (2024) show that BRM's error bound is (see their Eq. (19))

$$\left| J_{\widehat{f}^\pi}(\pi) - J(\pi) \right| \lesssim \frac{V_{\max}}{1-\gamma} \cdot \sqrt{\frac{C^\pi \log(|\mathcal{F}|/\delta)}{n}}. \quad (17)$$

In the linear setting, their $C^\pi$ is $C^\pi_{\text{lin}}$ (see their Eq. (23)), and $\log|\mathcal{F}| \approx d$ based on a standard covering-number argument. Under such translation, the main $O(n^{-\frac{1}{2}})$ term in our Eq. (10) matches Eq. (17) in the coverage and horizon dependence, and is sharper than the latter as our bound is "directional" and does not depend on $\log|\mathcal{F}| \approx d$. In contrast, the concentration event in BRM's analysis is that Eq. (16) concentrates to $\mathbb{E}_{\mu^D}[(f - \mathcal{T}^\pi f)^2]$, which is independent of the initial state. This makes the nature of Eq. (17) a function-estimation guarantee (see Appendix A), similar to our Theorem 2 (which does depend on $d$). Whether Eq. (17) can be improved to a directional bound (i.e., no $\log|\mathcal{F}|$) for general non-linear $\mathcal{F}$ is an interesting open problem.

## 5.4 UNIFICATION WITH MARGINALIZED IMPORTANCE SAMPLING

In Section 5.3 we mentioned that many algorithms designed for general function approximation reduce to LSTDQ when linear classes are used. Another example is Minimax Weight Learning (MWL; Uehara et al., 2020), a representative method for marginalized importance sampling, whose key idea is illustrated by the following inequality: given $\mathcal{F}$ such that $Q^\pi \in \mathcal{F}$, $\forall w : \mathcal{S} \times \mathcal{A} \to \mathbb{R}$,

$$\left| \frac{1}{1-\gamma} \mathbb{E}_{\mu^D}[w(s,a)r] - J(\pi) \right| \le \sup_{f\in\mathcal{F}} \left| J_f(\pi) + \frac{1}{1-\gamma} \mathbb{E}_{\mu^D}[w(s,a) \cdot (\gamma f(s',\pi) - f(s,a))] \right|, \quad (18)$$

so learning $w$ from some $\mathcal{W}$ class that minimizes (the empirical estimate of) the RHS to $\approx 0$ ensures that $\frac{1}{1-\gamma} \mathbb{E}_{\mu^D}[w(s,a)r]$ is a good estimation of $J(\pi)$. Theoretically, if some $w^\star \in \mathcal{W}$ sets the RHS of Eq. (18) to 0, finite-sample guarantees can be established, where coverage is reflected by the magnitude of $w^\star$. As an example, $w^\star(s,a) = \mu^\pi(s,a)/\mu^D(s,a)$ always sets the RHS to 0, and we pay the size of $w^\star$ as the coverage parameter (e.g., $\|\mu^\pi/\mu^D\|_\infty$) through concentration inequalities; see Xie & Jiang (2020b, Section 6.2) for further discussions on this.

When both $\mathcal{W}$ and $\mathcal{F}$ are linear, Uehara et al. (2020) show that the MWL algorithm is equivalent to LSTDQ. We now show that their coverage parameters and guarantees, when improved with insights from follow-up works, coincide with our analyses in the linear setting. In particular, Zhang & Jiang (2024) point out that the $w^\star$ that minimizes the population objective Eq. (18) takes a different form in the linear case: $w^\star(s,a) = (1-\gamma)\phi_0^\top A^{-1}\phi(s,a)$. An immediate implication is that

$$\mathbb{E}_{\mu^D}[w^\star(s,a)^2] = C^\pi_\phi.$$

That is, the second moment of $w^\star$ on data is precisely our coverage parameter. While Uehara et al. (2020) measures the size of $w^\star$ by $\|w^\star\|_\infty$ due to the use of Hoeffding's inequality, replacing it with Bernstein's will improve $\|w^\star\|_\infty$ to $\mathbb{E}_{\mu^D}[w^\star(s,a)^2]$ in the main $O(n^{-1/2})$ term, which matches our bound in Eq.(10) except that we do not have $d$ dependence; see Appendix C.7 for further details.

## 6 CONCLUSION AND DISCUSSION

We tackled the fundamental problem of linear off-policy evaluation under the minimal assumption of realizability. We re-analyzed a canonical algorithm for this setting, LSTDQ, and developed error bounds that introduced the feature-dynamics coverage, a new notion of coverage that tightens and sharpens our understanding of this setting. This parameter admits a natural interpretation as coverage in a feature-induced dynamical system, while simultaneously generalizing special cases such as aggregated concentrability with state abstraction features and linear coverage with Bellman-complete features. Altogether, our results serve as clearer and more unified foundation for the theory of linear OPE.

## DISCLOSURE OF LLM USAGE

In the initial phase of the project, the authors had a vague conjecture and rough road-map of the main results in the paper, and used an LLM to execute the plan further to verify the feasibility of the project. We also subsequently used LLMs to help with literature review and proofs with some elementary linear-algebraic lemmas.

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

APPENDIX

## A  DIMENSION-FREE GUARANTEE FOR CONTEXTUAL BANDITS

In Section 4 we claim that the OPE bound for contextual bandits is *dimension-free* (Eq. (9)), i.e., independent of the feature dimension $d$. This may come as a surprise, as similar bounds are frequently used in RL for calculating uncertainty bonuses (Abbasi-Yadkori et al., 2011; Jin et al., 2020; 2021), but they take the form of [8]

$$\|\phi_0\|_{\widehat{\Sigma}^{-1}} \sqrt{\frac{d + \log(1/\delta)}{n}} V_{\max}. \tag{19}$$

In comparison, our Eq. (9) does not have the $d$ factor, which we explain here. To start, we first rewrite the error as the average of terms across the data points. Here we temporarily use subscript $i$ to denote the $i$-th data point $(s_i, a_i, r_i)$ (in most of the main text we use subscript to denote the time step of a state or action, such as in $s_t$):

$$\phi_0^\top(\widehat{\theta}_{\text{lstd}} - \theta^\star) = \phi_0^\top \widehat{\Sigma}^{-1}(\widehat{b} - \widehat{A}\theta^\star) = \phi_0^\top \widehat{\Sigma}^{-1}\left(\frac{1}{n}\sum_{i=1}^n \phi(s_i, a_i)\epsilon_i\right) = \frac{1}{n}\sum_{i=1}^n w_i \epsilon_i.$$

Here $\epsilon_i := r_i - \mathbb{E}_{r \sim R(\cdot|s,a)}[r]$ is zero-mean reward noise, and $w_i := \phi_0^\top \widehat{\Sigma}^{-1}\phi(s_i, a_i)$. Since $w_i$ only depends on $s_i, a_i$ and is independent of $r_i$, by treating $s_i, a_i$ as fixed and only considering the randomness of $r_i$ given $s_i, a_i$, we have (see proof techniques from Lemma 1) w.p. $\geq 1 - \delta$,

$$\left|\frac{1}{n}\sum_{i=1}^n w_i \epsilon_i\right| \lesssim \sqrt{\frac{\sum_{i=1}^n w_i^2/n}{n} \log(1/\delta)} \cdot V_{\max}.$$

Eq. (9) follows by noticing that $\sum_{i=1}^n w_i^2/n = \|\phi_0\|_{\widehat{\Sigma}^{-1}}^2$.

A few remarks are in order:

---

[8]Such analyses often assume ridge regularization, i.e., adding $\lambda I$ to $\widehat{\Sigma}$.

**Unknown $\mu_0$.** In Section 2 we assume $\mu_0$ (and thus $\phi_0$) is known for convenience, and Eq. (9) holds without such an assumption. In fact, the most natural setup for contextual bandits is $\mu_0 = \mu^D$, and the states observed in the data can be used to approximate $\phi_0$. In this case, we can replace $\phi_0$ in the above analysis with $\widehat{\phi}_0 := \frac{1}{n} \sum_{i=1}^{n} \phi(s_i, \pi)$. Then, the above bound still holds since $\widehat{\phi}_0$ does not depend on the random reward, but $\widehat{\phi}_0^\top \theta^\star \neq J(\pi)$. This additional error can be handled as $\widehat{\phi}_0^\top \theta^\star - J(\pi) = \frac{1}{n} \sum_{i=1}^{n} (\phi(s_i, \pi)^\top \theta^\star - J(\pi))$, which also enjoys dimension-free concentration.

**Return vs. function estimation.** A key element in the dimension-free guarantee is that we are stating it for a fixed initial distribution $\mu_0$. In contrast, guarantees like Eq. (19) hold for all possible $\mu_0$ simultaneously, since the concentration event in its proof does not depend on $\mu_0$. In this sense, the two results differ in the form of guarantees they offer: Eq. (19) is "w.p. $\geq 1 - \delta$, $\forall \mu_0$", while Eq. (9) is "$\forall \mu_0$, w.p. $\geq 1 - \delta$", and the factor-of-$d$ gap can be explained away by union bounding over all possible $\phi_0$ in Eq. (9). In fact, Eq. (19) (c.f. Theorem 1) is useful for establishing function-estimation guarantees; see Appendix D.

**Comparison to Theorem 1.** Our main theorems for LSTDQ (Theorems 1 and 2) do not subsume Eq. (9) as a special case as the above analysis for Eq. (9) cannot be directly extended to the $\gamma > 0$ regime. This is because the counterpart of $w_i$ will depend on $\widehat{A}$ when $\gamma > 0$, whose definition involves the random next-state $s'$ in the data. On the other hand, the counterpart of $\epsilon_i$ is $Q^\pi(s_i, a_i) - r_i - \gamma Q^\pi(s_i', a_i')$ (see proof of Lemma 1), and $\epsilon_i$ will no longer be zero mean and independent when conditioned on $w_i$. That said, our Theorem 1 is still dimension-free in the main $1/\sqrt{n}$ term, but it comes with a lower-order term that depends on $d$.

**Data adaptivity.** Eq. (9) also crucially relies on the fact that data is collected in a non-adaptive manner, i.e., later actions are not chosen based on the random reward observed in earlier data points, otherwise $\{\epsilon_i\}$ will not be zero mean and independent when conditioned on $\{w_i\}$. In contrast, Eq. (19) can handle adaptively collect data and is often used in online RL where adaptive data collection is inevitable.

**Related works.** While Eq. (9) is generally known in the statistics community, to our knowledge it is less known to the RL theory community, and we believe the idea is worth spreading. There are, however, results that share similar spirits. For example, Duan & Wang (2020) shows that FQE in linear MDPs enjoy a form of guarantee similar to our Eq. (12), where the main $1/\sqrt{n}$ term only depends on coverage and not the dimensionality. Yin & Wang (2020, Lemma 3.4) establish OPE guarantee for finite-horizon tabular RL, where the $1/\sqrt{n}$ term has no polynomial dependence on the number of states and actions (which corresponds to our $d$), though their result measures coverage by a reachability parameter which translates to our $\lambda_{\min}(\Sigma)$.

## B  Proofs of Section 4

### B.1  Proof of Theorem 2

**Theorem 2** (Empirical Coverage Bound)**.** *Under Assumptions 1 and 2, with probability at least $1 - \delta$,*

$$\left| J_{\widehat{Q}_{\mathrm{lstd}}}(\pi) - J(\pi) \right| \lesssim \frac{V_{\max}}{1 - \gamma} \cdot \sqrt{\frac{\widehat{C}_\phi^\pi \cdot (d + \log(1/\delta))}{n}} \tag{12}$$

*where*

$$\widehat{C}_\phi^\pi := (1 - \gamma)^2 \phi_0^\top \widehat{A}^{-1} \widehat{\Sigma} \widehat{A}^{-\top} \phi_0. \tag{13}$$

*We treat $\widehat{C}_\phi^\pi = +\infty$ if $\widehat{A}$ is not invertible.*

**Proof of Theorem 2.** Note that when $\widehat{A}$ is not invertible the guarantee trivially holds due to $\widehat{C}_\phi^\pi = +\infty$, so below we treat $\widehat{A}$ as invertible. This also implies the invertibility of $\widehat{\Sigma}$. Now,

$$\left| J_{\widehat{Q}_{\mathrm{lstd}}}(\pi) - J(\pi) \right| = \left| \mathbb{E}_{s_0 \sim \mu_0, a_0 \sim \pi} \left[ Q^\pi(s_0, a_0) - \widehat{Q}_{\mathrm{lstd}}(s_0, a_0) \right] \right|$$

$$= \left| \mathbb{E}_{s_0 \sim \mu_0, a_0 \sim \pi} \left[ \phi(s_0, a_0)^\top \left( \theta^\star - \widehat{\theta}_{\mathrm{lstd}} \right) \right] \right|,$$

where in the second line we have used realizability (Assumption 1) and the definition of $\widehat{Q}_{\text{lstd}}$. We can continue with simple algebra to find that:

$$
\begin{aligned}
\left|\mathbb{E}_{s_0\sim\mu_0,a_0\sim\pi}\left[\phi(s_0,a_0)^\top\left(\theta^\star-\widehat{\theta}_{\text{lstd}}\right)\right]\right| &= \left|\phi_0^\top\left(\theta^\star-\widehat{\theta}_{\text{lstd}}\right)\right| \\
&= \left|\phi_0^\top\widehat{A}^{-1}\left(\widehat{A}\theta^\star-\widehat{A}\widehat{\theta}_{\text{lstd}}\right)\right| \\
&= \left|\phi_0^\top\widehat{A}^{-1}\widehat{\Sigma}^{1/2}\widehat{\Sigma}^{-1/2}\left(\widehat{A}\theta^\star-\widehat{A}\widehat{\theta}_{\text{lstd}}\right)\right| \\
&\leq \left\|\widehat{\Sigma}^{1/2}\widehat{A}^{-\top}\phi_0\right\|_2\left\|\widehat{\Sigma}^{-1/2}\left(\widehat{A}\theta^\star-\widehat{b}\right)\right\|_2,
\end{aligned}
$$

where in the last line we have used Cauchy-Schwartz, and $\widehat{A}\widehat{\theta}_{\text{lstd}}=\widehat{b}$ (by invertibility). We then note that

$$
\left\|\widehat{\Sigma}^{1/2}\widehat{A}^{-\top}\phi_0\right\|_2 = \sqrt{\phi_0^\top\widehat{A}^{-1}\widehat{\Sigma}\widehat{A}^{-\top}\phi_0} = \frac{1}{1-\gamma}\sqrt{\widehat{C}_\phi^\pi},
$$

which yields that

$$
\left|J_{\widehat{Q}_{\text{lstd}}}(\pi)-J(\pi)\right| \leq \frac{1}{1-\gamma}\sqrt{\widehat{C}_\phi^\pi}\left\|\widehat{\Sigma}^{-1/2}\left(\widehat{A}\theta^\star-\widehat{b}\right)\right\|_2.
$$

Then, Eq. (12) will be obtained by establishing the following concentration lemma.

**Lemma 1.** *Fix any unit vector $u$ (i.e., $\|u\|_2=1$), with probability at least $1-\delta$,*

$$
|u^\top\widehat{\Sigma}^{-1/2}(\widehat{A}\theta^\star-\widehat{b})| \lesssim V_{\max}\sqrt{\frac{\log(1/\delta)}{n}},
$$

*where the guarantee is treated as vacuous if $\widehat{\Sigma}$ is not invertible. As a corollary, w.p. $\geq 1-\delta$,*

$$
\|\widehat{\Sigma}^{-1/2}(\widehat{A}\theta^\star-\widehat{b})\|_2 \lesssim V_{\max}\sqrt{\frac{d+\log(1/\delta)}{n}}.
$$

$\square$

**Proof of Lemma 1.** We first rewrite $\widehat{A}\theta^\star-\widehat{b}$ as the average of terms across the data points. Here we temporarily use subscript $i$ to denote the $i$-th data point $(s_i,a_i,r_i,s_i')$ (in most of the main text we use subscript to denote the time step of a state or action, such as in $s_t$):

$$
\begin{aligned}
\widehat{A}\theta^\star-\widehat{b} &= \frac{1}{n}\sum_{i=1}^n\phi(s_i,a_i)\left(\phi(s_i,a_i)^\top\theta^\star-\gamma\phi(s_i',a_i')^\top\theta^\star-r_i\right) \\
&= \frac{1}{n}\sum_{i=1}^n\phi(s_i,a_i)\big(\underbrace{Q^\pi(s_i,a_i)-r_i-\gamma Q^\pi(s_i',a_i')}_{:=\varepsilon_i}\big).
\end{aligned}
$$

Thus, $\varepsilon_i$'s are independent zero-mean random variables when conditioned on the $(s_i,a_i)$ pairs. We will treat the design over $s_i,a_i$ as fixed and non-random and only consider the case when $\widehat{\Sigma}$ is invertible, since we provide vacuous guarantee otherwise. Note that the invertibility of $\widehat{\Sigma}$ (which is determined by the $(s_i,a_i)$ pairs) will not affect the validity of our concentration inequality below due to the fixed-design analysis.

Let $l := \widehat{\Sigma}^{-1/2}(\widehat{A}\theta^\star-\widehat{b})$, and

$$
u^\top l := u^\top\widehat{\Sigma}^{-1/2}\cdot\frac{1}{n}\sum_{i=1}^n\phi_i\epsilon_i = \frac{1}{n}\sum_{i=1}^n u^\top\widehat{\Sigma}^{-1/2}\phi_i\epsilon_i,
$$

where $\phi_i$ is a shorthand for $\phi(s_i,a_i)$. The expression is the average of $n$ independently distributed zero-mean random variables. Since $\epsilon_i$'s are the only randomness in this lemma and $|\epsilon_i|\leq 2V_{\max}$, each summand's boundedness is

$$
a_i := |u^\top\widehat{\Sigma}^{-1/2}\phi_i|\cdot 2V_{\max}.
$$

We now invoke Hoeffding's inequality for unequal ranges (Wainwright, 2019), which depends on the sum of range squared:

$$\sum_{i=1}^{n} a_i^2 = 4V_{\max}^2 \sum_{i=1}^{n} u^\top \widehat{\Sigma}^{-1/2} \phi_i \phi_i^\top \widehat{\Sigma}^{-1/2} u = 4V_{\max}^2 \cdot u^\top \widehat{\Sigma}^{-1/2} (n\widehat{\Sigma}) \widehat{\Sigma}^{-1/2} u = 4V_{\max}^2 n.$$

While individual $a_i$ can be large ($\|\widehat{\Sigma}^{-1/2}\phi_i\|_2$ can be as large as $\sqrt{n}$), the sum of their squares is $4V_{\max}^2 n$, which is identical to the situation when each individual summand is bounded by $2V_{\max}$, and Hoeffding's inequality does not distinguish between them. So w.p. $\geq 1 - \delta$,

$$|u^\top l| \lesssim V_{\max} \sqrt{\frac{1}{n} \log \frac{1}{\delta}}.$$

This proves the first statement. For the second statement, note that $\|l\|_2 = \sup_{\|u\|_2=1} u^\top l$, so it suffices to union bound over $\{u : \|u\|_2 = 1\}$ by a covering argument. Let $\mathcal{U} \subset \{u : \|u\|_2 = 1\}$ be a cover to be specified later. By union bound, w.p. $\geq 1 - \delta$,

$$\max_{u' \in \mathcal{U}} |u'^\top l| \lesssim V_{\max} \sqrt{\frac{1}{n} \log \frac{|\mathcal{U}|}{\delta}}.$$

Now for any unit $u$, let $u'$ be the closest vector in $\mathcal{U}$, and

$$u^\top l \leq |(u' + u - u')^\top l| \leq |u'^\top l| + (u - u')^\top l \leq |u'^\top l| + \|l\|_2 \|u - u'\|_2.$$

Taking supremum over $u$ on both hands, we obtain

$$\|l\|_2 = \sup_{\|u\|_2=1} u^\top l \leq \max_{u' \in \mathcal{U}} |u'^\top l| + \|l\|_2 \|u - u'\|_2,$$

so

$$\|l\|_2 \leq \frac{\max_{u' \in \mathcal{U}} |u'^\top l|}{1 - \|u - u'\|_2}.$$

We now ensure that the multiplicative factor $1/(1 - \|u - u'\|_2) \leq 2$ by building a cover such that $\|u - u'\|_2 \leq 1/2$, and this only requires a constant resolution, so $|\mathcal{U}| = \mathcal{O}(1)^d$. Combining the results so far and the lemma statement follows immediately. $\qquad\square$

## B.2 PROOF OF THEOREM 1

**Theorem 1** (Population Coverage Bound). *There exists $n_0$ such that when $n \geq n_0$, w.p. $\geq 1 - \delta$,*

$$\left| J_{\widehat{Q}_{\mathrm{lstd}}}(\pi) - J(\pi) \right| \lesssim \frac{V_{\max}}{1 - \gamma} \sqrt{\frac{C_\phi^\pi \cdot \log(1/\delta)}{n}} + o(\sqrt{1/n}), \tag{10}$$

*where*

$$C_\phi^\pi := (1 - \gamma)^2 \phi_0^\top A^{-1} \Sigma A^{-\top} \phi_0 \tag{11}$$

*and $n_0$ and the $o(1/\sqrt{n})$ term may depend on $d$ and $1/\sigma_{\min}(A)$.*

**Proof of Theorem 1.** We obtain this from Theorem 2 by showing that $\widehat{C}_\phi^\pi$ can be replaced by $C_\phi^\pi$ at the cost of burn-in and lower-order terms. Note the following lemma, which we prove afterwards.

**Lemma 2.** *There exists $n_0 \approx \left( \frac{B_\phi^2 + \sigma_{\max}(A)}{\sigma_{\min}(A)} \right)^2 \log(d/\delta)$ such that when $n \geq n_0$, w.p. $\geq 1 - \delta$, $\widehat{A}$ and $\widehat{\Sigma}$ are invertible, and*

$$\left| C_\phi^\pi - \widehat{C}_\phi^\pi \right| \leq (1 - \gamma)^2 \left\| \Sigma^{1/2} A^{-\top} \phi_0 - \widehat{\Sigma}^{1/2} \widehat{A}^{-\top} \phi_0 \right\|_2 = \mathcal{O}(1/n^{1/4}),$$

*where the $\mathcal{O}$ hides factors of $\frac{1}{\sigma_{\min}(A)}$.*

We also recall Lemma 1. We union bound over the 3 events across these two lemmas, each assigning $\delta/3$ failure probability: note that Lemma 1 contains two statements, and we require the first holds with $u$ being the normalized vector of $\Sigma^{1/2} A^{-\top} \phi_0$. This way, Lemma 1 guarantees that:

$$|\phi_0^\top A^{-1} \Sigma^{-1/2} \widehat{\Sigma}^{-1/2} (\widehat{A}\theta^\star - \widehat{b})| \lesssim \|\phi_0^\top A^{-1} \Sigma^{-1/2}\|_2 \cdot V_{\max} \sqrt{\frac{\log(1/\delta)}{n}}, \qquad (20)$$

$$\|\widehat{\Sigma}^{-1/2} (\widehat{A}\theta^\star - \widehat{b})\|_2 \lesssim V_{\max} \sqrt{\frac{d + \log(1/\delta)}{n}}. \qquad (21)$$

Under the above high-probability events, we have $\widehat{A}$ and $\widehat{\Sigma}$ invertible, and

$$\left| J_{\widehat{Q}_{\mathrm{lstd}}}(\pi) - J(\pi) \right| = |\phi_0^\top (\theta^\star - \widehat{\theta})| = |\phi_0^\top \widehat{A}^{-1} \widehat{\Sigma}^{1/2} \widehat{\Sigma}^{-1/2} (\widehat{A}\theta^\star - \widehat{b})|$$

$$\leq \underbrace{|(\phi_0^\top \widehat{A}^{-1} \widehat{\Sigma}^{1/2} - \phi_0^\top A^{-1} \Sigma^{1/2}) \cdot \widehat{\Sigma}^{-1/2}(\widehat{A}\theta^\star - \widehat{b})|}_{(I)} + \underbrace{|\phi_0^\top A^{-1} \Sigma^{1/2} \cdot \widehat{\Sigma}^{-1/2}(\widehat{A}\theta^\star - \widehat{b})|}_{(II)}.$$

Term (II) is exactly the LHS of Eq. (20). Noting that $\|\phi_0^\top A^{-1} \Sigma^{-1/2}\|_2 = C_\phi^\pi / (1 - \gamma)$, this immediately gives us the $1/\sqrt{n}$ term in Eq. (10). It remains to show that term (I) is $o(\sqrt{1/n})$.

$$(I) \leq \|\phi_0^\top \widehat{A}^{-1} \widehat{\Sigma}^{1/2} - \phi_0^\top A^{-1} \Sigma^{1/2}\|_2 \, \|\widehat{\Sigma}^{-1/2}(\widehat{A}\theta^\star - \widehat{b})\|_2.$$

Eq.(21) shows that $\|\widehat{\Sigma}^{-1/2}(\widehat{A}\theta^\star - \widehat{b})\|_2 = \mathcal{O}(1/\sqrt{n})$, and Lemma 2 shows that $\|\phi_0^\top \widehat{A}^{-1} \widehat{\Sigma}^{1/2} - \phi_0^\top A^{-1} \Sigma^{1/2}\|_2 = \mathcal{O}(1/n^{1/4})$. Put together, term (I) is $\mathcal{O}(1/n^{3/4})$ which proves the claim.

$\square$

**Proof of Lemma 2.** We first give a matrix concentration lemma, whose proof is postponed until the end of this lemma.

**Lemma 3.** *When* $n \gtrsim \left( \frac{B_\phi^2 + \sigma_{\max}(A)}{\sigma_{\min}(A)} \right)^2 \log(d/\delta)$, *with probability at least* $1 - \delta$, *we have:*

$$\|\Sigma - \widehat{\Sigma}\|_2 \lesssim \epsilon(\Sigma) := \sqrt{\frac{\lambda_{\max}(\Sigma)(B_\phi^2 + \lambda_{\max}(\Sigma)) \log(d/\delta)}{n}}. \qquad (22)$$

*and*

$$\|A - \widehat{A}\|_2 \lesssim \epsilon(A) := \left( B_\phi^2 + \sigma_{\max}(A) \right) \sqrt{\frac{\log(d/\delta)}{n}},$$

*and*

$$\|\widehat{A}^{-1} - A^{-1}\|_2 \leq \frac{1}{\sigma_{\min}(A)^2} \|A - \widehat{A}\|_2 = \mathcal{O}\left( \frac{B_\phi^2 + \sigma_{\max}(A)}{\sigma_{\min}(A)^2} \sqrt{\frac{\log(d/\delta)}{n}} \right). \qquad (23)$$

*As a consequence,* $\widehat{A}$ *and* $\widehat{\Sigma}$ *are invertible with high probability.*

Since $\widehat{A}$ and $\widehat{\Sigma}$ are invertible, we can write:

$$\left| C_\phi^\pi - \widehat{C}_\phi^\pi \right| = (1 - \gamma)^2 \left| \|\Sigma^{1/2} A^{-\top} \phi_0\|_2 - \|\widehat{\Sigma}^{1/2} \widehat{A}^{-\top} \phi_0\|_2 \right|$$

$$\leq (1 - \gamma)^2 \left\| \Sigma^{1/2} A^{-\top} \phi_0 - \widehat{\Sigma}^{1/2} \widehat{A}^{-\top} \phi_0 \right\|_2$$

$$\leq (1 - \gamma)^2 \left( \left\| \widehat{\Sigma}^{1/2} \left( A^{-\top} - \widehat{A}^{-\top} \right) \phi_0 \right\|_2 + \left\| \left( \widehat{\Sigma}^{1/2} - \Sigma^{1/2} \right) A^{-\top} \phi_0 \right\|_2 \right)$$

$$\leq (1 - \gamma)^2 B_\phi \left( \left\| \widehat{\Sigma}^{1/2} \right\|_2 \|A^{-1} - \widehat{A}^{-1}\|_2 + \left\| \widehat{\Sigma}^{1/2} - \Sigma^{1/2} \right\|_2 \|A^{-1}\|_2 \right).$$

The second line is the intermediate quantity in the lemma statement and the rest of the analysis always bounds $|C_\phi^\pi - \widehat{C}_\phi^\pi|$ through the above relaxation, so we will not separately mention the bound on $\left\| \Sigma^{1/2} A^{-\top} \phi_0 - \widehat{\Sigma}^{1/2} \widehat{A}^{-\top} \phi_0 \right\|_2$.

Let $\varepsilon(\Sigma^{1/2}) = \|\Sigma^{1/2} - \widehat{\Sigma}^{1/2}\|_2$ and $\varepsilon(A^{-1}) = \|A^{-1} - \widehat{A}^{-1}\|_2$. Note that the above inequalities imply

$$\left| C_\phi^\pi - \widehat{C}_\phi^\pi \right| \leq (1-\gamma)^2 B_\phi \left( \left( \lambda_{\max}(\Sigma^{1/2}) + \varepsilon(\Sigma^{1/2}) \right) \varepsilon(A^{-1}) + \varepsilon(\Sigma^{1/2}) \frac{1}{\sigma_{\min}(A)} \right). \qquad (24)$$

We conclude by bounding $\varepsilon(\Sigma^{1/2})$ and $\varepsilon(A^{-1})$. The bound on $\varepsilon(A^{-1})$ follows from (23). To obtain a bound on $\varepsilon(\Sigma^{1/2})$, we can use (22) in combination with the the inequality $\|\Sigma^{1/2} - \widehat{\Sigma}^{1/2}\|_2 \leq \sqrt{\|\Sigma - \widehat{\Sigma}\|_2}$ (van Hemmen & Ando, 1980)[9], to obtain:

$$\|\Sigma^{1/2} - \widehat{\Sigma}^{1/2}\|_2 \leq \sqrt{\|\Sigma - \widehat{\Sigma}\|_2} \leq \sqrt{\epsilon(\Sigma)} = \mathcal{O}\left( \left( \frac{\lambda_{\max}(\Sigma)(B_\phi^2 + \lambda_{\max}(\Sigma))\log(d/\delta)}{n} \right)^{1/4} \right).$$

Returning to Eq. (24) and combining everything, we have:

$$\left| C_\phi^\pi - \widehat{C}_\phi^\pi \right| \leq \frac{(1-\gamma)^2 B_\phi \left( B_\phi^2 + \sigma_{\max}(A) \right)}{\sigma_{\min}(A)^2} \sqrt{\frac{\log(d/\delta)}{n}} \left( \left( \frac{\lambda_{\max}(\Sigma)(B_\phi^2 + \lambda_{\max}(\Sigma))\log(d/\delta)}{n} \right)^{1/4} \right.$$
$$\left. + \sqrt{\lambda_{\max}(\Sigma)} \right)$$
$$+ \frac{(1-\gamma)^2 B_\phi}{\sigma_{\min}(A)} \left( \frac{\lambda_{\max}(\Sigma)(B_\phi^2 + \lambda_{\max}(\Sigma))\log(d/\delta)}{n} \right)^{1/4}$$
$$= \mathcal{O}(1/n^{1/4}).$$

$\square$

**Proof of Lemma 3.** We firstly establish the bound on $\|\widehat{\Sigma} - \Sigma\|_2$. To do this, we use Matrix Bernstein (Lemma 8). Abbreviate $X_i := \phi(s_i, a_i)$, and let $Z_i = X_i X_i^\top - \Sigma$ be the centered matrices. For the almost sure bound, we have

$$\|Z_i\|_2 \leq \|X_i X_i^\top\|_2 + \|\Sigma\|_2 \leq \|X_i\|_2^2 + \lambda_{\max}(\Sigma) \leq B_\phi^2 + \lambda_{\max}(\Sigma).$$

For the variance term, we have:

$$\begin{aligned} \left\| \mathbb{E}\left[ (X_i X_i^\top - \Sigma)^2 \right] \right\|_2 &= \left\| \mathbb{E}\left[ (X_i X_i^\top)^2 \right] - \Sigma^2 \right\|_2 \\ &\leq \left\| B_\phi^2 \mathbb{E}\left[ X_i X_i^\top \right] - \Sigma^2 \right\|_2 \\ &\leq B_\phi^2 \lambda_{\max}(\Sigma) + \lambda_{\max}(\Sigma)^2. \end{aligned}$$

This yields

$$\begin{aligned} \|\widehat{\Sigma} - \Sigma\|_2 &\leq \sqrt{\frac{2\lambda_{\max}(\Sigma)(B_\phi^2 + \lambda_{\max}(\Sigma))\log(2d/\delta)}{n}} + \frac{2(B_\phi^2 + \lambda_{\max}(\Sigma))\log(2d/\delta)}{3n} \\ &\lesssim \sqrt{\frac{\lambda_{\max}(\Sigma)(B_\phi^2 + \lambda_{\max}(\Sigma))\log(d/\delta)}{n}}. \end{aligned}$$

We now establish the bound on $\|\widehat{A} - A\|_2$ again via Matrix Bernstein (Lemma 8). Define the notation $X_i = \phi(s_i, a_i)$ and $X_i' = \phi(s_i, a_i) - \gamma\phi(s_i', a_i')$. Then we let $Z_i = X_i(X_i')^\top - A$ denote the centered matrices. For the almost sure norm bound, we have

$$\|Z_i\|_2 \leq \|X_i\|_2 \|X_i'\|_2 + \|A\|_2 \leq B_\phi^2(1+\gamma) + \sigma_{\max}(A) \leq 2B_\phi^2 + \sigma_{\max}(A).$$

For the variance terms, we have:

$$\begin{aligned} \left\| \mathbb{E}\left[ \left( X_i(X_i')^\top - A \right) \left( X_i(X_i')^\top - A \right)^\top \right] \right\|_2 &= \left\| \mathbb{E}\left[ X_i(X_i')^\top X_i' X_i^\top - AX_i' X_i^\top - X_i' X_i^\top A + AA^\top \right] \right\|_2 \\ &\leq 4B_\phi^2 \left\| \mathbb{E}\left[ X_i X_i^\top \right] \right\|_2 + \left\| AA^\top \right\|_2 \\ &= 4B_\phi^2 \lambda_{\max}(\Sigma) + \sigma_{\max}(A)^2, \end{aligned}$$

---

[9]See also this answer by user "jlewk" on Math StackExchange: https://math.stackexchange.com/a/3968174

as well as

$$\left\|\mathbb{E}\left[\left(X_i(X_i')^\top - A\right)^\top \left(X_i(X_i')^\top - A\right)\right]\right\|_2 = \left\|\mathbb{E}\left[X_i'(X_i)^\top X_i(X_i')^\top - A^\top X_i(X_i')^\top - X_i'X_i^\top A + A^\top A\right]\right\|_2$$
$$\leq 4B_\phi^4 + \left\|AA^\top\right\|_2$$
$$= 4B_\phi^4 + \sigma_{\max}(A)^2.$$

With the latter, the variance term and the norm bound are of the same order, which gives

$$\|\widehat{A} - A\|_2 \lesssim \left(B_\phi^2 + \sigma_{\max}(A)\right)\sqrt{\frac{\log(d/\delta)}{n}}.$$

Then, to bound $\|A - \widehat{A}^{-1}\|_2$, we note the following lemma.

**Lemma 4** ((Stewart & Sun, 1990)). *Let $A \in \mathbb{R}^{m \times n}$, with $m \geq n$ and let $\widetilde{A} = A + E$. Then,*

$$\epsilon(A^{-1}) \leq \frac{1+\sqrt{5}}{2}\max\left\{\|A^{-1}\|_2, \|\widetilde{A}^{-1}\|_2\right\}\|E\|_2.$$

*Furthermore, if $\|E\|_2 \leq \sigma_{\min}(A)/2$, then*

$$\|\widetilde{A}^{-1} - A^{-1}\|_2 \lesssim \|A^{-1}\|_2^2\|E\|_2.$$

This immediately implies that, for $\|A - \widehat{A}\|_2 \leq \sigma_{\min}(A)/2$, we have

$$\|\widehat{A}^{-1} - A^{-1}\|_2 \leq \frac{1}{\sigma_{\min}(A)^2}\|A - \widehat{A}\|_2 = \mathcal{O}\left(\frac{B_\phi^2 + \sigma_{\max}(A)}{\sigma_{\min}(A)^2}\sqrt{\frac{\log(d/\delta)}{n}}\right)$$

This latter condition is equivalent to

$$\left(B_\phi^2 + \sigma_{\max}(A)\right)\sqrt{\frac{\log(d/\delta)}{n}} \lesssim \frac{\sigma_{\min}(A)}{2} \implies n \gtrsim \left(\frac{B_\phi^2 + \sigma_{\max}(A)}{\sigma_{\min}(A)}\right)^2 \log(d/\delta),$$

which we have set as the burn-in time in the lemma statement. Lastly, Eq. (23) and the reverse triangle inequality shows that $\widehat{A}$ (and thus $\widehat{\Sigma}$) are invertible with high probability. $\qquad\square$

## C  PROOFS OF SECTION 5

### C.1  PROOF OF PROPOSITION 1

**Proposition 1.** *Define a deterministic linear dynamical system $\{\mu_{\phi,t}^\pi\}_{t\geq 0}$, with $\mu_{\phi,0}^\pi := \phi_0$, and $\forall t \geq 0$,*

$$\mu_{\phi,t+1}^\pi = B^\pi \, \mu_{\phi,t}^\pi.$$

*When $\rho(B^\pi) < 1/\gamma$,[10] define the feature occupancy in $B^\pi$ as $\mu_\phi^\pi := (1-\gamma)\sum_{t\geq 0}\gamma^t\mu_{\phi,t}^\pi$, then*

$$C_\phi^\pi = (\mu_\phi^\pi)^\top\Sigma^{-1}\mu_\phi^\pi.$$

**Proof of Proposition 1.** Recall that we defined $B^\pi = (\Sigma^{-1}\Sigma_{\mathrm{cr}})^\top$. We note that

$$A = \Sigma - \gamma\Sigma_{\mathrm{cr}} = \Sigma(I - \gamma(B^\pi)^\top).$$

Substituting this into $C_\phi^\pi$, we arrive at the expression:

$$C_\phi^\pi = (1-\gamma)^2\phi_0^\top A^{-1}\Sigma A^{-T}\phi_0 = (1-\gamma)^2\phi_0^\top(I - \gamma B^\pi)^{-\top}\Sigma^{-1}(I - \gamma B^\pi)^{-1}\phi_0. \qquad (25)$$

Note that when $\rho(B^\pi) < 1/\gamma$, the matrix $(I - \gamma B^\pi)^{-1}$ has the series expansion:

$$(I - \gamma B^\pi)^{-1} = \sum_{t=0}^\infty \gamma^t(B^\pi)^t.$$

Thus, we notice that

$$(I - \gamma B^\pi)^{-1}\phi_0 = \sum_{t=0}^\infty \gamma^t(B^\pi)^t\phi_0 = \sum_{t=0}^\infty \gamma^t\mu_{\phi,t}^\pi = \frac{1}{1-\gamma}\mu_\phi^\pi.$$

Substituting this into Eq. (25) gives the result. $\qquad\square$

---

[10]Recall that $\rho$ denotes the spectral radius of a matrix.

## C.2 NEW ON-POLICY CONDITION

Prior to our work, the only notable discussion of coverage in LSTD(Q) is its benign guarantees in a strict on-policy setting, where data distribution $\mu^D$ is an invariant distribution of $P^\pi$. One can further relax this condition to $\Sigma_{\mathrm{cr}} \preceq \beta\Sigma$ as $\beta = 1$ when $\mu^D$ is invariant under $\pi$. This condition is handled by Perdomo et al. (2023) and inherited by our results.

Given the dynamical system interpretation in Section 5.1, we are able to establish a novel on-policy result as follows:

**Proposition 2.** *Assume $\phi$ contains a bias term, that is, there exists $\theta_0 \in \mathbb{R}^d$ such that $\phi(s,a)^\top \theta_0 \equiv 1 \,\forall s, a$. If $\rho(B^\pi) < 1/\gamma$ and $\mathbb{E}_{\mu^D}[\phi(s,a)] = \mathbb{E}_{\mu^D}[\phi(s',\pi)] = \phi_0$, then $C_\phi^\pi \le 1$.*

The condition $\Sigma_{\mathrm{cr}} \preceq \beta\Sigma$ relates the *distributions* of $(s,a)$ and $(s',a')$, where Proposition 2 only requires their means to be identical. We provide a proof below.

**Proof of Proposition 2.** Let $(\phi(s,a) \mapsto Y)$ be any regression problem where $(s,a) \sim \mu^D$ and $Y$ is an arbitrary real-valued random variable jointed distributed with $(s,a)$. Let $\theta_Y = \Sigma^{-1}\mathbb{E}[\phi(s,a)Y]$ (where $\mathbb{E}[\cdot]$ here is w.r.t. the above joint distribution) be the population solution to linear regression. Then, when $\phi$ contains a bias term, it is known that

$$\mathbb{E}[\phi(s,a)^\top \theta_Y] = \mathbb{E}[Y],$$

even if $\mathbb{E}[Y|s,a]$ is not linear in $\phi$. This can be proved easily by multiplying $\theta_0$ on both sides of $\Sigma \times \theta_Y = \mathbb{E}[\phi(s,a)Y]$:

$$\theta_0^\top \mathbb{E}[\phi(s,a)\phi(s,a)^\top]\theta_Y = \theta_0^\top \mathbb{E}[\phi(s,a)Y] \;\Rightarrow\; \mathbb{E}[\phi(s,a)^\top \theta_Y] = \mathbb{E}[Y].$$

Since $B^\pi$ linearly predicts each coordinate of $\phi(s',a')$ separately, we apply the above result and obtain

$$\mathbb{E}_{\mu^D}[\phi(s',a')] = \mathbb{E}_{\mu^D}[B^\pi \phi(s,a)] = B^\pi \mathbb{E}_{\mu^D}[\phi(s,a)].$$

Let $\phi_D = \mathbb{E}_{\mu^D}[\phi(s,a)]$. Then the condition of the proposition tells us that

$$\phi_D = B^\pi \phi_D.$$

Since $\mu_{\phi,0}^\pi = \phi_D$, we immediately have $\mu_{\phi,t}^\pi = \phi_D, \forall t$. Given $\rho(B^\pi) < 1/\gamma$, Proposition 1 holds and $\mu_\phi^\pi = \phi_D$, so

$$C_\phi^\pi = (\phi_D)^\top \Sigma^{-1}\phi_D \le 1.$$

$\square$

## C.3 PROOF OF PROPOSITION 3

**Proposition 3.** *We have that:*

$$\sqrt{C_\phi^\pi} = (1-\gamma)\|(I - \gamma\Sigma^{-1/2}\Sigma_{\mathrm{cr}}\Sigma^{-1/2})^{-\top}\Sigma^{-1/2}\phi_0\|_2 \le (1-\gamma)\frac{\|\phi_0\|_{\Sigma^{-1}}}{\sigma_{\min}(I - \gamma\Sigma^{-1/2}\Sigma_{\mathrm{cr}}\Sigma^{-1/2})}.$$

*Furthermore, the gap between the LHS and the RHS may be arbitrarily large.*

**Proof of Proposition 3.** The derivation of the equality is as follows:

$$\frac{1}{(1-\gamma)^2}C_\phi^\pi = \phi_0^\top A^{-1}\Sigma A^{-T}\phi_0$$

$$= \phi_0(\Sigma - \gamma\Sigma_{\mathrm{cr}})^{-1}\Sigma(\Sigma - \gamma\Sigma_{\mathrm{cr}})^{-\top}\phi_0$$

$$= \phi_0^\top \left(\Sigma^{1/2}(I - \gamma\Sigma^{-1/2}\Sigma_{\mathrm{cr}}\Sigma^{-1/2})\Sigma^{1/2}\right)^{-1}\Sigma\left(\Sigma^{1/2}(I - \gamma\Sigma^{-1/2}\Sigma_{\mathrm{cr}}\Sigma^{-1/2})\Sigma^{1/2}\right)^{-\top}\phi_0$$

$$= \phi_0^\top \Sigma^{-1/2}(I - \gamma\Sigma^{-1/2}\Sigma_{\mathrm{cr}}\Sigma^{-1/2})^{-1}(I - \gamma\Sigma^{-1/2}\Sigma_{\mathrm{cr}}\Sigma^{-1/2})^{-\top}\Sigma^{-1/2}\phi_0$$

$$= \|(I - \gamma\Sigma^{-1/2}\Sigma_{\mathrm{cr}}\Sigma^{-1/2})^{-\top}\Sigma^{-1/2}\phi_0\|_2^2.$$

For the inequality, we simply take the operator norm on the matrix $(I - \gamma\Sigma^{-1/2}\Sigma_{\mathrm{cr}}\Sigma^{-1/2})^{-\top}$:

$$\|(I - \gamma\Sigma^{-1/2}\Sigma_{\mathrm{cr}}\Sigma^{-1/2})^{-\top}\Sigma^{-1/2}\phi_0\|_2 \le \left\|(I - \gamma\Sigma^{-1/2}\Sigma_{\mathrm{cr}}\Sigma^{-1/2})^{-\top}\right\|_2 \left\|\Sigma^{-1/2}\phi_0\right\|_2$$

$$= \frac{\|\phi_0\|_{\Sigma^{-1}}}{\sigma_{\min}(I - \gamma\Sigma^{-1/2}\Sigma_{\mathrm{cr}}\Sigma^{-1/2})}.$$

The fact that this represents an arbitrarily large improvement follows by picking an instance where $I - \gamma\Sigma^{-1/2}\Sigma_{\mathrm{cr}}\Sigma^{-1/2}$ is near-singular but $\Sigma^{-1/2}\phi_0$ is in a non-singular direction. For example, consider the instance where $d = 2$, the initial distribution $\mu_0$ is deterministic on a state with feature $e_2$, the data distribution $\mu^D$ is uniform over the feature vectors $\sqrt{2}e_1$ and $\sqrt{2}e_2$, and all the states in the data distribution transition to a state with feature vector $\phi' = \frac{2}{\gamma}(1 - \varepsilon)e_1$. Then, it is simple to verify that $\phi_0 = e_2$, $\Sigma = I$, $\Sigma_{\mathrm{cr}} = \frac{1}{\gamma}\begin{pmatrix} 1 - \varepsilon & 0 \\ 0 & 0 \end{pmatrix}$, and $I - \gamma\Sigma^{-1/2}\Sigma_{\mathrm{cr}}\Sigma^{-1/2} = \begin{pmatrix} \varepsilon & 0 \\ 0 & 1 \end{pmatrix}$. Thus, the LHS is equal to 1 but the RHS is equal to $1/\varepsilon$, which can be made arbitrarily large by taking $\varepsilon \to_+ 0$. $\qquad\square$

### C.4 On Aggregated Concentrability

Section 5.2 mention that prior works establish aggregated concentrability as the coverage parameter for learning with $Q^\pi$-irrelevant abstractions. However, these works focus on the model selection setting where learning under abstraction is used as a subroutine, and the notion of aggregated concentrability is either implicit in the analysis or not given in a clean form (Xie & Jiang, 2020b; Zhang & Jiang, 2021; Jia et al., 2024). Therefore, here we provide a clean sketch for learning under abstractions based on the spirit of these earlier works.

The key observation is that **learning can be viewed as entirely happening in $M_\psi$ (Definition 1)**:[11]

1. **Data:** The data-generating process is identical to that from $M_\psi$, in the sense that the following two generation process for $(k, a, r, k')$ are identically distributed:

$$(s, a) \sim \mu^D, r \sim R(\cdot|s, a), r' \sim P(\cdot|s, a), k = \psi(s), k' = \psi(s')$$
$$\iff (k, a) \sim \phi^D, r \sim R_\psi(s, a), k' \sim P_\psi(\cdot|k, a),$$

   where we recall that $\phi(s, a) = \mathbf{e}_{\psi(s), a}$ in this setting and $\phi^D = \mathbb{E}_{(s,a) \sim \mu^D}[\phi(s, a)]$ is a distribution over $\{1, \ldots, K\} \times \mathcal{A}$.

2. **Learning algorithm:** LSTDQ is equivalent to tabular model-based OPE in the empirical estimate of $M_\psi$. That is, $\widehat{\theta}_{\mathrm{lstd}}$ is precisely $Q^\pi_{\widehat{M}_\psi}$, where $\widehat{M}_\psi$ is the tabular MLE estimation of $M_\psi$ obtained from data $\{(k, a, r, k')\}$.

3. **Learning target:** Since $Q^\pi$ is realizable by the abstraction, we can write its "compressed version" $[Q^\pi]_\psi(k, a) = Q^\pi(s, a), \forall s \in \psi^{-1}(k)$. $[Q^\pi]_\psi$ is precisely the Q-function of $\pi$ (recall that in Section 5.2 we have $\pi(\cdot|s) = \pi(\cdot|\psi(s))$) in $M_\psi$, namely $[Q^\pi]_\psi = Q^\pi_{M_\psi}$ (Jiang, 2018; Amortila et al., 2024a).

4. **Error measure:** The return estimation error guarantees also follow a similar translation: let $\psi_0$ in Definition 1 be the initial distribution in $M_\psi$,

$$J(\pi) - J_{\widehat{Q}_{\mathrm{lstd}}}(\pi) = J_{M_\psi}(\pi) - J_{\widehat{M}_\psi}(\pi).$$

Given the equivalence, we can simply invoke any guarantee for $J_{M_\psi}(\pi) - J_{\widehat{M}_\psi}(\pi)$ and it will equally apply to $J(\pi) - J_{\widehat{Q}_{\mathrm{lstd}}}(\pi)$. However, the former is a textbook-standard instance of tabular model-based OPE, where a standard coverage parameter is $\chi^2$ of the density ratio w.r.t. $M_\psi$ as the groundtruth model and $\phi^D$ as the data distribution. This yields the aggregated concentrability in Definition 1.

---

[11]Definition 1 ommited the definition of reward function $R_\psi$. For stochastic rewards, the probability distribution $R_\psi(r|k, a)$ is a mixture of $R(r|s, a)$ similar to the transition case: $R_\psi(r|k, a) = \frac{\mu^D(s, a)R(r|s, a)}{\phi^D(k, a)}$.

**General state-action aggregation.** Definition 1 is restricted to $\pi$ that is consistent with the given abstraction $\psi$. Here we show a more general analysis for arbitrary $\pi$, which also handles general one-hot $\phi$ that are not necessarily induced from state abstraction $\psi$, corresponding to an arbitrary size-$d$ partition over $\mathcal{S} \times \mathcal{A}$. In this case, an abstract MDP analogous to $M_\psi$ (with $\{1, \ldots, K\}$ as its state space) is not well defined; nevertheless, the notion of aggregated concentrability can be extended to handle this setting, and its equivalence to our $C_\phi^\pi$ still holds. We briefly sketch the analyses here, which includes (1) an analysis of learning under general one-hot $\phi$, (2) showing that the guarantee depends on a more general notion of aggregated concentrability, and (3) it is equivalent to our $C_\phi^\pi$. There are two approaches that both achieve the goal.

**Approach 1.** This approach directly extends the analysis above and is similar to Jia et al. (2024). Instead of an MDP over $\{1, \ldots, K\}$ (with $K = d/|\mathcal{A}|$) as its state space, we can directly define an abstract Markov reward process (MRP) $(R_\phi, P_\phi)$ with state space $[d]$, and obtain the similar data-generation equivalence: (for this analysis we will abuse the notation and treat the output of $\phi$ as its one-hot index, $\phi : \mathcal{S} \times \mathcal{A} \to \{1, \ldots, d\}$) let $\phi^D = \mathbb{E}_{(s,a) \sim \mu^D}[\phi(s,a)]$, and we have

$$(s,a) \sim \mu^D, r \sim R(\cdot|s,a), s' \sim P(\cdot|s,a), k = \phi(s,a), k' = \psi(s')$$
$$\iff k \sim \phi^D, r \sim R_\phi(\cdot|k), k' \sim P_\phi(\cdot|k).$$

Similar to before, we can also show that the value function in the MRP is equivalent to $Q^\pi$, and LSTDQ is equivalent to the model-based solution, and so on.

**Approach 2.** An alternative approach is to define the "abstract MDP" $M_\phi$ over the *original state space* $\mathcal{S}$ (Jiang, 2018; Xie & Jiang, 2020a; Zhang & Jiang, 2021):

$$R_\phi(s,a) = \frac{\sum_{(\tilde{s},\tilde{a}):\phi(\tilde{s},\tilde{a})=\phi(s,a)} \mu^D(\tilde{s},\tilde{a})R(s,a)}{\sum_{(\tilde{s},\tilde{a}):\phi(\tilde{s},\tilde{a})=\phi(s,a))} \mu^D(\tilde{s},\tilde{a})},$$
$$P_\phi(\cdot|s,a) = \frac{\sum_{(\tilde{s},\tilde{a}):\phi(\tilde{s},\tilde{a})=\phi(s,a)} \mu^D(\tilde{s},\tilde{a})P(\cdot|s,a)}{\sum_{(\tilde{s},\tilde{a}):\phi(\tilde{s},\tilde{a})=\phi(s,a))} \mu^D(\tilde{s},\tilde{a})}.$$

Unlike Approach 1 which requires us to extend some MDP results to MRPs, this version still defines $M_\phi$ as an MDP, albeit over the original state space $\mathcal{S}$, and the initial state distribution is still $\mu_0$. It has piecewise-constant (w.r.t. the partition induced by $\phi$) reward and transition functions, and within each partition it straightforwardly (and seemingly naïvely) mixes the true reward and transition according to the relative weighting in the data distribution. Similar to before, we have $Q^\pi = Q_{M_\phi}^\pi$.

To connect LSTDQ to this abstract model, we define the empirical projected Bellman operator $\widehat{\mathcal{T}}_\phi^{\mu^D} : \mathbb{R}^{\mathcal{S} \times \mathcal{A}} \to \mathbb{R}^{\mathcal{S} \times \mathcal{A}}$ in the same fashion as Definition 2 of Xie & Jiang (2020a), and show that (1) $\widehat{Q}_{\text{lstd}}$ is the fixed point of this operator, and (2) fixing any $f$, $\widehat{\mathcal{T}}_\phi^\pi f$ concentrates towards $\mathcal{T}_{M_\phi}^\pi f$ with a sample complexity independent of $|\mathcal{S}|$. By union bounding over all piecewise-constant (i.e., linear-in-$\phi$) $f$, we establish that

$$\mathbb{E}_{(s,a) \sim \mu^D}[(\widehat{Q}_{\text{lstd}} - \mathcal{T}_{M_\phi}^\pi \widehat{Q}_{\text{lstd}})(s,a)^2] \leq \varepsilon_{\text{stat}},$$

where $\varepsilon_{\text{stat}}$ is the statistical error term (see Xie & Jiang (2020a, Lemma 9)). Then, error propagation is analyzed by invoking simulation lemma in $M_\phi$:

$$J_{\widehat{Q}_{\text{lstd}}}(\pi) - J(\pi) = \mathbb{E}_{s \sim \mu_0, a \sim \pi}[\widehat{Q}_{\text{lstd}}(s,a)] - J_{M_\phi}(\pi)$$
$$= \frac{1}{1-\gamma} \mathbb{E}_{(s,a) \sim d_{M_\phi}^\pi}[\widehat{Q}_{\text{lstd}} - \mathcal{T}_{M_\phi}^\pi \widehat{Q}_{\text{lstd}}].$$

The last step is error translation from $\mu^D$ to $d_{M_\phi}^\pi$, where a naïve translation would yield the size of $d_{M_\phi}^\pi/\mu^D$ which is not the same as our $C_\phi^\pi$. However, note that $\widehat{Q}_{\text{lstd}}$ and $\mathcal{T}_{M_\phi}^\pi(\cdot)$ are both piecewise-constant (i.e., linear in $\phi$), allowing us to projecting each distribution onto the feature space $(d_{M_\phi}^\pi \to \mathbb{E}_{(s,a) \sim d_{M_\phi}^\pi}[\phi(s,a)], \mu^D \to \mathbb{E}_{(s,a) \sim \mu^D}[\phi(s,a)] = \phi^D)$ before calculating their density ratio, which recovers $C_\phi^\pi$.

## C.5 PROOF OF PROPOSITION 4

**Proposition 4.** *When $\phi(s,a) = \mathbf{e}_{\psi(s),a}$ is induced by a state abstraction $\psi$ and $\pi$ depends on $s$ only through $\psi(s)$, recall that $\phi^D = \mathbb{E}_{\mu^D}[\phi]$ is a distribution over $(k,a)$ pairs, and we have*

$$C_\phi^\pi = \mathbb{E}_{(k,a)\sim\phi^D}[(\mu_{M_\psi}^\pi/\phi^D)^2].$$

**Proof of Proposition 4.** We compute the $A$ matrix. Below, we define $P^\pi(s',a' \mid s,a) = P(s' \mid s,a)\pi(a' \mid s')$, $P(k' \mid s,a) = \sum_{s':\psi(s')=k} P(s' \mid s,a)$, and

$$P^\pi(k',a' \mid s,a) = P(k' \mid s,a)\pi(a' \mid k'),$$

which is valid since $\pi$ is consistent with the state abstraction. To start, the covariance matrix $\Sigma$ becomes

$$\Sigma = \mathbb{E}_{s,a\sim\mu^D}\left[\phi(s,a)\phi(s,a)^\top\right] = \sum_{k\in[K],a\in\mathcal{A}}\sum_{s\in\mathcal{S}:\psi(s)=k} \mu^D(s,a)\mathbf{e}_{k,a}\mathbf{e}_{k,a}^\top$$

$$= \sum_{k\in[K],a\in\mathcal{A}} \mathbf{e}_{k,a}\mathbf{e}_{k,a}^\top \left(\sum_{s\in\mathcal{S}:\psi(s)=k} \mu^D(s,a)\right)$$

$$= \sum_{k\in[K],a\in\mathcal{A}} \mathbf{e}_{k,a}\mathbf{e}_{k,a}^\top \phi^D(k,a) = \mathrm{diag}(\phi^D),$$

where we recalled the definition of $\phi^D(k,a) = \sum_{s\in\mathcal{S}:\psi(s)=k} \mu^D(s,a)$, and $\mathrm{diag}(\phi^D)$ is the diagonal matrix with elements of $\phi^D$ on the diagonal. Let's examine the cross-covariance $\Sigma_{\mathrm{cr}}$.

$$\Sigma_{\mathrm{cr}} = \mathbb{E}_{s,a\sim\mu^D}\left[\phi(s,a)\phi(s',a')^\top\right]$$

$$= \sum_{s\in\mathcal{S},a\in\mathcal{A}} \mu^D(s,a)\phi(s,a) \sum_{s'\in\mathcal{S},a'\in\mathcal{A}} P^\pi(s',a' \mid s,a)\phi(s',a')^\top$$

$$= \sum_{k\in[K],a\in\mathcal{A}} \mathbf{e}_{k,a} \sum_{s\in\mathcal{S}:\psi(s)=k} \mu^D(s,a)\left(\sum_{k'\in[K],a'\in\mathcal{A}}\sum_{s'\in\mathcal{S}:\psi(s')=k'} P^\pi(s',a' \mid s,a)\mathbf{e}_{k',a'}^\top\right)$$

$$= \sum_{k\in[K],a\in\mathcal{A}} \mathbf{e}_{k,a} \sum_{s\in\mathcal{S}:\psi(s)=k} \mu^D(s,a)\left(\sum_{k'\in[K],a'\in\mathcal{A}} \mathbf{e}_{k',a'}^\top P^\pi(k',a' \mid s,a)\right)$$

$$= \sum_{k\in[K],a\in\mathcal{A}} \mathbf{e}_{k,a} \sum_{k'\in[K],a'\in\mathcal{A}} \mathbf{e}_{k',a'}^\top \sum_{s\in\mathcal{S}:\psi(s)=k} \mu^D(s,a)P^\pi(k',a' \mid s,a)$$

$$= \sum_{k\in[K],a\in\mathcal{A}} \mathbf{e}_{k,a} \sum_{k'\in[K],a'\in\mathcal{A}} \mathbf{e}_{k',a'}^\top \phi^D(k,a)\left(\frac{\sum_{s\in\mathcal{S}:\psi(s)=k} \mu^D(s,a)P^\pi(k',a' \mid s,a)}{\phi^D(k,a)}\right)$$

$$:= \sum_{k\in[K],a\in\mathcal{A}} \mathbf{e}_{k,a} \sum_{k'\in[K],a'\in\mathcal{A}} \mathbf{e}_{k',a'}^\top \phi^D(k,a)P_\psi^\pi(k',a' \mid k,a)$$

$$= \Sigma\left(P_\psi^\pi\right)^\top,$$

where we recall $P_\psi^\pi$ as the transition kernel of policy $\pi$ in $M_\psi$:

$$P_\psi^\pi(k',a' \mid k,a) = \frac{\sum_{s:\psi(s)=k} \mu^D(s,a)P^\pi(k',a' \mid s,a)}{\phi^D(k,a)}.$$

Putting our expressions for $\Sigma$ and $\Sigma_{\mathrm{cr}}$ together, we immediately have

$$B^\pi = P_\psi^\pi.$$

Given that the initial state distribution in $M_\psi$ is $\psi_0$, we can compute its initial state-action distribution

$$\psi_0(k_0)\pi(a_0|k_0) = \sum_{s_0:\psi(s_0)=k_0} \mu_0(s_0)\pi(a_0|k_0) = \mathbb{E}_{s\sim\mu_0,a\sim\pi}[\mathbb{I}[\psi(s)=k_0, a=a_0]]$$

$$= \mathbb{E}_{s\sim\mu_0,a\sim\pi}[\mathbf{e}_{k_0,a_0}(\psi(s),a)] = \phi_0(k_0,a_0),$$

implying that the initial state-action distribution happens to be $\phi_0$. Finally, our coverage coefficient becomes

$$
\begin{aligned}
C_\phi^\pi &= (1-\gamma)^2 \phi_0^\top (1-\gamma B^\pi)^{-\top} \Sigma^{-1} (I - \gamma B^\pi)^{-1} \phi_0 \\
&= (1-\gamma)^2 \phi_0^\top (I - \gamma P_\psi^\pi)^{-\top} \mathrm{diag}(\phi^D)^{-1} (I - \gamma P_\psi^\pi)^{-1} \phi_0 \\
&= (\mu_{M_\psi}^\pi)^\top \mathrm{diag}(\phi^D)^{-1} \mu_{M_\psi}^\pi = \mathbb{E}_{(k,a)\sim\phi^D}\left[ (\mu_{M_\psi}^\pi / \phi^D)^2 \right].
\end{aligned}
$$

$\square$

### C.6 PROOF OF PROPOSITION 5

**Proposition 5.** *Let $\mathcal{F}_\phi := \{\phi^\top \theta : \theta \in \mathbb{R}^d\}$ be the space of functions linear in $\phi$. Assume $\mathcal{F}_\phi$ satisfies Bellman-completeness (Assumption 3). Then, (1) $B^\pi$ becomes an exact model for next-feature prediction, i.e., $\mathbb{E}_{s'\sim P(\cdot|s,a)}[\phi(s',\pi)] = B^\pi \phi(s,a)$, (2) $\mu_\phi^\pi = \phi^\pi$, (3) $\rho(B^\pi) \leq 1$, and (4)*

$$
C_\phi^\pi = C_{\mathrm{lin}}^\pi = (\phi^\pi)^\top \Sigma^{-1} \phi^\pi.
$$

**Proof of Proposition 5.**

**(1) $B^\pi$ is an exact next-feature predictor:** $\mathbb{E}_{s'\sim P(\cdot|s,a)}[\phi(s,a)] = B^\pi \phi(s,a)$ **for all** $(s,a)$. First, we show that under Bellman completeness $\mathcal{F}_\phi$ is also closed under the transition operator $\mathcal{P}^\pi := \mathcal{T}^\pi - R$, that is, $\mathcal{P}^\pi f \in \mathcal{F}_\phi$ for all $f \in \mathcal{F}_\phi$.

The linearity of $\mathcal{F}_\phi$ together with Bellman completeness immediately imply that the reward function is linear, or $R \in \mathcal{F}_\phi$. Define $f_0 \in \mathcal{F}_\phi$ to be the function corresponding to the parameter $\theta = \mathbf{0}_d$, so that $f_0(s,a) = 0$ for all $(s,a)$; we have $\mathcal{T}^\pi f_0 = R \in \mathcal{F}_\phi$.

Next, fix any $f \in \mathcal{F}_\phi$ and observe that $\mathcal{T}^\pi f$ is also linear, since $\mathcal{T}^\pi f \in \mathcal{F}_\phi$ under Bellman completeness. It follows that $\mathcal{T}^\pi f - R = \mathcal{P}^\pi f \in \mathcal{F}_\phi$ because the difference of two functions linear in the same features is also linear in those features, which proves that $\mathcal{F}_\phi$ is closed under $\mathcal{P}^\pi$. This closure implies that for any $f \in \mathcal{F}_\phi$, there exists some $\theta_f \in \mathbb{R}^d$ such that

$$
\phi(s,a)^\top \theta_f = (\mathcal{P}^\pi f)(s,a) = \mathbb{E}_{s'\sim P(\cdot|s,a)}[f(s',\pi)].
$$

To prove the stated claim we will utilize choice instantations of such functions and their corresponding parameters. For $i \in [d]$, define the function $f_i := \langle \phi, \mathbf{e}_i \rangle \in \mathcal{F}_\phi$, and let $\theta_i \in \mathbb{R}^d$ be such that

$$
\phi(s,a)^\top \theta_i = \mathbb{E}_{s'\sim P(\cdot|s,a)}[f_i(s',\pi)], \ \forall (s,a).
$$

Then for all $(s,a)$,

$$
\mathbb{E}_{s'\sim P(\cdot|s,a)}[\phi(s',\pi)] =
\begin{bmatrix}
\mathbb{E}_{s'\sim P(\cdot|s,a)}[f_1(s',\pi)] \\
\mathbb{E}_{s'\sim P(\cdot|s,a)}[f_2(s',\pi)] \\
\vdots \\
\mathbb{E}_{s'\sim P(\cdot|s,a)}[f_d(s',\pi)]
\end{bmatrix}
=
\underbrace{\begin{bmatrix}
- & \theta_1^\top & - \\
- & \theta_2^\top & - \\
 & \vdots & \\
- & \theta_d^\top & -
\end{bmatrix}}_{(*)} \phi(s,a).
$$

Lastly, we will show that the above system of equations is satisfied by setting

$$
(*) = \Sigma_{\mathrm{cr}}^{-\top} \Sigma^{-1} = B^\pi.
$$

Right-multiplying both sides by $\phi(s,a)^\top$ then taking the expectation over $(s,a) \sim \mu^D$, we obtain

$$
\mathbb{E}_{(s,a,s',a')\sim\mu^D\times P\times\pi}\left[\phi(s',a')\phi(s,a)^\top\right] = B^\pi \mathbb{E}_{(s,a)\sim\mu^D}\left[\phi(s,a)\phi(s,a)^\top\right].
$$

Solving for $B^\pi$ and rearranging gives

$$B^\pi = \left(\mathbb{E}_{(s,a,s',a')\sim\mu^D\times P\times\pi}[\phi(s,a)\phi(s',a')^\top]\right)^\top \Sigma^{-1}$$
$$= \Sigma_{\text{cr}}^\top \Sigma^{-1},$$

which confirms that $B^\pi = (\Sigma^{-1}\Sigma_{\text{cr}})^\top$ satisfies for all $(s,a)$ the equivalence

$$\mathbb{E}_{s'\sim P(\cdot|s,a)}[\phi(s',\pi)] = B^\pi\phi(s,a).$$

**(2) Showing $\mu_\phi^\pi = \phi^\pi$.** Recall that $\phi^\pi = \mathbb{E}_{(s,a)\sim\mu^\pi}[\phi(s,a)]$. Using the Bellman flow equations for $\mu^\pi$, we obtain a recursive system of equations for the dynamics of $\phi^\pi$:

$$\phi^\pi = \sum_{s,a} \phi(s,a)\mu^\pi(s,a)$$

$$= \sum_{s,a} \phi(s,a)\left((1-\gamma)\mu_0^\pi(s,a) + \gamma\sum_{s',a'} P^\pi(s,a\mid s',a')\mu^\pi(s',a')\right)$$

$$= (1-\gamma)\phi_0 + \gamma\,\mathbb{E}_{(s,a)\sim\mu^\pi}\left[\mathbb{E}_{s'\sim P(\cdot|s,a)}[\phi(s',\pi)]\right]$$

$$= (1-\gamma)\phi_0 + \gamma\,\mathbb{E}_{(s,a)\sim\mu^\pi}[B^\pi\phi(s,a)]$$

$$= (1-\gamma)\phi_0 + \gamma B^\pi\phi^\pi,$$

where we invoke the result from **(1)** in the second-to-last line. Repeatedly expanding the RHS of the equation with the recursion,

$$\phi^\pi = (1-\gamma)\phi_0 + \gamma B^\pi\phi^\pi,$$
$$= (1-\gamma)\phi_0 + \gamma B^\pi((1-\gamma)\phi_0 + \gamma B^\pi\phi^\pi)$$
$$= (1-\gamma)\left(\phi_0 + \gamma B^\pi\phi_0 + \gamma^2(B^\pi)^2\phi^\pi\right)$$
$$\cdots$$
$$= (1-\gamma)\sum_{t=0}^\infty \gamma^t(B^\pi)^t\phi_0,$$

which is exactly the definition of $\mu_\phi^\pi$ from Proposition 1.

**(3) Showing $\rho(B^\pi) \le 1$.** The proof of (2) implies that for any $(s_0,a_0) \in \mathcal{S}\times\mathcal{A}$, $(B^\pi)^t\phi(s_0,a_0) = \mathbb{E}_{(s,a)\sim\mu_t^{\pi,s_0,a_0}}[\phi(s,a)]$, where $\mu_t^{\pi,s_0,a_0}$ is the $t$-th step state-action distribution under $\pi$ when the initial state-action pair is the given $(s_0,a_0)$. Given $\|\phi(s,a)\|_2 \le B_\phi, \forall(s,a)$, we have

$$(B^\pi)^t\phi(s_0,a_0) \le B_\phi, \quad \forall t.$$

Given that $\Sigma$ is full-rank, we can always find $\{(s_0^{(i)}, a_0^{(i)})\}_{i=1}^d$ such that $\{u_i := \phi(s_0^{(i)}, a_0^{(i)})\}_{i=1}^d$ forms a basis of $\mathbb{R}^d$. Then we have $\|(B^\pi)^t u_i\| \le B_\phi, \forall t$.

Now we show that $\|(B^\pi)^t\|_{\text{op}}$, where $\|\cdot\|_{\text{op}}$ is the operator norm, also has a finite bound that is independent of $t$. Recall that operator norm is the largest singular value; let the corresponding singular vector be $u$ and $\|u\|_2 = 1$, and we express $u = \sum_{i=1}^d \alpha_i u_i$. We have

$$\|(B^\pi)^t\|_{\text{op}} = \|(B^\pi)^t u\|_2 = \left\|\sum_{i=1}^d \alpha_i(B^\pi)^t u_i\right\|_2 \le \sum_{i=1}^d |\alpha_i| B_\phi =: v.$$

The key here is that the upper bound $v < \infty$ is independent of $t$. Plugging into the Gelfand's formula, we have

$$\rho(B^\pi) = \lim_{t\to\infty}\|(B^\pi)^t\|_{\text{op}}^{1/t} \le \lim_{t\to\infty} v^{1/t} = 1.$$

**(4) Proving equivalence** $C_\phi^\pi = C_{\text{lin}}^\pi$. Recalling Eq. (14) and the definition of $\mu_\phi^\pi$, following the proof of Proposition 1, when $\sigma_{\max}(B^\pi) < 1/\gamma$ we may write

$$
\begin{aligned}
C_\phi^\pi &= (1-\gamma)^2 \phi_0^\top A^{-1} \Sigma A^{-\top} \phi_0 \\
&= (1-\gamma)^2 \phi_0^\top (I - \gamma B^\pi)^{-\top} \Sigma^{-1} (I - \gamma B^\pi)^{-1} \phi_0. \\
&= (\mu_\phi^\pi)^\top \Sigma^{-1} \mu_\phi^\pi.
\end{aligned}
$$

Substituting the previously derived identity that $\mu_\phi^\pi = \phi^\pi$ in the last line,

$$
(\mu_\phi^\pi)^\top \Sigma^{-1} \mu_\phi^\pi = (\phi^\pi)^\top \Sigma^{-1} \phi^\pi = C_{\text{lin}}^\pi.
$$

$\square$

## C.7 DETAILS ON MWL

Here we expand on Section 5.4 and provide more details about the unification with MWL.

**MWL guarantee.** MWL is concerned with the following loss:

$$
L(w, f) := \left| J_f(\pi) + \tfrac{1}{1-\gamma} \mathbb{E}_{\mu^D}[w(s,a) \cdot (\gamma f(s', \pi) - f(s,a))] \right| \tag{26}
$$

Let $\widehat{L}(w, f)$ be its empirical estimate from data. The algorithm learns

$$
\widehat{w} = \arg\min_{w \in \mathcal{W}} \max_{f \in \mathcal{F}} \widehat{L}(w, f),
$$

and produces $J_{\widehat{w}}(\pi) := \tfrac{1}{1-\gamma} \mathbb{E}_{\mu^D}[\widehat{w}(s,a) \cdot r]$ as an estimation of $J(\pi)$. Uehara et al. (2020) show that when $Q^\pi \in \mathcal{F}$, the estimation error can be bounded as

$$
|J_{\widehat{w}}(\pi) - J(\pi)| \leq \min_{w \in \mathcal{W}} \max_{f \in \mathcal{F}} L(w, f) + \max_{w \in \mathcal{W}, f \in \mathcal{F}} |L(w, f) - \widehat{L}(w, f)|.
$$

The first term is the realizability error of $\mathcal{W}$. The second term is uniform deviation bound for $\widehat{L}(w, f)$; for finite $\mathcal{W}, \mathcal{F}$, Hoeffding with union bound gives: w.p. $\geq 1 - \delta$,

$$
\max_{w \in \mathcal{W}, f \in \mathcal{F}} |L(w, f) - \widehat{L}(w, f)| \lesssim \frac{B_\mathcal{W} B_\mathcal{F}}{1 - \gamma} \sqrt{\frac{1}{n} \ln \frac{|\mathcal{W}||\mathcal{F}|}{\delta}},
$$

where $B_\mathcal{W} := \max_{w \in \mathcal{W}} \|w\|_\infty$, and $B_\mathcal{F}$ is defined similarly. Since $\mathcal{F}$ models $Q^\pi$, we often assume $B_\mathcal{F} \lesssim V_{\max}$.

**Coverage in MWL.** In this analysis, the coverage parameter is manifested rather implicitly in $C_\mathcal{W}$: we need to find $w^\star \in \mathcal{W}$ to control the first term, and for simplicity let's focus on $w^\star$ such that $\max_{f \in \mathcal{F}} L(w^\star, f) = 0$; then the estimation error will only have the uniform deviation term. As mentioned in Section 5.2, a standard choice is $w^\star(s,a) = \mu^\pi(s,a)/\mu^D(s,a)$. Then, when $w^\star \in \mathcal{W}$, we have

$$
C_\mathcal{W} \geq \|w^\star\|_\infty = \|\mu^\pi/\mu^D\|.,
$$

which is the standard $\ell_\infty$ norm of density ratio. If $\mathcal{W}$ is properly designed to realize $w^\star$, we may assume $C_\mathcal{W} \lesssim \|w^\star\|_\infty$, and obtain a bound that explicitly depends on $\|\mu^\pi/\mu^D\|_\infty$.

**Connection to LSTDQ and unification with our results.** When $\mathcal{W}$ and $\mathcal{F}$ are linear classes in the same give feature $\phi$ (with appropriately bounded norms on the parameters), Uehara et al. (2020, Appendix A.3) show that MWL is equivalent to LSTDQ when $\widehat{A}$ is invertible, in the sense that

$$
J_{\widehat{w}}(\pi) = J_{\widehat{Q}_{\text{lstd}}}(\pi).
$$

In terms of analysis, the linear structure of the classes provide another choice of $w^\star$ that satisfies $\max_{f \in \mathcal{F}} L(w, f) = 0$, namely $w^\star(s,a) = (1-\gamma) \phi_0^\top A^{-1} \phi(s,a)$, whose data second-moment is

equal to $C_\phi^\pi$. To make this quantity appear in the guarantee, we calculate the following variance: for fixed $w \in \mathcal{W}$, $f \in \mathcal{F}$,

$$\text{Var}_{\mu^D}[w(s,a)(\gamma f(s',\pi) - f(s,a))] \leq \mathbb{E}_{\mu^D}[w(s,a)^2(\gamma f(s',\pi) - f(s,a))^2]$$
$$\lesssim \mathbb{E}_{\mu^D}[w(s,a)^2 B_{\mathcal{F}}^2] \leq B_{\mathcal{W},2}^2 B_{\mathcal{F}}^2,$$

where $B_{\mathcal{W},2} := \max_{w \in \mathcal{W}} \mathbb{E}_{\mu^D}[w(s,a)^2] \geq C_\phi^\pi$. Similar to before we may assume $B_{\mathcal{W},2} \lesssim \mathbb{E}_{\mu^D}[w^\star(s,a)]^2 = C_\phi^\pi$. Now by Bernstein's inequality and a standard covering argument over linear $\mathcal{W}$ and $\mathcal{F}$, we have (c.f. Xie & Jiang (2020b, Theorem 8))

$$\max_{w \in \mathcal{W}, f \in \mathcal{F}} |L(w,f) - \widehat{L}(w,f)| \lesssim \frac{V_{\max}}{1-\gamma} \left( \sqrt{\frac{C_\phi^\pi \cdot (d + \log(1/\delta))}{n}} + \frac{B_{\mathcal{W}}(d + \log(1/\delta))}{n} \right),$$

whose $\sqrt{1/n}$ term matches our Theorem 1 except that we do not depend on $d$.

## D  FUNCTION ESTIMATION GUARANTEES

For most of the paper we have focused on providing return estimation guarantees, i.e., error bounds for estimating $J(\pi)$. In some scenarios, it is desirable to obtain stronger *function estimation* guarantees (Huang & Jiang, 2022; Perdomo et al., 2023), that $\widehat{Q}_{\text{lstd}}$ and $Q^\pi$ are close as functions, typically measured by weighted 2-norm. Indeed, our proof of Theorem 2 can be easily adapted to provide the following guarantee:

**Theorem 3** (Function Estimation). *Under the same assumptions as Theorem 2, w.p. $\geq 1 - \delta$, for any $\nu \in \Delta(\mathcal{S} \times \mathcal{A})$,*

$$\sqrt{\mathbb{E}_{(s,a) \sim \nu}[(Q^\pi(s,a) - \widehat{Q}_{\text{lstd}}(s,a))^2]} \lesssim \frac{V_{\max}}{1-\gamma} \sqrt{\frac{\widehat{C}_{\text{fn}}^\pi \cdot d \log(1/\delta)}{n}},$$

*where $\widehat{C}_{\text{fn}}^\pi := (1-\gamma)^2 \mathbb{E}_{(s_0,a_0) \sim \nu} \left[ \|\widehat{\Sigma}^{1/2} \widehat{A}^{-\top} \phi(s_0,a_0)\|_2^2 \right]$.*

When $\nu = \mu_0 \circ \pi$ is a point-mass, the LHS of Theorem 3 coincides with that of Theorem 2, and the guarantees on the RHS are identical, too. Also recall that the naïve analysis based on $1/\sigma_{\min}(A)$ (Section 1) provides parameter identification (i.e., bounded $\|\widehat{\theta}_{\text{lstd}} - \theta^\star\|$), which immediately provides $\ell_\infty$ function-estimation guarantee. This result is directly implied by our Theorem 3, where the coverage parameter can be bounded as a function of $\sigma_{\min}(A)$ and $B_\phi$.

**Remark on $C_{\text{fn}}^\pi$.**  Similar to how we can replace $\widehat{C}_\phi^\pi$ with $C_\phi^\pi$ up to lower-order terms (see Theorem 2 and its proof), we can also obtain a variant of Theorem 3 that depends on the population version of $\widehat{C}_{\text{fn}}^\pi$, which we denote as $C_{\text{fn}}^\pi$. It is interesting to compare it to standard coverage parameters that enable function-estimation guarantees under completeness (Section 3). Note that the term inside $C_{\text{fn}}^\pi = \mathbb{E}_{(s_0,a_0) \sim \nu}[\cdot]$ is simply $C_\phi^\pi$ but for a deterministic initial state-action pair $(s_0, a_0)$. Applying Proposition 5, we have

$$C_{\text{fn}}^\pi = \mathbb{E}_{(s_0,a_0) \sim \nu} \left[ (\phi_{s_0,a_0}^\pi)^\top \Sigma^{-1} \phi_{s_0,a_0}^\pi \right],$$

where $\phi_{s_0,a_0}^\pi = \mathbb{E}_{(s,a) \sim \mu_{s_0,a_0}^\pi}[\phi(s,a)]$ is the expected feature under the occupancy induced from deterministic $s_0, a_0$ as the initial state-action pair. In comparison, the standard coverage in the literature is

$$C_{\text{lin,fn}}^\pi = \mathbb{E}_{(s,a) \sim \mu^\pi}[\phi(s,a)^\top \Sigma^{-1} \phi(s,a)].$$

As can be seen, our $C_{\text{fn}}^\pi$ is in between $C_\phi^\pi$ and $C_{\text{lin,fn}}^\pi$, since we partially marginalize out the portion of $\mu^\pi$ that can be attribute to each initial state-action pair, instead of measuring every single $(s,a) \sim \mu^\pi$ under $\Sigma^{-1}$ in a completely point-wise manner.

**Proof of Theorem 3.** We repeat a similar derivation to the proof of Theorem 1, noting that the proof holds when the initial state-action distribution $s_0 \sim \mu_0, a_0 \sim \pi$ changes to an arbitrary distribution $\nu$.

$$\mathbb{E}_{(s_0,a_0)\sim\nu}\left[\left(Q^\pi(s_0,a_0) - \widehat{Q}_{\mathrm{lstd}}(s_0,a_0)\right)^2\right]$$

$$= \mathbb{E}_{(s_0,a_0)\sim\nu}\left[\left(\phi(s_0,a_0)^\top\left(\theta^\star - \widehat{\theta}_{\mathrm{lstd}}\right)\right)^2\right]$$

$$= \mathbb{E}_{(s_0,a_0)\sim\nu}\left[\left(\phi(s_0,a_0)^\top\widehat{A}^{-1}\widehat{\Sigma}^{1/2}\widehat{\Sigma}^{-1/2}\widehat{A}\left(\theta^\star - \widehat{\theta}_{\mathrm{lstd}}\right)\right)^2\right]$$

$$\le \mathbb{E}_{(s_0,a_0)\sim\nu}\left[\left\|\widehat{\Sigma}^{1/2}\widehat{A}^{-\top}\phi(s_0,a_0)\right\|_2^2\left\|\widehat{\Sigma}^{-1/2}(\widehat{A}\theta^\star - \widehat{b})\right\|_2^2\right].$$

To conclude, we recall the concentration bound from Lemma 1, that w.p. $\ge 1 - \delta$,

$$\|\widehat{\Sigma}^{-1/2}(\widehat{A}\theta^\star - \widehat{b})\|_2^2 \lesssim V_{\max}^2\frac{d + \log(1/\delta)}{n}.$$

Plugging this in yields the proof. $\qquad\square$

# E $\quad \sigma_{\min}(A)$-INDEPENDENT POPULATION BOUND VIA LOSS MINIMIZATION ALGORITHM

Here we provide an alternative analysis to Theorem 1, where we are able to eliminate the dependence on $1/\sigma_{\min}(A)$, but the rates still depend on quantities necessary for the analysis of linear regression under random design (in our case, $1/\lambda_{\min}(\Sigma)$, though we expect that this can be further tightened to leverage-score-type conditions Hsu et al. (2011); Perdomo et al. (2023)). The analysis also requires a slight change of the LSTDQ algorithm to a loss-minimization form (Liu et al., 2025):

$$\widehat{\theta}_{\mathrm{lstd}} = \arg\min_{\theta\in\Theta}\|\widehat{\Sigma}^{-1/2}(\widehat{A}\theta - \widehat{b})\|_2. \tag{27}$$

In practice, when $\widehat{A}$ is near-singular, the inverse solution $\widehat{A}^{-1}\widehat{b}$ may have a very large norm which is clearly problematic, demanding some regularization to control the norm of the solution. The loss-minimization formulation of Eq. (27) is a natural abstraction of this process, where we search for $\widehat{\theta}$ in a pre-defined parameter space with bounded norm. If $\widehat{A}^{-1}\widehat{b} \in \Theta$, it is easy to see that the loss-minimization solution coincides with the inverse solution; when $\widehat{A}^{-1}\widehat{b} \notin \Theta$, Eq. (27) still outputs a bounded solution to ensure generalization and good statistical properties.

We will need the following boundedness assumption on $\Theta$.

**Assumption 4** (Boundedness of $\Theta$). *Assume* $\|\theta\|_2 \le B_\Theta, \forall\theta \in \Theta$.

**Additional linear algebraic notation.** For symmetric $\Sigma$, let $\kappa(\Sigma) = \frac{\lambda_{\max}(\Sigma)}{\lambda_{\min}(\Sigma)}$ be the condition number, where $\lambda_{\max}(\cdot)$ is the largest eigenvalue.

**Theorem 4.** *Assume that* $n \gtrsim \log(d/\delta)\kappa(\Sigma)B_\phi^2/\lambda_{\min}(\Sigma)$. *Under Assumptions 1, 2 and 4, the estimator in Eq. (27) satisfies that*

$$\left|J(\pi) - J_{\widehat{Q}_{\mathrm{lstd}}}(\pi)\right| \lesssim \frac{\sqrt{C_\phi^\pi}}{1-\gamma}\max\{B_\phi B_\Theta, R_{\max}\}^2\sqrt{\frac{d\log(B_\Theta n\delta^{-1})}{\lambda_{\min}(\Sigma)n}}$$

*with probability at least* $1 - \delta$.

**Proof of Theorem 4.** Let $\hat{\ell}(\theta)$ and $\ell(\theta)$ denote the empirical and population vectors:

$$\hat{\ell}(\theta) = \hat{A}\theta - \hat{b} \quad \text{and} \quad \ell(\theta) = A\theta - b.$$

Recall that $A\theta^\star = b$ and thus $\ell(\theta^\star) = 0$. This also implies $\|\Sigma^{-1/2}\ell(\theta^\star)\|_2 = 0$ by invertibility of $\Sigma$. We establish in the sequel the following concentration lemma.

**Lemma 5.** *With probability at least* $1 - \delta$, *we have that for all* $\theta \in \Theta$:

$$\left| \|\Sigma^{-1/2} \ell(\theta)\|_2 - \|\Sigma^{-1/2} \hat{\ell}(\theta)\|_2 \right| \leq \max\{B_\phi B_\Theta, R_{\max}\}^2 \sqrt{\frac{288 d \log(864 B_\Theta n \delta^{-1})}{\lambda_{\min}(\Sigma) n}} := \varepsilon_{\text{stat}}.$$

We also note the following simple technical lemma.

**Lemma 6.** *Let* $M = \Sigma^{-1/2} \widehat{\Sigma} \Sigma^{-1/2}$. *Then, for all* $v \in \mathbb{R}^d$, *we have*

$$v^\top \Sigma^{-1} v \leq \lambda_{\max}(M) v^\top \widehat{\Sigma}^{-1} v, \quad \text{and} \quad v^\top \widehat{\Sigma}^{-1} v \leq \frac{1}{\lambda_{\min}(M)} v^\top \Sigma^{-1} v.$$

Recall that $\widehat{\theta}$ satisfies $\arg\min_{\theta \in \Theta} \|\widehat{\Sigma}^{-1/2} \hat{\ell}(\theta)\|_2$. We now show that Lemma 5 and Lemma 6 imply that $\|\Sigma^{-1/2} \ell(\widehat{\theta})\|_2$ is small. This follows since:

$$\begin{aligned}
\|\Sigma^{-1/2} \ell(\widehat{\theta})\|_2 &\leq \|\Sigma^{-1/2} \hat{\ell}(\widehat{\theta})\|_2 + \varepsilon_{\text{stat}} \\
&\leq \sqrt{\lambda_{\max}(M)} \|\widehat{\Sigma}^{-1/2} \hat{\ell}(\widehat{\theta})\|_2 + \varepsilon_{\text{stat}} \\
&\leq \sqrt{\lambda_{\max}(M)} \|\widehat{\Sigma}^{-1/2} \hat{\ell}(\theta^\star)\|_2 + \varepsilon_{\text{stat}} \\
&\leq \sqrt{\frac{\lambda_{\max}(M)}{\lambda_{\min}(M)}} \|\Sigma^{-1/2} \hat{\ell}(\theta^\star)\|_2 + \varepsilon_{\text{stat}} \\
&\leq \sqrt{\kappa(M)} \|\Sigma^{-1/2} \ell(\theta^\star)\|_2 + \left(1 + \sqrt{\kappa(M)}\right) \varepsilon_{\text{stat}} \\
&= \left(1 + \sqrt{\kappa(M)}\right) \varepsilon_{\text{stat}} \\
&\leq 2\sqrt{\kappa(M)} \varepsilon_{\text{stat}}.
\end{aligned}$$

In the sequel, we also show concentration for the condition number of $M$ to a constant.

**Lemma 7.** *Let* $n \geq 16 \frac{\kappa(\Sigma)}{\lambda_{\min}(\Sigma)} \left(B_\phi^2 + \lambda_{\max}(\Sigma)\right) \log(6d/\delta)$. *Then, with probability at least* $1 - \delta$, *we have:*

$$\kappa(M) \leq 3$$

This implies that, under the condition on sample size, we have $\left\|\Sigma^{-1/2} \ell(\widehat{\theta})\right\|_2 \leq \sqrt{12} \varepsilon_{\text{stat}}$ with high-probability. We can now conclude the proof. Under the conditions and events stated above, we have:

$$\begin{aligned}
\left| \mathbb{E}_{s_0 \sim \mu_0, a_0 \sim \pi} \left[ Q^\pi(s_0, a_0) - \hat{Q}_{\text{lstd}}(s_0, a_0) \right] \right| &= \left| \mathbb{E}_{s_0 \sim \mu_0, a_0 \sim \pi} \left[ \phi(s_0, a_0)^\top \left( \theta^\star - \widehat{\theta}_{\text{lstd}} \right) \right] \right| \quad (28) \\
&= \left| \phi_0^\top \left( \theta^\star - \widehat{\theta}_{\text{lstd}} \right) \right| \\
&= \left| \phi_0^\top \left( A^{-1} b - \widehat{\theta}_{\text{lstd}} \right) \right| \\
&= \left| \phi_0^\top A^{-1} \left( b - A\widehat{\theta}_{\text{lstd}} \right) \right| \\
&= \left| \phi_0^\top A^{-1} \Sigma^{1/2} \Sigma^{-1/2} \left( b - A\widehat{\theta}_{\text{lstd}} \right) \right| \\
&= \left| \phi_0^\top A^{-1} \Sigma^{1/2} \Sigma^{-1/2} \left( b - A\widehat{\theta}_{\text{lstd}} \right) \right| \\
&\leq \left\| \Sigma^{1/2} A^{-T} \phi_0 \right\|_2 \left\| \Sigma^{-1/2} \left( A\widehat{\theta}_{\text{lstd}} - b \right) \right\|_2 \\
&= \left\| \Sigma^{1/2} A^{-T} \phi_0 \right\|_2 \left\| \Sigma^{-1/2} \left( A\widehat{\theta}_{\text{lstd}} - b \right) \right\|_2 \\
&= \sqrt{\phi_0^\top A^{-1} \Sigma A^{-T} \phi_0} \, \|\Sigma^{-1/2} \ell(\widehat{\theta})\|_2 \\
&\leq \frac{1}{1 - \gamma} \sqrt{C_\phi^\pi} \sqrt{12} \varepsilon_{\text{stat}}, \quad (29)
\end{aligned}$$

as desired. We now establish Lemmas 5 to 7.

$\square$

**Proof of Lemma 5.** Let $\theta$ be fixed for now, and $\Delta(\theta) = \hat{\ell}(\theta) - \ell(\theta)$. Note that by the reverse triangle inequality,

$$\left| \|\Sigma^{-1/2}\ell(\theta)\|_2 - \|\Sigma^{-1/2}\hat{\ell}(\theta)\|_2 \right| \leq \left\| \Sigma^{-1/2}\Delta(\theta) \right\|_2 = \left\| \Sigma^{-1/2}(\hat{A} - A)\theta - \Sigma^{-1/2}(\hat{b} - b) \right\|_2.$$

We use Vector Bernstein (Lemma 9) to show that this is small. Let $X_i = \phi(s_i, a_i)$, $Y_i = \phi(s_i, a_i) - \gamma\phi(s'_i, a'_i)$, and $\Delta_i(\theta) = X_i(Y_i^\top \theta - r_i) - (A\theta - b)$ denote the centered vectors. Note that

$$\|\Sigma^{-1/2}\Delta_i(\theta)\|_2 \leq \frac{1}{\sqrt{\lambda_{\min}(\Sigma)}} 2\max\{\|X_i(Y_i^\top \theta - r_i)\|_2, \|A\theta - b\|_2\}.$$

We have the following bound:

$$\begin{aligned}
\|A\theta - b\|_2 &= \left\| \mathbb{E}\Big[ \phi(s,a)\Big( (\phi(s,a) - \gamma\phi(s',a'))^\top \theta - r(s,a) \Big) \Big] \right\|_2 \\
&\leq \left\| \mathbb{E}\big[ \phi(s,a)\phi(s,a)^\top \theta \big] \right\|_2 + \gamma \left\| \mathbb{E}\big[ \phi(s,a)\phi(s',a')^\top \theta \big] \right\|_2 + \left\| \mathbb{E}[\phi(s,a)r(s,a)] \right\|_2 \\
&\leq (1 + \gamma)\max_{s,a}\|\phi(s,a)\|_2^2 \|\theta\|_2 + \max_{s,a}\|\phi(s,a)\|_2 R_{\max} \\
&\leq 3B_\phi \max\{B_\phi B_\Theta, R_{\max}\}. \tag{30}
\end{aligned}$$

We remark that with a similar derivation, this bound applies just as well to $\|X_i(Y_i^\top\theta - r_i)\|_2$, so in fact we have

$$\|\Sigma^{-1/2}\Delta_i(\theta)\|_2 \leq \frac{6}{\sqrt{\lambda_{\min}(\Sigma)}} B_\phi \max\{B_\phi B_\Theta, R_{\max}\}.$$

For the variance bound, we simply use that

$$\mathbb{E}\Big[ \|\Sigma^{-1/2}\Delta_i(\theta)\|_2^2 \Big] \leq \left( \frac{6}{\sqrt{\lambda_{\min}(\Sigma)}} B_\phi \max\{B_\phi B_\Theta, R_{\max}\} \right)^2.$$

Then, we conclude via Lemma 9 that

$$\|\Sigma^{-1/2}\Delta(\theta)\|_2 \leq B_\phi \max\{B_\phi B_\Theta, R_{\max}\} \sqrt{\frac{32\log(288\delta^{-1})}{\lambda_{\min}(\Sigma)n}}.$$

We now apply a covering argument over $\theta \in \Theta$. Let $\Theta_0 \subseteq \Theta$ be an $L_2$-covering of $\Theta$ of size $\mathcal{N}(\varepsilon)$, satisfying for for each $\theta \in \Theta$ there exists a covering member $\rho(\theta) \in \Theta_0$ satisfying $\|\theta - \rho(\theta)\|_2 \leq \varepsilon$. Via a simple triangle inequality:

$$\left\| \Sigma^{-1/2}\Delta(\theta) \right\|_2 \leq \left\| \Sigma^{-1/2}\Delta(\rho(\theta)) \right\|_2 + \left\| \Sigma^{-1/2}(\Delta(\theta) - \Delta(\rho(\theta))) \right\|_2.$$

We bound the latter term as a function of $\varepsilon$.

$$\begin{aligned}
\left\| \Sigma^{-1/2}(\Delta(\theta) - \Delta(\rho(\theta))) \right\|_2 &= \left\| \Sigma^{-1/2}\Big( A - \hat{A} \Big)(\theta - \rho(\theta)) \right\|_2 \\
&\leq \frac{1}{\sqrt{\lambda_{\min}(\Sigma)}} 2\max\Big\{ \sigma_{\max}(A), \sigma_{\max}(\hat{A}) \Big\}\varepsilon.
\end{aligned}$$

We notice that $\max\Big\{ \sigma_{\max}(A), \sigma_{\max}(\hat{A}) \Big\} \leq 2B_\phi^2$ via a similar reasoning to Eq. (30). This leaves us with:

$$\begin{aligned}
\left\| \Sigma^{-1/2}\Delta(\theta) \right\|_2 &\leq B_\phi \max\{B_\phi B_\Theta, R_{\max}\} \sqrt{\frac{32\log(288|\Theta_0|\delta^{-1})}{\lambda_{\min}(\Sigma)n}} + \frac{2B_\phi^2}{\sqrt{\lambda_{\min}(\Sigma)}}\varepsilon, \\
&\leq B_\phi \max\{B_\phi B_\Theta, R_{\max}\} \sqrt{\frac{32d\log(864\delta^{-1}/\varepsilon)}{\lambda_{\min}(\Sigma)n}} + \frac{2B_\phi^2}{\sqrt{\lambda_{\min}(\Sigma)}}\varepsilon,
\end{aligned}$$

where we have applied a union bound over the set $\Theta_0$, which is of size at most $(3B_\Theta/\varepsilon)^d$ for $\varepsilon \in (0, 1]$ by standard covering number bounds (Vershynin, 2018), since $\Theta \subset \{\theta \in \mathbb{R}^d : \|\theta\|_2 \le B_\Theta\}$. Picking $\varepsilon = 1/\sqrt{n}$ lets us conclude that, with probability at least $1 - \delta$, for all $\theta \in \Theta$,

$$\left\|\Sigma^{-1/2}\Delta(\theta)\right\|_2 \le B_\phi \max\{B_\phi B_\Theta, R_{\max}\}\sqrt{\frac{288d \log(864B_\Theta n\delta^{-1})}{\lambda_{\min}(\Sigma)n}},$$

as desired. $\qquad\square$

**Proof of Lemma 6.** Define $w = \Sigma^{-1/2}v$, so that $v = \Sigma^{1/2}w$. Then:

$$v^\top \widehat{\Sigma}^{-1} v = w^\top \Sigma^{1/2}\widehat{\Sigma}^{-1}\Sigma^{1/2}w = w^\top M^{-1}w,$$

so we have

$$\frac{1}{\lambda_{\max}(M)}v^\top\Sigma^{-1}v = \frac{1}{\lambda_{\max}(M)}w^\top w \le w^\top M^{-1}w \le \frac{1}{\lambda_{\min}(M)}w^\top w = \frac{1}{\lambda_{\min}(M)}v^\top\Sigma^{-1}v,$$

which establishes both bounds. $\qquad\square$

**Proof of Lemma 7.** We firstly establish that

$$\|\widehat{\Sigma} - \Sigma\|_2 \le \sqrt{\frac{8\lambda_{\max}(\Sigma)(B_\phi^2 + \lambda_{\max}(\Sigma))\log(6d/\delta)}{n}} =: \varepsilon_{\mathrm{op}}. \tag{31}$$

To do this, we use Matrix Bernstein (Lemma 8). Abbreviate $X_i := \phi(s_i, a_i)$, and let $Z_i = X_i X_i^\top - \Sigma$ be the centered matrices. Note that $\|Z_i\|_2 \le \|X_i X_i^\top\|_2 + \|\Sigma\|_2 \le \|X_i\|_2^2 + \lambda_{\max}(\Sigma) \le B_\phi^2 + \lambda_{\max}(\Sigma)$. For the variance term, we have:

$$\begin{aligned}\left\|\mathbb{E}\big[(X_i X_i^\top - \Sigma)^2\big]\right\|_2 &= \left\|\mathbb{E}\big[(X_i X_i^\top)^2\big] - \Sigma^2\right\|_2 \\ &\le \left\|B_\phi^2\, \mathbb{E}\big[X_i X_i^\top\big] - \Sigma^2\right\|_2 \\ &\le B_\phi^2 \lambda_{\max}(\Sigma) + \lambda_{\max}(\Sigma)^2.\end{aligned}$$

This yields

$$\|\hat{\Sigma} - \Sigma\|_2 \le \sqrt{\frac{2\lambda_{\max}(\Sigma)(B_\phi^2 + \lambda_{\max}(\Sigma))\log(2d/\delta)}{n}} + \frac{2(B_\phi^2 + \lambda_{\max}(\Sigma))\log(2d/\delta)}{3n}$$

The slow term dominates when $n$ is large enough:

$$n \ge \frac{2(B_\phi^2 + \lambda_{\max}(\Sigma))\log(2d/\delta)}{9\lambda_{\max}(\Sigma)} = \frac{2}{9}\log(2d/\delta)\left(\frac{B_\phi^2}{\lambda_{\max}(\Sigma)} + 1\right). \tag{32}$$

Note that this is implied by our assumption on $n$, since $\lambda_{\max}(\Sigma) \ge \lambda_{\min}(\Sigma)$ and $\kappa(\Sigma) \ge 1$. Thus, under this condition we have

$$\|\hat{\Sigma} - \Sigma\|_2 \le 2\sqrt{\frac{2\lambda_{\max}(\Sigma)(B_\phi^2 + \lambda_{\max}(\Sigma))\log(2d/\delta)}{n}} = \varepsilon_{\mathrm{op}},$$

as desired. Now, note that this implies

$$\left\|\Sigma^{-1/2}(\widehat{\Sigma} - \Sigma)\Sigma^{-1/2}\right\|_2 = \|M - I\|_{\mathrm{op}} \le \frac{1}{\lambda_{\min}(\Sigma)}\epsilon_{\mathrm{op}} := \varepsilon_M.$$

Note that our assumption on $n$ implies that

$$n \ge 16\frac{\kappa(\Sigma)}{\lambda_{\min}(\Sigma)}(B_\phi^2 + \lambda_{\max}(\Sigma))\log(2d/\delta)$$

$$\implies \varepsilon_M = \frac{2}{\lambda_{\min}(\Sigma)}\sqrt{2\lambda_{\max}(\Sigma)(B_\phi^2 + \lambda_{\max}(\Sigma))\log(2d/\delta)} \le \frac{1}{\sqrt{2}}$$

In particular, this implies

$$(1 - \varepsilon_M)I \preceq M \preceq (1 + \varepsilon_M)I,$$

so that $\lambda_{\min}(M) \geq 1 - \varepsilon_M$ and $\lambda_{\max}(M) \leq 1 + \varepsilon_M$, which implies

$$\kappa(M) \leq \frac{1 + \varepsilon_M}{1 - \varepsilon_M},$$

and thus

$$1 + \sqrt{\kappa(M)} \leq 1 + \sqrt{\frac{1 + \varepsilon_M}{1 - \varepsilon_M}} \leq 1 + (1 + 2\varepsilon_M) = 2 + 2\varepsilon_M \leq 3,$$

again using that $\varepsilon_M < 1/\sqrt{2}$ and the inequality $\sqrt{1+x/1-x} \leq 1 + 2x$, which is easily seen to be true by algebraic manipulations for $x < 1/\sqrt{2}$.

$\square$

## F    TECHNICAL TOOLS

**Lemma 8** (Matrix Bernstein, Tropp (2012)). *Let $S_1, \ldots, S_n \in \mathbb{R}^{d_1 \times d_2}$ be random, independent matrices satisfying $\mathbb{E}[S_k] = 0$, $\max\{\|\mathbb{E}[S_k S_k^\top]\|_{\mathrm{op}}, \|\mathbb{E}[S_k^\top S_k]_{\mathrm{op}}\|\} \leq \sigma^2$, and $\|S_k\|_{\mathrm{op}} \leq L$ almost surely for all $k$. Then, with probability at least $1 - \delta$ for any $\delta \in (0, 1)$,*

$$\left\| \frac{1}{n} \sum_{k=1}^n S_k \right\|_{\mathrm{op}} \leq \sqrt{\frac{2\sigma^2 \log((d_1 + d_2)/\delta)}{n}} + \frac{2L \log((d_1 + d_2)/\delta)}{3n}$$

**Lemma 9** (Vector Bernstein, Minsker (2017)). *Let $v_1, \ldots, v_n$ be independent vectors in $\mathbb{R}^d$ such that $\mathbb{E}[v_k] = 0$, $\mathbb{E}[\|v_k\|_2^2] \leq \sigma^2$, and $\|v_k\|_2 \leq L$ almost surely for all $k$. Then, with probability at least $1 - \delta$ for any $\delta \in (0, 1)$,*

$$\left\| \frac{1}{n} \sum_{i=1}^n v_i \right\|_2 \leq \sqrt{\frac{2\sigma^2 \log(28/\delta)}{n}} + \frac{2L \log(28/\delta)}{3n}.$$

**Lemma 10** (Vector Martingale Bernstein (Pinelis, 1994; Martinez-Taboada & Ramdas, 2024)). *Let $(X_t)_{t \leq T}$ be a martingale sequence of vectors in $\mathbb{R}^d$ adapted to a filtration $(\mathcal{F}_t)_{t \leq T}$, such that $\mathbb{E}_{t-1}[X_t] = 0$, and $\|X_t\|_2 \leq B$, and $\sum_{t=1}^T \mathbb{E}_{t-1}\left[\|X_t\|^2\right] \leq \sigma^2$. Then, with probability at least $1 - \delta$, we have:*

$$\left\| \sum_{t=1}^T X_t \right\|_2 \leq \sqrt{2\sigma^2 \log(2/\delta)} + \frac{2}{3} B \log(2/\delta).$$

