# A UNIFYING VIEW OF COVERAGE IN LINEAR OFF-POLICY EVALUATION

## ABSTRACT

Off-policy evaluation (OPE) is a fundamental task in reinforcement learning (RL). In the classic setting of *linear OPE*, finite-sample guarantees often take the form

$$\text{Evaluation error} \leq \text{poly}(C^\pi, d, 1/n, \log(1/\delta)),$$

where $d$ is the dimension of the features and $C^\pi$ is a ***feature coverage parameter*** that characterizes the degree to which the visited features lie in the span of the data distribution. Though such guarantees are well-understood for several popular algorithms under the Bellman-completeness assumption, this form of guarantee has not yet been achieved in the minimal setting where it is only assumed that

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

 $\widehat{A} = \widehat{\Sigma}$, and therefore $\widehat{C}_\phi^\pi = \phi_0^\top \widehat{\Sigma}^{-1} \phi_0 = \|\phi_0\|_{\widehat{\Sigma}^{-1}}^2$. Since Eq. (9) is the well-established bound for linear regression, this demonstrates the tightness of our bound on its dependence on $\widehat{C}_\phi^\pi$, $d$, and $n$.

That said, a caveat of Eq. (10) is that $\widehat{C}_\phi^\pi$ is a random variable. While such a situation is fairly common in offline RL theory (Jin et al., 2021), ideally we would like to have its population version $C_\phi^\pi$ (Eq. (13)), which does not depend on data randomness and is more interpretable; our interpretation in Section 5 will also focus on $C_\phi^\pi$. This is exactly what we offer in Corollary 1, where the asymptotically dominating term $O(1/\sqrt{n})$ is identical to Eq. (10) with $\widehat{C}_\phi^\pi$ replaced by $C_\phi^\pi$, but the spectral properties of $\Sigma$ and $A$ may enter the burn-in term $(n_0)$ and the fast-rate term $(o(\sqrt{1/n}))$. The fact that Eq. (12) is less clean and requires additional assumptions than Eq. (10) is somewhat expected, as this is also the case for linear regression, where a bound that depends on $\|\phi_0\|_{\Sigma^{-1}}$ (instead of $\|\phi_0\|_{\widehat{\Sigma}^{-1}}$) also requires additional assumptions (Hsu et al., 2011; Oliveira, 2016; Mourtada, 2022). In Appendix D, we provide another result that eliminates the dependence on $1/\sigma_{\min}(A)$ in the $C_\phi^\pi$ bound, in exchange for a $1/\lambda_{\min}(\Sigma)$ dependence, which we expect can be further sharpened to leverage-score-type conditions (Hsu et al., 2011; Perdomo et al., 2023). Finally, as we will see in Section 5.3, our bound is also tight when comparing to existing OPE guarantees analyzed under Bellman completeness.

## 5 UNDERSTANDING THE COVERAGE PARAMETER

In this section we provide interpretations of $C_\phi^\pi$ as a coverage parameter and discuss how it addresses the issues mentioned in Section 1. First, it is clear that $C_\phi^\pi$ is invariant to feature rescaling, thanks to the introduction of $\phi_0$. That said, the expression $C_\phi^\pi = (1-\gamma)^2 \phi_0^\top A^{-1} \Sigma A^{-\top} \phi_0$ does not lend itself to easy intuition, let alone how it connects to and unifies existing results.

**Warm-up: the tabular case.** We start with the tabular setting and show that $C_\phi^\pi$ becomes something familiar, offering some basic intuitions as well as assurance that $C_\phi^\pi$, a quantity that falls out of the IV concentration analyses, holds meaningful interpretations in RL. The key is to rewrite $C_\phi^\pi$ as

$$\phi_0^\top A^{-1} \Sigma A^{-\top} \phi_0 = \phi_0^\top (I - \gamma B^\pi)^{-1} \Sigma^{-1} (I - \gamma B^\pi)^{-\top} \phi_0, \quad \text{where } B^\pi := \Sigma^{-1} \Sigma_{\text{cr}}. \quad (14)$$

The tabular setting can be viewed as a special case of linear function approximation with $d = |\mathcal{S} \times \mathcal{A}|$, and $\phi(s,a) = \mathbf{e}_{s,a}$ is the unit vector with the $(s,a)$-th coordinate being 1 and all other coordinates being 0. In this case, $\phi_0$ is simply the vector representation of the initial state-action distribution $\mu_0^\pi$, where $(s_0, a_0) \sim \mu_0^\pi \Leftrightarrow s_0 \sim \mu_0, a_0 \sim \pi$; $B^\pi$ is an $|\mathcal{S} \times \mathcal{A}| \times |\mathcal{S} \times \mathcal{A}|$ matrix with $[B^\pi]_{(s,a),(s',a')} = P^\pi(s', a'|s, a) = P(s'|s, a)\pi(a'|s')$, i.e., the transition kernel of the Markov chain over $\mathcal{S} \times \mathcal{A}$ induced by policy $\pi$. Put together, we have the textbook identity

$$(1-\gamma)\phi_0^\top (I - \gamma B^\pi)^{-1} = (\mu^\pi)^\top,$$

where we recall the definition of the discounted occupancy $\mu^\pi$ from Eq. (2). Plugging it into $C_\phi^\pi$,

$$C_\phi^\pi = (\mu^\pi)^\top \Sigma^{-1} \mu^\pi = \sum_{s,a} \mu^\pi(s,a)^2 / \mu^D(s,a) = \mathbb{E}_{\mu^D}[(\mu^\pi/\mu^D)^2],$$

which is the $\chi^2$-divergence between $\mu^\pi$ and $\mu^D$ up to a constant shift and has appeared as a tight coverage parameter (especially when compared to $\|\mu^\pi/\mu^D\|_\infty$ (Xie & Jiang, 2020b)) when coverage is measured based on density ratios.

### 5.1 GENERAL INTERPRETATION

We now offer the interpretation for the general setting. Note that $B^\pi$ can be viewed as the multi-variate linear-regression solution of the regression problem $\phi(s,a) \mapsto \phi(s', \pi)$, thus

$$\mathbb{E}_{s' \sim P(\cdot|s,a)}[\phi(s', \pi)] \approx B^\pi \phi(s,a). \quad (15)$$

In general, the above relationship is only approximate (in Section 5.3 we will see that it becomes *exact* under an additional assumption), although $B^\pi$ is the best linear predictor. This leads to the following interpretation of $C_\phi^\pi$ (see Appendix B.1 for the proof):

**Proposition 1.** *Define a deterministic linear dynamical system $\{x_t\}_{t \geq 0}$, with $x_0 := \phi_0$, and $\forall t \geq 0$,*

$$x_{t+1} = (B^\pi)^\top x_t.$$

*When $\rho(B^\pi) < 1/\gamma$, define the feature occupancy in $B^\pi$ as $\mu_\phi^\pi := (1 - \gamma) \sum_{t \geq 0} \gamma^t x_t$, then*

$$C_\phi^\pi = (\mu_\phi^\pi)^\top \Sigma^{-1} \mu_\phi^\pi.$$

The proposition rewrites $C_\phi^\pi$ in a form that closely resembles the standard notion of linear coverage in the literature, where we see the expected feature occupancy under the target policy ($\mu_\phi^\pi$ here) measured under the data-covariance norm $\Sigma^{-1}$; see Section 5.3. Accordingly, we call $C_\phi^\pi$ the **feature-dynamics coverage**. The difference is that here the feature occupancy is defined in a deterministic dynamical system $B^\pi$ instead of the true MDP. Furthermore, while the latter, $\phi^\pi := \mathbb{E}_{(s,a) \sim \mu^\pi}[\phi(s, a)]$, is always bounded, $\mu_\phi^\pi$, on the other hand, may not be bounded in general and $\{x_t\}_{t \geq 0}$ may actually diverge. The connection between LSTD and the linear dynamical system $B^\pi$ was first identified by Parr et al. (2008) (see also Duan & Wang (2020)), though they focused on the algebraic equivalence between LSTD and the model-based solution in $B^\pi$, and did not perform finite-sample analyses or connect this to the notion of coverage.

**When is feature-dynamics coverage well-behaved?** Our bound sharpens and generalizes existing understanding of when linear OPE using only realizability is possible. We provide a comparison to Perdomo et al., whose analysis was shown to be sharp and subsume many prior conditions known in the literature. They establish that, under some regularity assumptions, $\|\Sigma^{1/2}(\theta^\star - \widehat{\theta})\|_2 \lesssim \frac{1}{\sigma_{\min}(I - \gamma \Sigma^{-1/2} \Sigma_{\mathrm{cr}} \Sigma^{-1/2})} \cdot \varepsilon_{\mathrm{stat}}$, for some $\varepsilon_{\mathrm{stat}}$ which is polynomial in $d, 1/n, \log(1/\delta)$, and spectral properties of $\Sigma$. While they only show function-estimation guarantee on $\mu^D$ (c.f. Appendix C), this intermediate result immediately implies a return-estimation guarantee comparable to ours:

$$\left| J_{\widehat{Q}_{\mathrm{lstd}}}(\pi) - J(\pi) \right| \leq \frac{\|\phi_0\|_{\Sigma^{-1}}}{\sigma_{\min}(I - \gamma \Sigma^{-1/2} \Sigma_{\mathrm{cr}} \Sigma^{-1/2})} \cdot \varepsilon_{\mathrm{stat}}.$$

As we have already shown that our statistical rate is tight, it suffices to compare our $C_\phi^\pi$ to their multiplicative factor in front of $\varepsilon_{\mathrm{stat}}$. In particular, we establish the following relationship (see Appendix B.2 for the proof).

**Proposition 2.**

$$\sqrt{C_\phi^\pi} = (1 - \gamma)\|(I - \gamma \Sigma^{-1/2} \Sigma_{\mathrm{cr}} \Sigma^{-1/2})^{-T} \Sigma^{-1/2} \phi_0\|_2 \leq (1 - \gamma) \frac{\|\phi_0\|_{\Sigma^{-1}}}{\sigma_{\min}(I - \gamma \Sigma^{-1/2} \Sigma_{\mathrm{cr}} \Sigma^{-1/2})}.$$

This demonstrates that our coverage parameter provides a tighter return-estimation guarantee compared to the approach of Perdomo et al. (2023). As an immediate consequence, we also subsume other known conditions for this setting that were captured by Perdomo et al., including on-policy sampling (Tsitsiklis & Van Roy, 1997), Bellman completeness, low distribution shift (Wang et al., 2021), symmetric stability (Mou et al., 2022a), and contractivity (Kolter, 2011) (see the discussion in Perdomo et al. for formal definitions). Furthermore, we consider the $1/\sigma_{\min}$-type bound to provide little intuition about necessary coverage conditions for this fundamental task, and the unification of $C_\phi^\pi$ with existing concepts in the literature to be a major contribution.

## 5.2 RECOVERING AGGREGATED CONCENTRABILITY

State abstractions are a special case of linear function approximation, where each state $s$ is mapped to one of the $K$ abstract states, $\psi(s) \in \{1, \ldots, K\}$, effectively treating states with the same $\psi(s)$ as aggregated and equivalent to reduce the size of the state space. Under the abstraction scheme, the natural model-based solution coincides with LSTDQ with $\phi(s, a) = \mathbf{e}_{k,a}$. When we only assume realizability (Assumption 1), this has been analyzed by Xie & Jiang (2020a); Zhang & Jiang (2021); Jia et al. (2024) with the following notion of *aggregated concentrability* as its coverage parameter.

**Definition 1** (Aggregated concentrability). *Given $\psi : \mathcal{S} \to \{1, \ldots, K\}$, define the abstract MDP $M_\phi = (\mathcal{S}_\phi, \mathcal{A}, P_\phi, R_\phi, \gamma, \mu_0)$ where[4] $\mathcal{S}_\phi = \{1, \ldots, K\}$, and*

$$P_\phi(k'|k, a) = \frac{\mu^D(s, a) \cdot \sum_{s : \psi(s) = k} \left( \sum_{s' : \psi(s') = k'} P(s'|s, a) \right)}{\sum_{s : \psi(s) = k} \mu^D(s, a)}.$$

---

[4]The definition of $R_\phi$ is irrelevant for our purpose and thus omitted.

Figure 1: Illustration of the evolution of occupancies under the true dynamics $P^\pi$ (top row) and that of features under the compressed dynamics $B^\pi$ (bottom row). Under Bellman completeness, the dashed blue arrows hold and two routes ($\rightarrow \ldots \rightarrow \downarrow$ vs. $\downarrow \rightarrow \ldots \rightarrow$) yield the same expected feature vectors, but they are generally different without such an assumption.

*For any $\pi$ that only depends on $s$ through $\psi(s)$, aggregated concentrability refers to measures of $\mu_{M_\phi}^\pi / \mu_\phi^D$, either in $\|\cdot\|_\infty$ or $\chi^2$ form, where $\mu_{M_\phi}^\pi$ is discounted occupancy in MDP $M_\phi$, and $\mu_\phi^D(k, a) = \sum_{s:\psi(s)=k} \mu^D(s, a)$.*

In this definition, $P_\phi$ is the dynamics over the abstract state space, and it is easy to see that the transition kernel of abstract-state pairs under $\pi$ is $P_\phi^\pi = B^\pi$, and $\Sigma = \text{diag}(\mu_\phi^D)$. As a result, $C_\phi^\pi$ recovers the $\chi^2$ version of aggregated concentrability (see Appendix B.3 for the proof):

**Proposition 3.** *When $\phi$ is induced by a state abstraction $\psi$ and $\pi$ depends on $s$ only through $\psi(s)$,*

$$C_\phi^\pi = \mathbb{E}_{(k,a)\sim\mu_\phi^D}[(\mu_{M_\phi}^\pi / \mu_\phi^D)^2].$$

### 5.3 RECOVERING STANDARD LINEAR COVERAGE UNDER BELLMAN-COMPLETENESS

Prior results on abstractions leave an intriguing question open: they measure coverage by analyzing error propagation in $M_\phi$, which a lower-dimensional and approximate model **compressed** from $M$ by $\phi$, as evidenced by $\mu_{M_\phi}^\pi$ in the definition of aggregated concentrability; this is also consistent with our results in Section 5.1 where occupancy is measured in the compressed linear dynamical system $B^\pi$. On the other hand, the mainstream notion of coverage in linear OPE, obtained under the Bellman-completeness, is $C_{\text{lin}}^\pi = (\phi^\pi)^\top \Sigma^{-1} \phi^\pi$ (Eq. (5)), which is concerned with error propagation in the **true dynamics** $M$ since $\phi^\pi$ is defined w.r.t. the occupancy $\mu^\pi$ in $M$. This begs the question:

> *Is error-propagation in **compressed** models,*
> *as in ($Q^\pi$-irrelevant) abstractions, an exception and outlier?*

While anecdotally this has been the general perception from the community, our results below suggest otherwise, and *the results that are seemingly disconnected with each other can be elegantly unified* through the following proposition (see Appendix B.4 for the proof):

**Proposition 4.** *Let $\mathcal{F}_\phi := \{\phi^\top \theta : \theta \in \mathbb{R}^d\}$ be the space of functions linear in $\phi$. Assume $\mathcal{F}_\phi$ satisfies Bellman-completeness (Assumption 3). Then, (1) $B^\pi$ becomes an exact model for next-feature prediction, i.e., $\mathbb{E}_{s'\sim P(\cdot|s,a)}[\phi(s', \pi)] = (B^\pi)^\top \phi(s, a)$, (2) $\mu_\phi^\pi = \phi^\pi$, (3) $\rho(B^\pi) \leq 1$, and (4)*

$$C_\phi^\pi = C_{\text{lin}}^\pi = (\phi^\pi)^\top \Sigma^{-1} \phi^\pi.$$

The essence of the proposition is illustrated in Figure 1, showing that the expected features produced by the groundtruth dynamics ($\phi^\pi$) and the compressed dynamics ($\mu_\phi^\pi = (1 - \gamma) \sum_t \gamma^t x_t$) coincide under Bellman-completeness, thus demonstrating that **error propagation through true dynamics is a special case of and thus unified with that through compressed dynamics.**

**Connection to Bellman Residual Minimization (BRM).** Many (if not most) algorithms for learning $Q^\pi$ with general function approximation coincide with LSTDQ under linear function approximation (Antos et al., 2008; Xie et al., 2021; Uehara et al., 2020), and this fact allows us to compare our bound to the more general analyses in the literature. Among those algorithms, BRM

is a well-investigated example, which approximates $Q^\pi$ by solving the following minimax problem (Antos et al., 2008):

$$\widehat{f}^\pi = \arg\min_{f \in \mathcal{F}} \sup_{f' \in \mathcal{F}} \left( \mathbb{E}_{\mathcal{D}}[(f(s,a) - r - f(s',\pi)^2] - \mathbb{E}_{\mathcal{D}}[(f'(s,a) - r - f(s',\pi)^2] \right), \quad (16)$$

whose finite-sample guarantee can be established under Bellman completeness (Assumption 3). Antos et al. (2008); Xie et al. (2021) show that when $\mathcal{F}$ is linear, the solution coincides with LSTDQ, so we can compare the guarantee of BRM under linear $\mathcal{F}$ with our Theorem 1. Jiang & Xie (2024) show that BRM's error bound is (see their Eq. (18))

$$\left| J_{\widehat{f}^\pi}(\pi) - J(\pi) \right| \lesssim \frac{V_{\max}}{1 - \gamma} \cdot \sqrt{\frac{C^\pi \log(|\mathcal{F}|/\delta)}{n}}.$$

In the linear setting, their $C^\pi$ is $C_{\mathrm{lin}}^\pi$ (see their Eq. (22)), and $\log |\mathcal{F}| \approx d$ based on a standard covering-number argument. Under such translation, the main $O(n^{-\frac{1}{2}})$ term in our Eq. (12) match the guarantee of BRM, not only in coverage, but also in horizon and $d$ dependence.

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

. To proceed, we note that, when $\widehat{A}$ is invertible, we have $\widehat{A}\widehat{\theta}_{\mathrm{lstd}} - \hat{b} = \widehat{A}\left( \widehat{A}^{-1}\hat{b} \right) - \hat{b} = 0$, and thus, $\left\| \widehat{\Sigma}^{-1/2} \left( \widehat{A}\widehat{\theta}_{\mathrm{lstd}} - b \right) \right\|_2 = 0$ whenever $\widehat{\Sigma}$ is invertible. Thus,

$$\left\| \widehat{\Sigma}^{-1/2} \left( \widehat{A}\theta^\star - \widehat{A}\widehat{\theta}_{\mathrm{lstd}} \right) \right\|_2 \leq \left( \left\| \widehat{\Sigma}^{-1/2} \left( \widehat{A}\theta^\star - \hat{b} \right) \right\|_2 + \left\| \widehat{\Sigma}^{-1/2} \left( \widehat{A}\widehat{\theta}_{\mathrm{lstd}} - \hat{b} \right) \right\|_2 \right)$$
$$\leq \left\| \widehat{\Sigma}^{-1/2} \left( \widehat{A}\theta^\star - \hat{b} \right) \right\|_2.$$

We then note that

$$\left\| \widehat{\Sigma}^{1/2} \widehat{A}^{-\top} \phi_0 \right\|_2 = \sqrt{\phi_0^\top \widehat{A}^{-1} \widehat{\Sigma} \widehat{A}^{-T} \phi_0} = \frac{1}{1 - \gamma} \sqrt{\widehat{C}_\phi^\pi},$$

which yields that

$$\left| J_{\widehat{Q}_{\mathrm{lstd}}}(\pi) - J(\pi) \right| \leq \frac{1}{1 - \gamma} \sqrt{\widehat{C}_\phi^\pi} \left\| \widehat{\Sigma}^{-1/2} \left( \widehat{A}\theta^\star - \hat{b} \right) \right\|_2.$$

The proof will be concluded by establishing the following concentration lemma.

**Lemma 1.** *With probability $1 - \delta$ over the randomness of the rewards and sampled transitions, we have:*

$$\|\widehat{\Sigma}^{-1/2}(\widehat{A}\theta^\star - \hat{b})\|_2 \leq \mathcal{O}\left( V_{\max} \sqrt{\frac{d \log(1/\delta)}{n}} \right).$$

$\square$

**Proof of Lemma 1.** We firstly note that

$$\hat{A}\theta^\star - \hat{b} = \frac{1}{n} \sum_{i=1}^n \phi(s_i, a_i)\big(\phi(s_i, a_i)^\top \theta^\star - \gamma\phi(s_i', a_i')^\top \theta^\star - r_i\big)$$

$$= \frac{1}{n} \sum_{i=1}^n \phi(s_i, a_i)\big(\underbrace{Q^\pi(s_i, a_i) - r_i - \gamma Q^\pi(s_i', a_i')}_{:=\varepsilon_i}\big).$$

Thus, $\hat{A}\theta^\star - \hat{b}$ is a random variable that is conditionally zero-mean, when taking conditional expectations over the $r_i \sim R(s_i, a_i)$, $s_i' \sim P(s_i, a_i)$, and $a_i' \sim \pi(\cdot \mid s_i')$ (keeping the design over $s_i, a_i$ fixed). Note that $|\varepsilon_i| \leq 2V_{\max}$.

We apply a vector martingale Bernstein inequality (Lemma 10) on the random variables $Z_i = \widehat{\Sigma}^{-1/2}\phi(s_i, a_i)\varepsilon_i$. For $i \in [n]$, we let $\mathcal{H}_i = \{s_1, a_1, \ldots, s_n, a_n\} \cup \{r_1, s_1', a_1', \ldots, r_i, s_i', a_i'\}$ denote the histories including the entire design over $s_i, a_i$ but only the first $i$ samples from $r_i, s_i', a_i'$. Note that $Z_i$ is adapted to the filtration generated by $\mathcal{H}_i$, and is a martingale difference sequence, since

$$\mathbb{E}_{i-1}[\widehat{\Sigma}^{-1/2}\phi(s_i, a_i)\varepsilon_i] = \mathbb{E}_{i-1}[\varepsilon_i]\|\phi(s_i, a_i)\|_{\widehat{\Sigma}^{-1}} = 0.$$

In the sequel we establish the following simple technical lemma.

**Lemma 2.** *Let $x_1, \ldots, x_n \in \mathbb{R}^d$, and assume $\widehat{\Sigma} = \frac{1}{n} \sum_{i=1}^n x_i x_i^\top$ is invertible. Then for all $i \in [n]$ we have:*

$$x_i^\top \widehat{\Sigma}^{-1} x_i \leq n$$

Using, Lemma 2, we then have that

$$\|\widehat{\Sigma}^{-1/2}\phi(s_i, a_i)\varepsilon_i\|_2 \leq 2V_{\max}\|\phi(s_i, a_i)\|_{\widehat{\Sigma}^{-1}} \leq 2V_{\max}\sqrt{n},$$

which establishes the norm bound. Lastly, for the variance term, we have:

$$\sum_{i=1}^n \mathbb{E}_{i-1}[\|X_i\|^2] = \sum_{i=1}^n \mathbb{E}_{i-1}[\varepsilon_i^2 \|\phi_i\|_{\widehat{\Sigma}^{-1}}^2] \leq 4V_{\max}^2 \sum_{i=1}^n \|\phi_i\|_{\widehat{\Sigma}^{-1}}^2.$$

Note that the summation is equivalent to:

$$\sum_{i=1}^n \phi_i^\top \widehat{\Sigma}^{-1} \phi_i = \sum_{i=1}^n \text{tr}(\widehat{\Sigma}^{-1}\phi_i\phi_i^\top) = n\text{tr}(\widehat{\Sigma}^{-1}\frac{1}{n}\sum_{i=1}^n \phi_i\phi_i^\top) = nd,$$

since the trace of the identity matrix is $d$. Plugging these observations into Lemma 10 gives that

$$\left\|\widehat{\Sigma}^{-1/2}(\hat{A}\theta^\star - \hat{b})\right\|_2 = \frac{1}{n}\left\|\sum_{i=1}^n Z_i\right\|_2 \leq \frac{1}{n}\left(V_{\max}\sqrt{8nd\log(2/\delta)} + \frac{2}{3}V_{\max}\sqrt{n}\log(2/\delta)\right)$$

$$= \mathcal{O}\left(V_{\max}\sqrt{\frac{d\log(1/\delta)}{n}}\right),$$

as desired. $\qquad\square$

**Proof of Lemma 2.** Consider the un-normalized empirical covariance matrix $\widehat{\Sigma}_{\text{un}} = \sum_{i=1}^n x_i x_i^\top$. Let $v = x_i^\top \Sigma_{\text{un}}^{-1} x_i$. For each $i$, let $\Sigma = S_i + x_i x_i^\top$. Note that $S_i = \sum_{j \neq i} x_j x_j^\top$ is a PSD matrix, as is $\widehat{\Sigma}_{\text{un}}^{-1} S_i \widehat{\Sigma}_{\text{un}}^{-1}$. Then, we have

$$0 \leq x_i^\top \widehat{\Sigma}_{\text{un}}^{-1} S_i \widehat{\Sigma}_{\text{un}}^{-1} x_i = x_i^\top \Sigma_{\text{un}}^{-1} x_i - (x_i^\top \Sigma_{\text{un}}^{-1} x_i)^2.$$

This implies that $v(1-v) \geq 0$, thus $0 \leq v \leq 1$. $\qquad\square$

A.2   PROOF OF COROLLARY 1

**Corollary 1.** *There exists $n_0$ such that when $n \geq n_0$, w.p. $\geq 1 - \delta$,*

$$\left| J_{\widehat{Q}_{\mathrm{lstd}}}(\pi) - J(\pi) \right| \lesssim \frac{V_{\max}}{1 - \gamma} \sqrt{\frac{C_\phi^\pi \cdot d \log(1/\delta)}{n}} + o(\sqrt{1/n}), \tag{12}$$

*where*

$$C_\phi^\pi := (1 - \gamma)^2 \phi_0^\top A^{-1} \Sigma A^{-\top} \phi_0 \tag{13}$$

*and $n_0$ and the $o(\sqrt{1/n})$ term may additionally depend on $1/\sigma_{\min}(A)$.*

**Proof of Corollary 1.** We begin by noting that it is sufficient to provide a high-probability bound on $\left| C_\phi^\pi - \widehat{C}_\phi^\pi \right| \leq \varepsilon_n$ for some $\varepsilon_n = o(1)$, since by Theorem 1 and the inequality $\sqrt{a + b} \leq \sqrt{a} + \sqrt{b}$ we will then obtain

$$\left| J_{\widehat{Q}_{\mathrm{lstd}}}(\pi) - J(\pi) \right| \lesssim \frac{V_{\max}}{1 - \gamma} \cdot \sqrt{\frac{\left( C_\phi^\pi + \varepsilon_n \right) \cdot d \log(1/\delta)}{n}}$$

$$\leq \frac{V_{\max}}{1 - \gamma} \cdot \sqrt{\frac{C_\phi^\pi \cdot d \log(1/\delta)}{n}} + \frac{V_{\max}}{1 - \gamma} \cdot \sqrt{\frac{\varepsilon_n \cdot d \log(1/\delta)}{n}}$$

$$= \frac{V_{\max}}{1 - \gamma} \cdot \sqrt{\frac{C_\phi^\pi \cdot d \log(1/\delta)}{n}} + o(\sqrt{1/n}).$$

We now proceed to bound $\left| C_\phi^\pi - \widehat{C}_\phi^\pi \right|$ with high probability. Towards this, we note that:

$$\left| C_\phi^\pi - \widehat{C}_\phi^\pi \right| = (1 - \gamma)^2 \left| \|\Sigma^{1/2} A^{-\top} \phi_0\|_2 - \|\widehat{\Sigma}^{1/2} \hat{A}^{-\top} \phi_0\|_2 \right|$$

$$\leq (1 - \gamma)^2 \left\| \Sigma^{1/2} A^{-\top} \phi_0 - \widehat{\Sigma}^{1/2} \hat{A}^{-\top} \phi_0 \right\|_2$$

$$\leq (1 - \gamma)^2 \left( \left\| \widehat{\Sigma}^{1/2} \left( A^{-\top} - \hat{A}^{-\top} \right) \phi_0 \right\|_2 + \left\| \left( \widehat{\Sigma}^{1/2} - \Sigma^{1/2} \right) A^{-\top} \phi_0 \right\|_2 \right)$$

$$\leq (1 - \gamma)^2 B_\phi \left( \left\| \widehat{\Sigma}^{1/2} \right\|_2 \|A^{-1} - \hat{A}^{-1}\|_2 + \left\| \widehat{\Sigma}^{1/2} - \Sigma^{1/2} \right\|_2 \|A^{-1}\|_2 \right),$$

via applications of the triangle inequality and operator norm bounds. Let $\varepsilon(\Sigma^{1/2}) = \|\Sigma^{1/2} - \widehat{\Sigma}^{1/2}\|_2$ and $\varepsilon(A^{-1}) = \|A^{-1} - \hat{A}^{-1}\|_2$. Note that the above inequalities imply

$$\left| C_\phi^\pi - \widehat{C}_\phi^\pi \right| \leq (1 - \gamma)^2 B_\phi \left( \left( \lambda_{\max}(\Sigma^{1/2}) + \varepsilon(\Sigma^{1/2}) \right) \varepsilon(A^{-1}) + \varepsilon(\Sigma^{1/2}) \frac{1}{\sigma_{\min}(A)} \right). \tag{18}$$

We conclude by bounding $\varepsilon(\Sigma^{1/2})$ and $\varepsilon(A^{-1})$. We first establish a concentration lemma for $\|\Sigma - \widehat{\Sigma}\|_2$ and $\|A - \hat{A}\|_2$, and then show how this can be converted to bounds for $\|\Sigma^{1/2} - \widehat{\Sigma}^{1/2}\|_2$ and $\|A^{-1} - \hat{A}^{-1}\|_2$. The following concentration lemma will be proved in the sequel. □

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

$$x_{t+1} = (B^\pi)^\top x_t.$$

*When $\rho(B^\pi) < 1/\gamma$, define the feature occupancy in $B^\pi$ as $\mu_\phi^\pi := (1 - \gamma)\sum_{t\geq 0}\gamma^t x_t$, then*

$$C_\phi^\pi = (\mu_\phi^\pi)^\top\Sigma^{-1}\mu_\phi^\pi.$$

**Proof of Proposition 1.** Recall that we defined $B^\pi = \Sigma^{-1}\Sigma_{\mathrm{cr}}$. We note that

$$A = \Sigma - \gamma\Sigma_{\mathrm{cr}} = \Sigma(I - \gamma B^\pi).$$

Substituting this into $C_\phi^\pi$, we arrive at the expression:

$$C_\phi^\pi = (1-\gamma)^2\phi_0^\top A^{-1}\Sigma A^{-T}\phi_0 = (1-\gamma)^2\phi_0^\top(I - \gamma B^\pi)^{-1}\Sigma^{-1}(I - \gamma B^\pi)^{-T}\phi_0. \quad (19)$$

Note that when $\rho(B^\pi) < 1/\gamma$, the matrix $(I - \gamma B^\pi)^{-\top}$ has the series expansion:

$$(I - \gamma B^\pi)^{-\top} = \sum_{t=0}^{\infty}\gamma^t((B^\pi)^\top)^t.$$

Thus, we notice that

$$(I - \gamma B^\pi)^{-\top}\phi_0 = \sum_{t=0}^{\infty}\gamma^t((B^\pi)^\top)^t\phi_0 = \sum_{t=0}^{\infty}\gamma^t x_t = \frac{1}{1-\gamma}\mu_\phi^\pi.$$

Substituting this into Eq. (19) gives the result. $\square$

## B.2 PROOF OF PROPOSITION 2

**Proposition 2.**

$$\sqrt{C_\phi^\pi} = (1-\gamma)\|(I - \gamma\Sigma^{-1/2}\Sigma_{\mathrm{cr}}\Sigma^{-1/2})^{-T}\Sigma^{-1/2}\phi_0\|_2 \le (1-\gamma)\frac{\|\phi_0\|_{\Sigma^{-1}}}{\sigma_{\min}(I - \gamma\Sigma^{-1/2}\Sigma_{\mathrm{cr}}\Sigma^{-1/2})}.$$

**Proof of Proposition 2.** The derivation is as follows:

$$
\begin{aligned}
\phi_0^\top A^{-1}\Sigma A^{-T}\phi_0 &= \phi_0(\Sigma - \gamma\Sigma_{\mathrm{cr}})^{-1}\Sigma(\Sigma - \gamma\Sigma_{\mathrm{cr}})^{-T}\phi_0 \\
&= \phi_0^\top \left(\Sigma^{1/2}(I - \gamma\Sigma^{-1/2}\Sigma_{\mathrm{cr}}\Sigma^{-1/2})\Sigma^{1/2}\right)^{-1}\Sigma\left(\Sigma^{1/2}(I - \gamma\Sigma^{-1/2}\Sigma_{\mathrm{cr}}\Sigma^{-1/2})\Sigma^{1/2}\right)^{-T}\phi_0 \\
&= \phi_0^\top \Sigma^{-1/2}(I - \gamma\Sigma^{-1/2}\Sigma_{\mathrm{cr}}\Sigma^{-1/2})^{-1}(I - \gamma\Sigma^{-1/2}\Sigma_{\mathrm{cr}}\Sigma^{-1/2})^{-T}\Sigma^{-1/2}\phi_0 \\
&= \|(I - \gamma\Sigma^{-1/2}\Sigma_{\mathrm{cr}}\Sigma^{-1/2})^{-T}\Sigma^{-1/2}\phi_0\|_2^2.
\end{aligned}
$$

$\square$

## B.3 PROOF OF PROPOSITION 3

**Proposition 3.** *When $\phi$ is induced by a state abstraction $\psi$ and $\pi$ depends on $s$ only through $\psi(s)$,*

$$C_\phi^\pi = \mathbb{E}_{(k,a)\sim\mu_\phi^D}[(\mu_{M_\phi}^\pi/\mu_\phi^D)^2].$$

**Proof of Proposition 3.** Let $\phi(s,a) = e_{\psi(s),a}$, where $\psi$ is the state abstraction function. We compute the $A$ matrix. Below, we define $P^\pi(s',a' \mid s,a) = P(s' \mid s,a)\pi(a' \mid s')$, $P(k' \mid s,a) = \sum_{s':\psi(s')=k} P(s' \mid s,a)$, and

$$P^\pi(k',a' \mid s,a) = P(k' \mid s,a)\pi(a' \mid k'),$$

which is valid since $\pi$ is consistent with the state abstraction. To start, the covariance matrix $\Sigma$ becomes

$$
\begin{aligned}
\Sigma = \mathbb{E}_{s,a\sim\mu^D}\left[\phi(s,a)\phi(s,a)^\top\right] &= \sum_{k\in[K],a\in[A]} \sum_{s\in\mathcal{S}:\psi(s)=k} \mu^D(s,a)e_{k,a}e_{k,a}^\top \\
&= \sum_{k\in[K],a\in[A]} e_{k,a}e_{k,a}^\top \left(\sum_{s\in\mathcal{S}:\psi(s)=k} \mu^D(s,a)\right) \\
&= \sum_{k\in[K],a\in[A]} e_{k,a}e_{k,a}^\top \mu_\phi^D(k,a) := \bar{D}_{\mathrm{data}} \in \mathbb{R}^{[K]\cdot[A]\times[K]\cdot[A]},
\end{aligned}
$$

where we recalled the definition of $\mu_\phi^D(k, a) = \sum_{s \in \mathcal{S}: \psi(s)=k} \mu^D(s, a)$, and introduced the diagonal matrix $\bar{D}_{\text{data}}$ with elements $\mu_\phi^D(k, a)$ along the diagonal. Let's examine the cross-covariance $\Sigma_{\text{cr}}$.

$$
\begin{aligned}
\Sigma_{\text{cr}} &= \mathbb{E}_{s,a \sim \mu^D}\left[\phi(s, a)\phi(s', a')^\top\right] \\
&= \sum_{s \in \mathcal{S}, a \in \mathcal{A}} \mu^D(s, a)\phi(s, a) \sum_{s' \in \mathcal{S}, a' \in \mathcal{A}} P^\pi(s', a' \mid s, a)\phi(s', a')^\top \\
&= \sum_{k \in [K], a \in [A]} e_{k,a} \sum_{s \in \mathcal{S}: \psi(s)=k} \mu^D(s, a)\left(\sum_{k' \in [K], a' \in [A]} \sum_{s' \in \mathcal{S}: \psi(s')=k'} P^\pi(s', a' \mid s, a)e_{k',a'}^\top\right) \\
&= \sum_{k \in [K], a \in [A]} e_{k,a} \sum_{s \in \mathcal{S}: \psi(s)=k} \mu^D(s, a)\left(\sum_{k' \in [K], a' \in [A]} e_{k',a'}^\top P^\pi(k', a' \mid s, a)\right) \\
&= \sum_{k \in [K], a \in [A]} e_{k,a} \sum_{k' \in [K], a' \in [A]} e_{k',a'}^\top \sum_{s \in \mathcal{S}: \psi(s)=k} \mu^D(s, a)P^\pi(k', a' \mid s, a) \\
&= \sum_{k \in [K], a \in [A]} e_{k,a} \sum_{k' \in [K], a' \in [A]} e_{k',a'}^\top \mu_\phi^D(k, a)\left(\frac{\sum_{s \in \mathcal{S}: \psi(s)=k} \mu^D(s, a)P^\pi(k', a' \mid s, a)}{\mu_\phi^D(k, a)}\right) \\
&:= \sum_{k \in [K], a \in [A]} e_{k,a} \sum_{k' \in [K], a' \in [A]} e_{k',a'}^\top \mu_\phi^D(k, a)P_\phi^\pi(k', a' \mid k, a) \\
&= \bar{D}_{\text{data}} P_\phi^\pi,
\end{aligned}
$$

where we recall the definition of the aggregated transition matrix $P_\phi^\pi$ with elements

$$
P_\phi^\pi(k', a' \mid k, a) = \frac{\sum_{s: \psi(s)=k} \mu^D(s, a)P^\pi(k', a' \mid s, a)}{\mu_\phi^D(k, a)}.
$$

Putting our expressions for $\Sigma$ and $\Sigma_{\text{cr}}$ together, we conclude that

$$
A = \bar{D}_{\text{data}}(I - \gamma P_\phi^\pi).
$$

Note that $P_\phi^\pi$ is the $\pi$-dependent transition kernel of the MDP $M_\phi$ over the state space $[K]$ with action space $[A]$. We assign the MDP an initial state-action distribution $\mu_{0,\phi}$ in the canonical way:

$$
\mu_{0,\phi}^\pi(k, a) = \sum_{s: \psi(s)=k} \mu_0(s)\pi(a \mid s) = \mu_0(k)\pi(a \mid k),
$$

again using the fact that $\pi$ is consistent with the abstraction $\psi$. Note that in the state abstraction setting, we have

$$
\phi_0 = \mathbb{E}_{s_0 \sim \mu_0, a \sim \pi}[\phi(s, a)] = \sum_{k,a} e_{k,a} \sum_{s: \psi(s)=k} \mu_0(s)\pi(a \mid s) = \mu_{0,\phi}^\pi,
$$

Finally, our coverage coefficient becomes

$$
C_\phi^\pi = (1 - \gamma)^2 \phi_0 A^{-1} \Sigma A^{-T} \phi_0 = (1 - \gamma)^2 \mu_{0,\phi}^\pi (I - \gamma P_\phi^\pi)^{-1} \bar{D}_{\text{data}}^{-1} (I - \gamma P_\phi^\pi)^{-T} \mu_{0,\phi}^\pi.
$$

Since $P_\phi^\pi$ is a stochastic kernel with spectral radius less than 1, we have

$$
(I - \gamma P_\phi^\pi)^{-T} \mu_{0,\phi} = \sum_{t \geq 0} \gamma^t ((P_\phi^\pi)^t))^\top \mu_{0,\phi} = \frac{1}{1 - \gamma} \mu_{M_\phi}^\pi,
$$

i.e. this is precisely the discounted occupancy of policy $\pi$ in the abstract MDP $M_\phi$. Thus,

$$
C_\phi^\pi = (\mu_{M_\phi}^\pi)^\top \bar{D}_{\text{data}}^{-1}(\mu_{M_\phi}^\pi) = \mathbb{E}_{k,a \sim \mu_\phi^D}\left[(\mu_{M_\phi}^\pi / \mu_\phi^D)^2\right],
$$

as desired.

$\square$

### B.4 Proof of Proposition 4

**Proposition 4.** *Let $\mathcal{F}_\phi := \{\phi^\top \theta : \theta \in \mathbb{R}^d\}$ be the space of functions linear in $\phi$. Assume $\mathcal{F}_\phi$ satisfies Bellman-completeness (Assumption 3). Then, (1) $B^\pi$ becomes an exact model for next-feature prediction, i.e., $\mathbb{E}_{s' \sim P(\cdot|s,a)}[\phi(s', \pi)] = (B^\pi)^\top \phi(s,a)$, (2) $\mu_\phi^\pi = \phi^\pi$, (3) $\rho(B^\pi) \leq 1$, and (4)*

$$C_\phi^\pi = C_{\mathrm{lin}}^\pi = (\phi^\pi)^\top \Sigma^{-1} \phi^\pi.$$

**Proof of Proposition 4.**

**(1) $B^\pi$ is an exact next-feature predictor:** $\mathbb{E}_{s' \sim P(\cdot|s,a)}[\phi(s,a)] = B^\pi \phi(s,a)$ **for all** $(s,a)$. First, we show that under Bellman completeness $\mathcal{F}_\phi$ is also closed under the transition operator $\mathcal{P}^\pi := \mathcal{T}^\pi - R$, that is, $\mathcal{P}^\pi f \in \mathcal{F}_\phi$ for all $f \in \mathcal{F}_\phi$.

The linearity of $\mathcal{F}_\phi$ together with Bellman completeness immediately imply that the reward function is linear, or $R \in \mathcal{F}_\phi$. Define $f_0 \in \mathcal{F}_\phi$ to be the function corresponding to the parameter $\theta = \mathbf{0}_d$, so that $f_0(s,a) = 0$ for all $(s,a)$; we have $\mathcal{T}^\pi f_0 = R \in \mathcal{F}_\phi$.

Next, fix any $f \in \mathcal{F}_\phi$ and observe that $\mathcal{T}^\pi f$ is also linear, since $\mathcal{T}^\pi f \in \mathcal{F}_\phi$ under Bellman completeness. It follows that $\mathcal{T}^\pi f - R = \mathcal{P}^\pi f \in \mathcal{F}_\phi$ because the difference of two functions linear in the same features is also linear in those features, which proves that $\mathcal{F}_\phi$ is closed under $\mathcal{P}^\pi$. This closure implies that for any $f \in \mathcal{F}_\phi$, there exists some $\theta_f \in \mathbb{R}^d$ such that

$$\phi(s,a)^\top \theta_f = (\mathcal{P}^\pi f)(s,a) = \mathbb{E}_{s' \sim P(\cdot|s,a)}[f(s', \pi)].$$

To prove the stated claim we will utilize choice instantiations of such functions and their corresponding parameters. For $i \in [d]$, define the function $f_i := \langle \phi, \mathbf{e}_i \rangle \in \mathcal{F}_\phi$, and let $\theta_i \in \mathbb{R}^d$ be such that

$$\phi(s,a)^\top \theta_i = \mathbb{E}_{s' \sim P(\cdot|s,a)}[f_i(s', \pi)], \ \forall(s,a).$$

Then for all $(s,a)$,

$$\mathbb{E}_{s' \sim P(\cdot|s,a)}[\phi(s', \pi)] = \begin{bmatrix} \mathbb{E}_{s' \sim P(\cdot|s,a)}[f_1(s', \pi)] \\ \mathbb{E}_{s' \sim P(\cdot|s,a)}[f_2(s', \pi)] \\ \vdots \\ \mathbb{E}_{s' \sim P(\cdot|s,a)}[f_d(s', \pi)] \end{bmatrix} = \underbrace{\begin{bmatrix} - & \theta_1^\top & - \\ - & \theta_2^\top & - \\ & \vdots & \\ - & \theta_d^\top & - \end{bmatrix}}_{(*)} \phi(s,a).$$

Lastly, we will show that the above system of equations is satisfied by setting

$$(*) = \Sigma_{\mathrm{cr}}^{-\top} \Sigma^{-1} = (B^\pi)^\top.$$

Right-multiplying both sides by $\phi(s,a)^\top$ then taking the expectation over $(s,a) \sim \mu^D$, we obtain

$$\mathbb{E}_{(s,a,s',a') \sim \mu^D \times P \times \pi}\left[\phi(s',a')\phi(s,a)^\top\right] = (B^\pi)^\top \mathbb{E}_{(s,a) \sim \mu^D}\left[\phi(s,a)\phi(s,a)^\top\right].$$

Solving for $(B^\pi)^\top$ and rearranging gives

$$\begin{aligned} (B^\pi)^\top &= \left(\mathbb{E}_{(s,a,s',a') \sim \mu^D \times P \times \pi}\left[\phi(s,a)\phi(s',a')^\top\right]\right)^\top \Sigma^{-1} \\ &= \Sigma_{\mathrm{cr}}^\top \Sigma^{-1}, \end{aligned}$$

which confirms that $B^\pi = \Sigma^{-1}\Sigma_{\mathrm{cr}}$ satisfies for all $(s,a)$ the equivalence

$$\mathbb{E}_{s' \sim P(\cdot|s,a)}[\phi(s', \pi)] = (B^\pi)^\top \phi(s,a).$$

**(2) Showing $\mu_\phi^\pi = \phi^\pi$.** Recall that $\phi^\pi = \mathbb{E}_{(s,a)\sim\mu^\pi}[\phi(s,a)]$. Using the Bellman flow equations for $\mu^\pi$, we obtain a recursive system of equations for the dynamics of $\phi^\pi$:

$$\phi^\pi = \sum_{s,a} \phi(s,a)\mu^\pi(s,a)$$

$$= \sum_{s,a} \phi(s,a)\left((1-\gamma)\mu_0^\pi(s,a) + \gamma\sum_{s',a'}P^\pi(s,a\mid s',a')\mu^\pi(s',a')\right)$$

$$= (1-\gamma)\phi_0 + \gamma\,\mathbb{E}_{(s,a)\sim\mu^\pi}\left[\mathbb{E}_{s'\sim P(\cdot|s,a)}[\phi(s',\pi)]\right]$$

$$= (1-\gamma)\phi_0 + \gamma\,\mathbb{E}_{(s,a)\sim\mu^\pi}\left[(B^\pi)^\top\phi(s,a)\right]$$

$$= (1-\gamma)\phi_0 + \gamma(B^\pi)^\top\phi^\pi,$$

where we invoke the result from **(1)** in the second-to-last line. Repeatedly expanding the RHS of the equation with the recursion,

$$\phi^\pi = (1-\gamma)\phi_0 + \gamma(B^\pi)^\top\phi^\pi,$$

$$= (1-\gamma)\phi_0 + \gamma(B^\pi)^\top\left((1-\gamma)\phi_0 + \gamma(B^\pi)^\top\phi^\pi\right)$$

$$= (1-\gamma)\left(\phi_0 + \gamma(B^\pi)^\top\phi_0 + \gamma^2\left((B^\pi)^\top\right)^2\phi^\pi\right)$$

$$\cdots$$

$$= (1-\gamma)\sum_{t=0}^{\infty}\gamma^t\left((B^\pi)^\top\right)^t\phi_0,$$

which is exactly the definition of $\mu_\phi^\pi$ from Proposition 1.

**(3) Showing $\rho(B^\pi) \leq 1$.** The proof of (2) implies that for any $(s_0,a_0) \in \mathcal{S}\times\mathcal{A}$, $((B^\pi)^\top)^t\phi(s_0,a_0) = \mathbb{E}_{(s,a)\sim\mu_t^{\pi,s_0,a_0}}[\phi(s,a)]$, where $\mu_t^{\pi,s_0,a_0}$ is the $t$-th step state-action distribution under $\pi$ when the initial state-action pair is the given $(s_0,a_0)$. Given $\|\phi(s,a)\| \leq B_\phi, \forall(s,a)$, we have

$$((B^\pi)^\top)^t\phi(s_0,a_0) \leq B_\phi, \forall t.$$

Given that $\Sigma$ is full-rank, we can always find $\{(s_0^{(i)},a_0^{(i)})\}_{i=1}^d$ such that $\{u_i := \phi(s_0^{(i)},a_0^{(i)})\}_{i=1}^d$ forms a basis of $\mathbb{R}^d$. Then we have $\|((B^\pi)^\top)^t u_i\| \leq B_\phi, \forall t$.

Now we show that $\|((B^\pi)^\top)^t\|_{\mathrm{op}}$, where $\|\cdot\|_{\mathrm{op}}$ is the operator norm, also has a finite bound that is independent of $t$. Recall that operator norm is the largest singular value; let the corresponding singular vector be $u$, and we express $u = \sum_{i=1}^d \alpha_i u_i$. We have

$$\|((B^\pi)^\top)^t\|_{\mathrm{op}} = \|((B^\pi)^\top)^t u\| = \|\sum_{i=1}^d \alpha_i((B^\pi)^\top)^t u_i\| \leq \sum_{i=1}^d |\alpha_i| B_\phi =: v.$$

The key here is that the upper bound $v < \infty$ is independent of $t$. Plugging into the Gelfand's formula, we have

$$\rho(B^\pi) = \rho((B^\pi)^\top) = \lim_{t\to\infty}\|((B^\pi)^\top)^t\|_{\mathrm{op}}^{1/t} \leq \lim_{t\to\infty} v^{1/t} = 1.$$

**(4) Proving equivalence $C_\phi^\pi = C_{\mathrm{lin}}^\pi$.** Recalling Eq. (14) and the definition of $\mu_\phi^\pi$, following the proof of Proposition 1, when $\sigma_{\max}(B^\pi) < 1/\gamma$ we may write

$$C_\phi^\pi = (1-\gamma)^2\phi_0^\top A^{-1}\Sigma A^{-\top}\phi_0$$

$$= (1-\gamma)^2\phi_0^\top(I-\gamma B^\pi)^{-1}\Sigma^{-1}(I-\gamma B^\pi)^{-\top}\phi_0.$$

$$= \left(\mu_\phi^\pi\right)^\top\Sigma^{-1}\mu_\phi^\pi.$$

Substituting the previously derived identity that $\mu_\phi^\pi = \phi^\pi$ in the last line,

$$\left(\mu_\phi^\pi\right)^\top \Sigma^{-1} \mu_\phi^\pi = (\phi^\pi)^\top \Sigma^{-1} \phi^\pi = C_{\text{lin}}^\pi.$$

$\square$

## C   FUNCTION ESTIMATION GUARANTEES

For most of the paper we have focused on providing return estimation guarantees, i.e., error bounds for estimating $J(\pi)$. In some scenarios, it is desirable to obtain stronger *function estimation* guarantees (Huang & Jiang, 2022; Perdomo et al., 2023), that $\widehat{Q}_{\text{lstd}}$ and $Q^\pi$ are close as functions, typically measured by weighted 2-norm. Indeed, our proof of Theorem 1 can be easily adapted to provide the following guarantee:

**Theorem 2** (Function Estimation). *Under the same assumptions as Theorem 1, w.p. $\geq 1 - \delta$, for any $\nu \in \Delta(\mathcal{S} \times \mathcal{A})$,*

$$\sqrt{\mathbb{E}_{(s,a)\sim\nu}[(Q^\pi(s,a) - \hat{Q}_{\text{lstd}}(s,a))^2]} \lesssim \frac{V_{\max}}{1-\gamma}\sqrt{\frac{\widehat{C}_{\text{fn}}^\pi \cdot d\log(1/\delta)}{n}},$$

*where $\widehat{C}_{\text{fn}}^\pi := (1-\gamma)^2 \, \mathbb{E}_{(s_0,a_0)\sim\nu}\left[\|\widehat{\Sigma}^{1/2}\widehat{A}^{-\top}\phi(s_0,a_0)\|_2^2\right].$*

When $\nu = \mu_0 \circ \pi$ is a point-mass, the LHS of Theorem 2 coincides with that of Theorem 1, and the guarantees on the RHS are identical, too. Also recall that the naïve analysis based on $1/\sigma_{\min}(A)$ (Section 1) provides parameter identification (i.e., bounded $\|\widehat{\theta}_{\text{lstd}} - \theta^\star\|$), which immediately provides $\ell_\infty$ function-estimation guarantee. This result is directly implied by our Theorem 2, where the coverage parameter can be bounded as a function of $\sigma_{\min}(A)$ and $B_\phi$.

**Remark on $C_{\text{fn}}^\pi$.**   Similar to Corollary 1 we can induce a corollary that depends on the population version of $\widehat{C}_{\text{fn}}^\pi$, which we denote as $C_{\text{fn}}^\pi$. It is interesting to compare it to standard coverage parameters that enable function-estimation guarantees under completeness (Section 3). Note that the term inside $C_{\text{fn}}^\pi = \mathbb{E}_{(s_0,a_0)\sim\nu}[\cdot]$ is simply $C_\phi^\pi$ but for a deterministic initial state-action pair $(s_0, a_0)$. Applying Proposition 4, we have

$$C_{\text{fn}}^\pi = \mathbb{E}_{(s_0,a_0)\sim\nu}\left[\left(\phi_{s_0,a_0}^\pi\right)^\top \Sigma^{-1} \phi_{s_0,a_0}^\pi\right],$$

where $\phi_{s_0,a_0}^\pi = \mathbb{E}_{(s,a)\sim\mu_{s_0,a_0}^\pi}[\phi(s,a)]$ is the expected feature under the occupancy induced from deterministic $s_0, a_0$ as the initial state-action pair. In comparison, the standard coverage in the literature is

$$C_{\text{lin,fn}}^\pi = \mathbb{E}_{(s,a)\sim\mu^\pi}[\phi(s,a)^\top \Sigma^{-1} \phi(s,a)].$$

As can be seen, our $C_{\text{fn}}^\pi$ is in between $C_\phi^\pi$ and $C_{\text{lin,fn}}^\pi$, since we partially marginalize out the portion of $\mu^\pi$ that can be attribute to each initial state-action pair, instead of measuring every single $(s,a) \sim \mu^\pi$ under $\Sigma^{-1}$ in a completely point-wise manner.

**Proof of Theorem 2.**   We repeat a similar derivation to Eq. (21), noting that the proof holds when the initial state-action distribution $s_0 \sim \mu_0, a_0 \sim \pi$ changes to an arbitrary distribution $\nu$.

$$\mathbb{E}_{(s_0,a_0)\sim\nu}\left[\left(Q^\pi(s_0,a_0) - \hat{Q}_{\text{lstd}}(s_0,a_0)\right)^2\right]$$

$$= \mathbb{E}_{(s_0,a_0)\sim\nu}\left[\left(\phi(s_0,a_0)^\top\left(\theta^\star - \widehat{\theta}_{\text{lstd}}\right)\right)^2\right]$$

$$= \mathbb{E}_{(s_0,a_0)\sim\nu}\left[\left(\phi(s_0,a_0)^\top \hat{A}^{-1}\widehat{\Sigma}^{1/2}\widehat{\Sigma}^{-1/2}\hat{A}\left(\theta^\star - \widehat{\theta}_{\text{lstd}}\right)\right)^2\right]$$

$$\leq \mathbb{E}_{(s_0,a_0)\sim\nu}\left[\left\|\widehat{\Sigma}^{1/2}\hat{A}^{-T}\phi(s_0,a_0)\right\|_2^2\left\|\widehat{\Sigma}^{-1/2}\hat{A}\left(\theta^\star - \widehat{\theta}_{\text{lstd}}\right)\right\|_2^2\right]$$

As in the proof of Theorem 1, we note that

$$\left\|\widehat{\Sigma}^{-1/2}\hat{A}\left(\theta^\star - \widehat{\theta}_{\mathrm{lstd}}\right)\right\|_2 \le \left\|\widehat{\Sigma}^{-1/2}(\hat{A}\theta^\star - \hat{b})\right\|_2 + \left\|\widehat{\Sigma}^{-1/2}(\hat{A}\widehat{\theta}_{\mathrm{lstd}} - \hat{b})\right\|_2$$
$$\le \left\|\widehat{\Sigma}^{-1/2}(\hat{A}\theta^\star - \hat{b})\right\|_2,$$

since $\hat{A}\widehat{\theta}_{\mathrm{lstd}} - \hat{b} = 0$ and $\widehat{\Sigma}$ is invertible. To conclude, we recall that the concentration bound from Lemma 1, which implies that

$$\|\widehat{\Sigma}^{-1/2}(\hat{A}\theta^\star - \hat{b})\|_2^2 = \mathcal{O}\left(V_{\max}^2 \frac{d\log(1/\delta)}{n}\right).$$

Plugging this in yields the proof. $\qquad\square$

## D  LOSS MINIMIZATION ALGORITHM

Here we provide an alternative analysis to Corollary 1, where we are able to eliminate the dependence on $1/\sigma_{\min}(A)$, but the rates still depend on $1/\sigma_{\min}(\Sigma)$. The analysis also requires a slight change of the LSTDQ algorithm to a loss-minimization form (Liu et al., 2025):

$$\widehat{\theta}_{\mathrm{lstd}} = \arg\min_{\theta\in\Theta}\|\widehat{\Sigma}^{-1/2}(\widehat{A}\theta - \widehat{b})\|_2. \tag{20}$$

In practice, when $\widehat{A}$ is near-singular, the inverse solution $\widehat{A}^{-1}\widehat{b}$ may have a very large norm which is clearly problematic, demanding some regularization to control the norm of the solution. The loss-minimization formulation of Eq. (20) is a natural abstraction of this process, where we search for $\widehat{\theta}$ in a pre-defined parameter space with bounded norm. If $\widehat{A}^{-1}\widehat{b} \in \Theta$, it is easy to see that the loss-minimization solution coincides with the inverse solution; when $\widehat{A}^{-1}\widehat{b} \notin \Theta$, Eq. (20) still outputs a bounded solution to ensure generalization and good statistical properties.

We will need the following boundedness assumption on $\Theta$.

**Assumption 4** (Boundedness of $\Theta$). *Assume* $\|\theta\|_2 \le B_\Theta, \ \forall\,\theta \in \Theta$.

**Additional linear algebraic notation.**  For symmetric $\Sigma$, let $\kappa(\Sigma) = \frac{\lambda_{\max}(\Sigma)}{\lambda_{\min}(\Sigma)}$ be the condition number, where $\lambda_{\max}(\cdot)$ is the largest eigenvalue. Let $\mathrm{tr}(A)$ be the trace of a matrix $A$.

**Theorem 3.** *Assume that* $n \gtrsim \log(d/\delta)\kappa(\Sigma)B_\phi^2/\lambda_{\min}(\Sigma)$. *Let* $\phi_0 = \mathbb{E}_{s_0\sim\mu_0,a_0\sim\pi}[\phi(s_0,a_0)]$ *denote the initial feature. Under Assumptions 1, 2 and 4, the estimator in Eq. (20) satisfies that*

$$\left|J(\pi) - J_{\widehat{Q}_{\mathrm{lstd}}}(\pi)\right| \lesssim \frac{\sqrt{C_\phi^\pi}}{1-\gamma}\max\{B_\phi B_\Theta, R_{\max}\}^2\kappa(\Sigma)\sqrt{\frac{d\log(B_\Theta n\delta^{-1})}{\lambda_{\min}(\Sigma)n}}$$

*with probability at least* $1 - \delta$.

**Proof of Theorem 3.**  Let $\hat{\ell}(\theta)$ and $\ell(\theta)$ denote the empirical and population vectors:

$$\hat{\ell}(\theta) = \hat{A}\theta - \hat{b} \quad\text{and}\quad \ell(\theta) = A\theta - b.$$

Recall that $A\theta^\star = b$ and thus $\ell(\theta^\star) = 0$. We establish in the sequel the following concentration lemma.

**Lemma 5.** *With probability at least* $1 - \delta$, *we have that for all* $\theta \in \Theta$:

$$\left|\|\Sigma^{-1/2}\ell(\theta)\|_2 - \|\Sigma^{-1/2}\hat{\ell}(\theta)\|_2\right| \le \max\{B_\phi B_\Theta, R_{\max}\}^2\sqrt{\frac{288d\log(864B_\Theta n\delta^{-1})}{\lambda_{\min}(\Sigma)n}} := \varepsilon_{\mathrm{stat}}.$$

We also note the following simple technical lemma.

**Lemma 6.** *For all $v \in \mathbb{R}^d$, we have*

$$v^\top \Sigma^{-1} v \leq \frac{\lambda_{\max}(\Sigma^{-1})}{\lambda_{\min}(\widehat{\Sigma}^{-1})} v^\top \widehat{\Sigma}^{-1} v, \quad and \quad v^\top \widehat{\Sigma}^{-1} v \leq \frac{\lambda_{\max}(\widehat{\Sigma}^{-1})}{\lambda_{\min}(\Sigma^{-1})} v^\top \Sigma^{-1} v.$$

Recall that $\widehat{\theta}$ satisfies $\arg\min_{\theta \in \Theta} \|\widehat{\Sigma}^{-1/2}\hat{\ell}(\theta)\|_2$. We now show that Lemma 5 and Lemma 6 imply that $\|\Sigma^{-1/2}\ell(\widehat{\theta})\|_2$ is small. This follows since:

$$\|\Sigma^{-1/2}\ell(\widehat{\theta})\|_2 \leq \|\Sigma^{-1/2}\hat{\ell}(\widehat{\theta})\|_2 + \varepsilon_{\text{stat}}$$

$$\leq \sqrt{\frac{\lambda_{\max}(\Sigma^{-1})}{\lambda_{\min}(\widehat{\Sigma}^{-1})}} \|\widehat{\Sigma}^{-1/2}\hat{\ell}(\widehat{\theta})\|_2 + \varepsilon_{\text{stat}}$$

$$\leq \sqrt{\frac{\lambda_{\max}(\Sigma^{-1})}{\lambda_{\min}(\widehat{\Sigma}^{-1})}} \|\widehat{\Sigma}^{-1/2}\hat{\ell}(\theta^\star)\|_2 + \varepsilon_{\text{stat}}$$

$$\leq \sqrt{\frac{\lambda_{\max}(\Sigma^{-1})}{\lambda_{\min}(\widehat{\Sigma}^{-1})} \cdot \frac{\lambda_{\max}(\widehat{\Sigma}^{-1})}{\lambda_{\min}(\Sigma^{-1})}} \|\Sigma^{-1/2}\hat{\ell}(\theta^\star)\|_2 + \varepsilon_{\text{stat}}$$

$$\leq \sqrt{\frac{\lambda_{\max}(\Sigma^{-1})}{\lambda_{\min}(\widehat{\Sigma}^{-1})} \cdot \frac{\lambda_{\max}(\widehat{\Sigma}^{-1})}{\lambda_{\min}(\Sigma^{-1})}} \|\Sigma^{-1/2}\ell(\theta^\star)\|_2 + \left(1 + \sqrt{\frac{\lambda_{\max}(\Sigma^{-1})}{\lambda_{\min}(\widehat{\Sigma}^{-1})} \cdot \frac{\lambda_{\max}(\widehat{\Sigma}^{-1})}{\lambda_{\min}(\Sigma^{-1})}}\right) \varepsilon_{\text{stat}}$$

$$= \left(1 + \sqrt{\kappa(\Sigma)\kappa(\widehat{\Sigma})}\right) \varepsilon_{\text{stat}}$$

$$\leq 2\sqrt{\kappa(\Sigma)\kappa(\widehat{\Sigma})} \varepsilon_{\text{stat}}.$$

In the sequel, we also show concentration for the condition number of $\widehat{\Sigma}$ to $\Sigma$.

**Lemma 7.** *Let $n \geq 32\log(6d/\delta)\kappa(\Sigma)\left(B_\phi^2/\lambda_{\min}(\Sigma) + \kappa(\Sigma)\right)$. Then, with probability at least $1 - \delta$, we have:*

$$\kappa(\widehat{\Sigma}) \leq 3\kappa(\Sigma)$$

This implies that, under the condition on sample size, we have $\left\|\Sigma^{-1/2}\ell(\widehat{\theta})\right\|_2 \leq \sqrt{12}\kappa(\Sigma)\varepsilon_{\text{stat}}$ with high-probability. We can now conclude the proof. Under the conditions and events stated above, we have:

$$\left|\mathbb{E}_{s_0 \sim \mu_0, a_0 \sim \pi}\left[Q^\pi(s_0, a_0) - \hat{Q}_{\text{lstd}}(s_0, a_0)\right]\right| = \left|\mathbb{E}_{s_0 \sim \mu_0, a_0 \sim \pi}\left[\phi(s_0, a_0)^\top\left(\theta^\star - \widehat{\theta}_{\text{lstd}}\right)\right]\right| \quad (21)$$

$$= \left|\phi_0^\top\left(\theta^\star - \widehat{\theta}_{\text{lstd}}\right)\right|$$

$$= \left|\phi_0^\top\left(A^{-1}b - \widehat{\theta}_{\text{lstd}}\right)\right|$$

$$= \left|\phi_0^\top A^{-1}\left(b - A\widehat{\theta}_{\text{lstd}}\right)\right|$$

$$= \left|\phi_0^\top A^{-1}\Sigma^{1/2}\Sigma^{-1/2}\left(b - A\widehat{\theta}_{\text{lstd}}\right)\right|$$

$$= \left|\phi_0^\top A^{-1}\Sigma^{1/2}\Sigma^{-1/2}\left(b - A\widehat{\theta}_{\text{lstd}}\right)\right|$$

$$\leq \left\|\Sigma^{1/2}A^{-T}\phi_0\right\|_2 \left\|\Sigma^{-1/2}\left(A\widehat{\theta}_{\text{lstd}} - b\right)\right\|_2$$

$$= \left\|\Sigma^{1/2}A^{-T}\phi_0\right\|_2 \left\|\Sigma^{-1/2}\left(A\widehat{\theta}_{\text{lstd}} - b\right)\right\|_2$$

$$= \sqrt{\phi_0^\top A^{-1}\Sigma A^{-T}\phi_0} \|\Sigma^{-1/2}\ell(\widehat{\theta})\|_2$$

$$\leq \frac{1}{1-\gamma}\sqrt{C_\phi^\pi}\sqrt{12}\kappa(\Sigma)\varepsilon_{\text{stat}}, \quad (22)$$

as desired. We now establish Lemmas 5 to 7.

$\square$

**Proof of Lemma 5.** Let $\theta$ be fixed for now, and $\Delta(\theta) = \hat{\ell}(\theta) - \ell(\theta)$. Note that by the reverse triangle inequality,

$$\left| \|\Sigma^{-1/2}\ell(\theta)\|_2 - \|\Sigma^{-1/2}\hat{\ell}(\theta)\|_2 \right| \le \left\| \Sigma^{-1/2}\Delta(\theta) \right\|_2 = \left\| \Sigma^{-1/2}(\hat{A} - A)\theta - \Sigma^{-1/2}(\hat{b} - b) \right\|_2.$$

We use Vector Bernstein (Lemma 9) to show that this is small. Let $X_i = \phi(s_i, a_i)$, $Y_i = \phi(s_i, a_i) - \gamma\phi(s_i', a_i')$, and $\Delta_i(\theta) = X_i(Y_i^\top\theta - r_i) - (A\theta - b)$ denote the centered vectors. Note that

$$\|\Sigma^{-1/2}\Delta_i(\theta)\|_2 \le \frac{1}{\sqrt{\lambda_{\min}(\Sigma)}} 2\max\{\|X_i(Y_i^\top\theta - r_i)\|_2, \|A\theta - b\|_2\}.$$

We have the following bound:

$$\|A\theta - b\|_2 = \left\| \mathbb{E}\left[ \phi(s, a)\left( (\phi(s, a) - \gamma\phi(s', a'))^\top\theta - r(s, a) \right) \right] \right\|_2$$

$$\le \left\| \mathbb{E}\left[ \phi(s, a)\phi(s, a)^\top\theta \right] \right\|_2 + \gamma\left\| \mathbb{E}\left[ \phi(s, a)\phi(s', a')^\top\theta \right] \right\|_2 + \|\mathbb{E}[\phi(s, a)r(s, a)]\|_2$$

$$\le (1 + \gamma)\max_{s,a}\|\phi(s, a)\|_2^2\|\theta\|_2 + \max_{s,a}\|\phi(s, a)\|_2 R_{\max}$$

$$\le 3B_\phi\max\{B_\phi B_\Theta, R_{\max}\}. \tag{23}$$

We remark that with a similar derivation, this bound applies just as well to $\|X_i(Y_i^\top\theta - r_i)\|_2$, so in fact we have

$$\|\Sigma^{-1/2}\Delta_i(\theta)\|_2 \le \frac{6}{\sqrt{\lambda_{\min}(\Sigma)}} B_\phi\max\{B_\phi B_\Theta, R_{\max}\}.$$

For the variance bound, we simply use that

$$\mathbb{E}\left[ \|\Sigma^{-1/2}\Delta_i(\theta)\|_2^2 \right] \le \left( \frac{6}{\sqrt{\lambda_{\min}(\Sigma)}} B_\phi\max\{B_\phi B_\Theta, R_{\max}\} \right)^2.$$

Then, we conclude via Lemma 9 that

$$\|\Sigma^{-1/2}\Delta(\theta)\|_2 \le B_\phi\max\{B_\phi B_\Theta, R_{\max}\}\sqrt{\frac{32\log(288\delta^{-1})}{\lambda_{\min}(\Sigma)n}}.$$

We now apply a covering argument over $\theta \in \Theta$. Let $\Theta_0 \subseteq \Theta$ be an $L_2$-covering of $\Theta$ of size $\mathcal{N}(\varepsilon)$, satisfying for for each $\theta \in \Theta$ there exists a covering member $\rho(\theta) \in \Theta_0$ satisfying $\|\theta - \rho(\theta)\|_2 \le \varepsilon$. Via a simple triangle inequality:

$$\left\| \Sigma^{-1/2}\Delta(\theta) \right\|_2 \le \left\| \Sigma^{-1/2}\Delta(\rho(\theta)) \right\|_2 + \left\| \Sigma^{-1/2}(\Delta(\theta) - \Delta(\rho(\theta))) \right\|_2.$$

We bound the latter term as a function of $\varepsilon$.

$$\left\| \Sigma^{-1/2}(\Delta(\theta) - \Delta(\rho(\theta))) \right\|_2 = \left\| \Sigma^{-1/2}\left( A - \hat{A} \right)(\theta - \rho(\theta)) \right\|_2$$

$$\le \frac{1}{\sqrt{\lambda_{\min}(\Sigma)}} 2\max\left\{ \sigma_{\max}(A), \sigma_{\max}(\hat{A}) \right\}\varepsilon.$$

We notice that $\max\left\{ \sigma_{\max}(A), \sigma_{\max}(\hat{A}) \right\} \le 2B_\phi^2$ via a similar reasoning to Eq. (23). This leaves us with:

$$\left\| \Sigma^{-1/2}\Delta(\theta) \right\|_2 \le B_\phi\max\{B_\phi B_\Theta, R_{\max}\}\sqrt{\frac{32\log(288|\Theta_0|\delta^{-1})}{\lambda_{\min}(\Sigma)n}} + \frac{2B_\phi^2}{\sqrt{\lambda_{\min}(\Sigma)}}\varepsilon,$$

$$\le B_\phi\max\{B_\phi B_\Theta, R_{\max}\}\sqrt{\frac{32d\log(864\delta^{-1}/\varepsilon)}{\lambda_{\min}(\Sigma)n}} + \frac{2B_\phi^2}{\sqrt{\lambda_{\min}(\Sigma)}}\varepsilon,$$

where we have applied a union bound over the set $\Theta_0$, which is of size at most $(3B_\Theta/\varepsilon)^d$ for $\varepsilon \in (0, 1]$ by standard covering number bounds (Vershynin, 2018), since $\Theta \subset \{\theta \in \mathbb{R}^d : \|\theta\|_2 \leq B_\Theta\}$. Picking $\varepsilon = 1/\sqrt{n}$ lets us conclude that, with probability at least $1 - \delta$, for all $\theta \in \Theta$,

$$\left\|\Sigma^{-1/2}\Delta(\theta)\right\|_2 \leq B_\phi \max\{B_\phi B_\Theta, R_{\max}\}\sqrt{\frac{288d\log(864B_\Theta n\delta^{-1})}{\lambda_{\min}(\Sigma)n}},$$

as desired. $\qquad\square$

**Proof of Lemma 6.** Follows from the fact that for any positive semi-definite matrix $M \in \mathbb{R}^{d\times d}$ and for any $v \in \mathbb{R}^d$, we have the inequalities

$$\lambda_{\min}(M)v^\top v \leq v^\top M v \leq \lambda_{\max}(M)v^\top v.$$

$\qquad\square$

**Proof of Lemma 7.** We firstly establish that

$$\|\widehat{\Sigma} - \Sigma\|_2 \leq \sqrt{\frac{8\lambda_{\max}(\Sigma)(B_\phi^2 + \lambda_{\max}(\Sigma))\log(6d/\delta)}{n}} =: \varepsilon_{\mathrm{op}}. \tag{24}$$

To do this, we use Matrix Bernstein (Lemma 8). Abbreviate $X_i := \phi(s_i, a_i)$, and let $Z_i = X_iX_i^\top - \Sigma$ be the centered matrices. Note that $\|Z_i\|_2 \leq \|X_iX_i^\top\|_2 + \|\Sigma\|_2 \leq \|X_i\|_2^2 + \lambda_{\max}(\Sigma) \leq B_\phi^2 + \lambda_{\max}(\Sigma)$. For the variance term, we have:

$$\begin{aligned}
\left\|\mathbb{E}\left[(X_iX_i^\top - \Sigma)^2\right]\right\|_2 &= \left\|\mathbb{E}\left[(X_iX_i^\top)^2\right] - \Sigma^2\right\|_2 \\
&\leq \left\|B_\phi^2\,\mathbb{E}\left[X_iX_i^\top\right] - \Sigma^2\right\|_2 \\
&\leq B_\phi^2\lambda_{\max}(\Sigma) + \lambda_{\max}(\Sigma)^2.
\end{aligned}$$

This yields

$$\|\hat{\Sigma} - \Sigma\|_2 \leq \sqrt{\frac{2\lambda_{\max}(\Sigma)(B_\phi^2 + \lambda_{\max}(\Sigma))\log(2d/\delta)}{n}} + \frac{2(B_\phi^2 + \lambda_{\max}(\Sigma))\log(2d/\delta)}{3n}$$

The slow term dominates when $n$ is large enough:

$$n \geq \frac{2(B_\phi^2 + \lambda_{\max}(\Sigma))\log(2d/\delta)}{\lambda_{\max}(\Sigma)} = 2\log(2d/\delta)\left(\frac{B_\phi^2}{\lambda_{\max}(\Sigma)} + 1\right). \tag{25}$$

Note that this is implied by our assumption on $n$, since $\lambda_{\max}(\Sigma) \geq \lambda_{\min}(\Sigma)$ and $\kappa(\Sigma) \geq 1$. Thus, under this condition we have

$$\|\hat{\Sigma} - \Sigma\|_2 \leq 2\sqrt{\frac{2\lambda_{\max}(\Sigma)(B_\phi^2 + \lambda_{\max}(\Sigma))\log(2d/\delta)}{n}} = \varepsilon_{\mathrm{op}},$$

as desired. Now, by Weyl's theorem (Horn & Johnson, 2012, Theorem 4.3.1), we have

$$|\lambda_{\min}(\hat{\Sigma}) - \lambda_{\min}(\Sigma)| \leq \|\widehat{\Sigma} - \Sigma\|_2 \leq \varepsilon_{\mathrm{op}},$$

which implies that

$$\lambda_{\min}(\hat{\Sigma}) \geq \frac{\lambda_{\min}(\Sigma)}{2} \tag{26}$$

using the condition that $\varepsilon_{\mathrm{op}} \leq \frac{\lambda_{\min}(\Sigma)}{2}$. This latter condition is equivalent to

$$\sqrt{\frac{8\lambda_{\max}(\Sigma)(B_\phi^2 + \lambda_{\max}(\Sigma))\log(2d/\delta)}{n}} \leq \frac{\lambda_{\min}(\Sigma)}{2}$$

$$\iff n \geq 32\log(2d/\delta)\kappa(\Sigma)\left(\frac{B_\phi^2}{\lambda_{\min}(\Sigma)} + \kappa(\Sigma)\right), \tag{27}$$

which is precisely our assumption on $n$. Similarly, an application of the reverse triangle inequality (or of Weyl's theorem again) yields,

$$|\lambda_{\max}(\hat{\Sigma}) - \lambda_{\max}(\Sigma)| \leq \|\hat{\Sigma} - \Sigma\|_2 \leq \varepsilon_{\mathrm{op}},$$

which implies that

$$\lambda_{\max}(\hat{\Sigma}) \leq \lambda_{\max}(\Sigma) + \varepsilon_{\mathrm{op}} \leq \frac{3}{2}\lambda_{\max}(\Sigma), \tag{28}$$

using the condition that $\varepsilon_{\mathrm{op}} \leq \frac{\lambda_{\min}(\Sigma)}{2} \leq \frac{\lambda_{\max}(\Sigma)}{2}$. Combining Eqs. (28) and (26), we have:

$$\kappa(\hat{\Sigma}) = \frac{\lambda_{\max}(\hat{\Sigma})}{\lambda_{\min}(\hat{\Sigma})} \leq \frac{3}{2}\frac{\lambda_{\max}(\Sigma)}{\lambda_{\min}(\hat{\Sigma})} \leq 3\frac{\lambda_{\max}(\Sigma)}{\lambda_{\min}(\Sigma)} = 3\kappa(\Sigma).$$

$\square$

# E    TECHNICAL TOOLS

**Lemma 8** (Matrix Bernstein, Tropp (2012)). *Let $S_1, \ldots, S_n \in \mathbb{R}^{d_1 \times d_2}$ be random, independent matrices satisfying $\mathbb{E}[S_k] = 0$, $\max\{\|\mathbb{E}[S_k S_k^\top]\|_{\mathrm{op}}, \|\mathbb{E}[S_k^\top S_k]_{\mathrm{op}}\|\} \leq \sigma^2$, and $\|S_k\|_{\mathrm{op}} \leq L$ almost surely for all $k$. Then, with probability at least $1 - \delta$ for any $\delta \in (0, 1)$,*

$$\left\|\frac{1}{n}\sum_{k=1}^{n} S_k\right\|_{\mathrm{op}} \leq \sqrt{\frac{2\sigma^2 \log((d_1 + d_2)/\delta)}{n}} + \frac{2L \log((d_1 + d_2)/\delta)}{3n}$$

**Lemma 9** (Vector Bernstein, Minsker (2017)). *Let $v_1, \ldots, v_n$ be independent vectors in $\mathbb{R}^d$ such that $\mathbb{E}[v_k] = 0$, $\mathbb{E}[\|v_k\|_2^2] \leq \sigma^2$, and $\|v_k\|_2 \leq L$ almost surely for all $k$. Then, with probability at least $1 - \delta$ for any $\delta \in (0, 1)$,*

$$\left\|\frac{1}{n}\sum_{i=1}^{n} v_i\right\|_2 \leq \sqrt{\frac{2\sigma^2 \log(28/\delta)}{n}} + \frac{2L \log(28/\delta)}{3n}.$$

**Lemma 10** (Vector Martingale Bernstein (Pinelis, 1994; Martinez-Taboada & Ramdas, 2024)). *Let $(X_t)_{t \leq T}$ be a martingale sequence of vectors in $\mathbb{R}^d$ adapted to a filtration $(\mathcal{F}_t)_{t \leq T}$, such that $\mathbb{E}_{t-1}[X_t] = 0$, and $\|X_t\|_2 \leq B$, and $\sum_{t=1}^{T} \mathbb{E}_{t-1}\left[\|X_t\|^2\right] \leq \sigma^2$. Then, with probability at least $1 - \delta$, we have:*

$$\left\|\sum_{t=1}^{T} X_t\right\|_2 \leq \sqrt{2\sigma^2 \log(2/\delta)} + \frac{2}{3}B\log(2/\delta).$$