# OpenReview forum: "A Unifying View of Coverage in Linear Off-policy Evaluation"
_ICLR.cc/2026/Conference — ICLR 2026 Poster_

### Official Review · Reviewer_jTim · 2025-10-21

**Soundness:** 3
**Presentation:** 3
**Contribution:** 3
**Rating:** 6
**Confidence:** 3

**Summary:**

This work makes an important contribution, studying offline linear policy evaluation via the linear temporal differences method without assuming Bellman completeness.

Moreover, the current submission shows that the analysis of LSTDQ offers a unification in the linear setting of several previous methods initially proposed for the general function approximation setting.

**Strengths:**

1)The bound in Corollary 1 scales with the population version of the coverage parameter, not only on the stochastic version.

2) The results are sound and interesting.

3) I really appreciated Section 5.3 and Proposition 2, which makes the comparison with the existing literature very clear.

4) I think that also the unification with MWL is an important contribution.

**Weaknesses:**

1) I think that the invertibility assumption on $\Sigma$ and $A$ is quite strong. Could you avoid this assumption adding a small scalar times the identity in the definition of $A$ and $\Sigma$?

2) Same comment holds for the empirical estimates $\hat{\Sigma}$ and $\hat{A}$

3) Corollary 1 seems to guarantee good approximation of the performance of the policy $\pi$ starting from states sampled from the initial distribution.
Can the Q value at states action pairs which are not visited under the initial distribution be estimated reliably with your method ?

4) In proposition 3, in the definition of $C^\pi_\phi$ the indices of $k,a$ do not appear inside the expectation, please fix this typo.

**Questions:**

1) See weaknesses.

2) How would a linear Monte carlo approach perform in this setting ?  Intuitively a Monte carlo method seems to me more suitable than temporal difference methods when the Bellman completeness assumption is not imposed.

---

> ### Author Response · Authors · 2025-11-17
>
> We thank the reviewer for the thoughtful comments and appreciation of our work as “an important contribution”. We respond to the questions below.
>
> ---
>
> **Invertibility of $\Sigma$ and $A$ and the empirical estimates**
>
> Invertibility of $\Sigma$ and $A$ are made to enable a clean analysis, and as far as we know, all prior finite-sample analyses of LSTD have made such assumptions. When $A$ is not invertible (which is implied by $\Sigma$’s non-invertibility), LSTD itself is not a well-defined algorithm even with an infinite amount of data since it explicitly needs to invert $A$. Therefore, if the goal is to analyze the original LSTD algorithm as-is, we simply have to make the invertibility assumption, otherwise LSTD itself becomes degenerate and cannot enjoy any guarantee. Furthermore, several prior works have shown that non-invertibility of $A$ causes fundamental hardness, e.g. Perdomo et al. 2022 and Amortila et al. 2023, in the sense that no algorithm can succeed at linear OPE in the worst-case when $A$ is not invertible. Therefore, the assumption is not an artifact of our analysis, but accurately characterizes the behavior of the algorithm.
>
> The next question is whether we can tweak the LSTD algorithm to handle non-invertible $\Sigma$ (but not $A$). Unlike linear regression, one cannot handle this by simply adding a small amount of ridge regression $\lambda I$ to the matrix being inverted (which is $\Sigma$ for linear regression but $A$ for LSTD), because $A$ is not a symmetric PSD matrix. That said, it may be possible to address non-invertible $\Sigma$ through a (non-trivial and non-standard) loss-minimization variant of LSTD that we explore in Appendix D. One can imagine that by adding $\lambda I$ to the \Sigma matrix in the estimator of Equation (20), we can mitigate this at the cost of introducing bias in the final error bound and obtaining a coverage parameter of $\phi\_0^\top A^{-1}(\Sigma + \lambda I) A^{-\top} \phi\_0$, thus potentially giving up the elegant linear-dynamical system interpretation of Section 5. Since our goal is to answer the key question “what is the right definition of coverage”, we choose to make simplification assumptions to enable elegant and clear definitions and bounds.
>
> For the empirical estimates of these matrices, we explicitly mentioned on Line 157 that we do not separately assume they are invertible (indeed, Assumption 2 is only about the population matrices, not the empirical ones). In corollary 1 where the bound depends on the population version of coverage, the burn-in condition ensures that these empirical matrices are invertible with high probability. In theorem 1 where the empirical coverage term shows up, we take the convention that the coverage term is infinity when the empirical matrices are not invertible; again this is reasonable because the LSTDQ algorithm itself ($\hat\theta = \hat{A}^{-1} \hat{b}$) is ill-defined, so no guarantee can be given whatsoever.
>
> ---
>
> **Can the Q value at states action pairs which are not visited under the initial distribution be estimated reliably with your method ?**
>
> Yes! Note that the LSTDQ algorithm itself **does not really know what the initial distribution $\mu\_0$ is**; it only has a dataset sampled from some arbitrary distribution $\mu^D$. Therefore, suppose you are curious about whether the estimate is accurate for some state-action distribution $\nu$; **you can just set $\mu\_0 = \nu$ and invoke our guarantee to answer the question.** Note that changing $\mu\_0$ will lead to a different value of $C\_\phi^\pi$, so our coverage parameter exactly tells you under which distributions the estimated Q-value is accurate. Furthermore, in addition to expected return estimation guarantees, our Appendix C also provides function-estimation guarantees, where the arbitrariness of the error-measuring distribution $\nu$ is stated more explicitly.

---

> > ### Author Response · Authors · 2025-11-17
> >
> > **(k,a) not appearing inside the expectation of Proposition 3**
> >
> > This is an abbreviation that is consistent with the convention we used earlier on Line 312. For the $\chi^2$ version of density ratio boundedness, it is standard to write $E\_{(s,a) \sim \mu^D}[(\mu^\pi/\mu^D)^2]$ as a shorthand for $E\_{(s,a) \sim \mu^D}[(\mu^\pi(s,a)/\mu^D(s,a))^2]$ in the literature (in fact even $(s,a)$ in the subscript is sometimes dropped). The situation is completely analogous in Proposition 3, as $k$ stands for the abstract state, and $\mu\_\phi^D$ is the data distribution of the abstract state-action pairs. We can certainly add clarification to avoid confusion.
> >
> > ---
> >
> > **How do Monte-Carlo methods work here, since they are suitable when Bellman completeness is not imposed?**
> >
> > Indeed, regressing onto the random return from Monte-Carlo (MC) rollouts does not require Bellman completeness. However, note that standard MC methods are **on-policy** methods (i.e., you need to rollout trajectories using the policy you want to evaluate), where we consider the more challenging **off-policy evaluation** setting. Extending MC to off-policy settings is possible, but it becomes (multi-step) importance sampling (i.e., in off-policy setting, you can regress onto importance weighted Monte-Carlo return and that will not require Bellman completeness), whose variance generally grows exponentially with horizon. In fact, from this point of view, TD methods (or generally, any method that leverages the Bellman equation) are exactly to help avoid such a curse of horizon. Please see the recent survey of Jiang & Xie (2025) for further elaboration on this perspective.
> >
> > Jiang and Xie. Statistical Science. Offline reinforcement learning in large state spaces: Algorithms and guarantees.

---

> > > ### Comment · Reviewer_jTim · 2025-11-20
> > >
> > > Thanks a lot for your clarification!
> > >
> > > I think that these clarifications would help future readers if you can incorporate them in your manuscript:)
> > >
> > > I think that the dependence of the coverage on the initial distribution is an important fact that could be highlighted better.

---

> > > > ### Author Response · Authors · 2025-11-20
> > > >
> > > > Dear reviewer jTim,
> > > >
> > > > Thanks for confirming our rebuttal and we will incorporate these clarifications (especially the dependence on initial distribution) in the final version. We believe we have concretely addressed your listed weaknesses and questions (invertibility, initial distribution, and Monte-Carlo methods), and would appreciate it if you can reconsider your score. If you still have major concerns or questions about the paper, please feel free to let us know and we will be happy to further address them.
> > > >
> > > > best,
> > > >
> > > > Authors

---

### Official Review · Reviewer_Jiag · 2025-10-31

**Soundness:** 3
**Presentation:** 3
**Contribution:** 2
**Rating:** 4
**Confidence:** 3

**Summary:**

The paper presents a finite-sample analysis of linear off-policy evaluation using LSTDQ under the assumption of linear realizability, framed through an instrumental-variable view of the TD equation. Its main technical contribution is the introduction of a scale-invariant "feature-dynamics coverage" term, defined in the feature-compressed MDP induced by the given representation, which leads to a value-error bound with the standard $\sqrt{d/n}$ dependence. A notable aspect is that this coverage quantity is shown to subsume several existing notions: it recovers the usual $\chi^2$ / density-ratio concentrability in the tabular case, the aggregated concentrability used in abstraction-based work, and the classical linear-coverage condition when Bellman-completeness is additionally assumed. In this way, the paper provides a unifying perspective on when LSTDQ can be expected to succeed off-policy, without proposing a new algorithm but clarifying the role and interpretation of coverage in this widely used setting.

**Strengths:**

- The paper introduces the feature-dynamics coverage parameter $C_\phi^\pi$, providing a unified perspective on coverage in linear off-policy evaluation (OPE). Derived from an IV view of the LSTDQ algorithm, $C_\phi^\pi$ quantifies how well features induced by the behavior policy capture the subspace relevant to the target policy. It interprets coverage as occurring within a feature-compressed MDP, linking the environment’s dynamics with the feature representation and offering a scale-invariant, interpretable notion of coverage.
- The analysis establishes a finite-sample bound for LSTDQ under the realizability assumption, avoiding the stronger Bellman-completeness requirement. The off-policy value estimation error

$$|J_{\hat{Q}_{\text{lstd}}}(\pi) - J(\pi)|$$

scales as $O(\sqrt{C^\pi_{\phi}   d \log(1/\delta) / n})$, matching the optimal linear regression rate for $\gamma = 0$. Expressing the bound in terms of $C_\phi^\pi$ highlights how coverage, feature dimension, and sample size interact. The IV-based formulation offers a clear statistical interpretation of LSTDQ as a solution to an errors-in-variables problem, yielding transparent and tight guarantees.
- The proposed framework unifies several prior notions of coverage in reinforcement learning. In the tabular case, $C_\phi^\pi$ reduces to the $\chi^2$-divergence between the target and behavior occupancies; under state abstraction, it coincides with aggregated concentrability; and with Bellman-completeness, it becomes the standard linear coverage. This unified formulation connects previously distinct definitions and provides a coherent understanding of coverage across tabular, abstracted, and linear approximation settings.

**Weaknesses:**

1. The paper focuses on the linear function approximation setting, assuming $Q_\pi(s,a) = \phi(s,a)^\top \theta^\star$. This assumption enables a clean finite-sample analysis of the LSTDQ estimator and the introduction of the coverage parameter in Equation (13). However, the framework relies on the invertibility of $\Sigma$ and $A$ and applies only to the linear regime. Recent work in off-policy evaluation has advanced toward general function approximation via eluder dimension, where representation error and nonlinearity become crucial and closer to real world settings.
2. Theorem 1 presents a finite-sample error bound of the form

$$|J_{Q^{\text{lstd}}}(\pi) - J(\pi)| \lesssim \frac{V_{\max}}{1 - \gamma}   \sqrt{\hat C^{\pi}_{\phi}   \frac{d \log(1/\delta)}{n}}$$

While this expression is elegant, $\frac{V_{\max}}{1-\gamma}$ and the coverage parameter $\hat{C^{\pi}_{\phi}}$

can be extremely large, and the feature dimension $d$ may also be high. In practice, for high-dimensional representations or discount factors $\gamma$ close to $1$ (i.e., long-horizon problems), the required sample size could become prohibitively large. Furthermore, Corollary 1 introduces a burn-in threshold $n_0$, which depends on spectral quantities such as $1/\sigma_{\min}(A)$. This implies that the theoretical guarantee may not hold -- or may severely degrade -- when $n < n_0$.

**Questions:**

Most of my concerns are already detailed in the cons section. I would be happy to raise the score if the authors can address them convincingly. One remaining question: The definition of the feature–dynamics coverage parameter $C_\phi^\pi$ is mathematically elegant but somewhat opaque in intuition. It would be helpful if the authors could provide more explanation or intuition for what this quantity measures—e.g., how it relates to state–action visitation overlap or spectral properties of the induced feature dynamics—and how to interpret large or small values of $C_\phi^\pi$ in practice. Can the authors give an concrete, real-world example for this?

---

> ### Author Response · Authors · 2025-11-17
>
> We thank the reviewer for the thoughtful comments and respond to your concerns below.
>
> ---
>
> **Linear function approximation is limited. Recent work in off-policy evaluation has advanced toward general function approximation with Eluder dimension.**
>
> Indeed there has been abundance in works that develop new OPE algorithms and analyze them in the regime of general function approximation, and we have connected our results to those general algorithms, such as BRM and MWL, in Sections 5.3 and 5.4. However, the goal of our paper is NOT to improve OPE methods so that they work better in practical scenarios, but to fill the gap in **conceptual understanding**: the main contribution is to point out that our fundamental understanding was incomplete and, in some sense even misled (w.r.t. whether error propagates through true dynamics vs compressed dynamics; Section 5.3), in such a basic setting. Therefore, that we only consider the basic linear setting is a feature, not a bug. Given all the works that improve OPE under general function approximation (which we have also worked on), it is particularly important to revisit the basics and make sure we base our development on a solid foundation.
>
> Moreover, you mentioned eluder dimension as a complexity measure for function approximation. According to our understanding of the RL literature, eluder dimension is not quite the right measure for OPE but more suitable for online RL, see e.g., Jin et al. (2021)’s work on Bellman-eluder dimension and earlier work of Wen & van Roy (2013). Eluder dimension enables bounded iteration complexity in online exploration but can be much larger than the standard statistical dimension (e.g., log-covering number). For example, Li et al (2022) showed that for ReLU networks eluder dimensions can be exponentially large. In contrast, typical OPE guarantees, as we mentioned in the paper, only rely on much more standard complexity measures (which are typically more well-behaved) such as the statistical dimension and the coverage parameter. If you have a particular line of work in mind about OPE with eluder dimension, we would appreciate it if you could share a pointer.
>
> ---
>
> **Dependence on $1/(1-\gamma)$, $d$, and $V\_{\max}$. Burn-in condition $n\ge n\_0$**
>
> The dependence on $1/(1-\gamma)$ is extremely ubiquitous and often inevitable in RL guarantees due to error propagation. For example, the analyses for general function approximation we mentioned in Sections 5.3 and 5.4 both have such a horizon factor. In fact, the horizon factor shows up in textbook analyses of approximate dynamic programming (see e.g., the seminal works of Csaba Szepesvari and Remi Munos), and many works demonstrate that such factors are inevitable in worst-case analyses; as a related example, recently Amortila et al. (2023) showed that the misspecification error will blow-up by the horizon factors in various versions of linear OPE.
>
> For dependence on $d$, indeed $d$ becomes prohibitively large in high-dimensional problems. This can be partially handled by introducing sparsity or using random projections (Ghavamzadeh et al. 2010). But again, handling such scenarios is orthogonal to the goal of the paper which is to fill the gap in understanding in the most basic setting.
>
> For $V\_{\max}$, it can be easily improved to the variance of the value function $V^\pi$ w.r.t. the random state transitions, as done in Duan & Wang (2020) and Yin & Wang (2021), and all it takes is to use Bernstein’s inequality in Lemma 1. This way, when the dynamics is close to deterministic, the variance term (which now replaces $V\_{\max}$) will be close to 0. However, we decide not to include this improvement as it makes bounds less readable and is orthogonal to the main goal of the paper.
>
> For burn-in condition $n\ge n\_0$, note that our Theorem 1 itself does not require such a burn-in condition. The corollary requires it to replace the empirical version of the coverage term by the population version. While the detailed form of $n\_0$ might have room of improvement, some burn-in conditions like this are **inevitable** even in the simplest, well-studied setting of **linear regression**; see Hsu et al. (2021); Oliveira (2016); Mourtada (2022) that we cited on Line 285.

---

> > ### Author Response · Authors · 2025-11-17
> >
> > **Invertibility of $\Sigma$ and $A$**
> >
> > These assumptions are made to enable a clean analysis, and as far as we know, all prior finite-sample analyses of LSTD have made such assumptions. When $A$ is not invertible (which is implied by $\Sigma$’s non-invertibility), LSTD itself is not a well-defined algorithm even with an infinite amount of data since it explicitly needs to invert $A$. Therefore, if the goal is to analyze the original LSTD algorithm as-is, we simply have to make the invertibility assumption, otherwise LSTD itself becomes degenerate and cannot enjoy any guarantee. Furthermore, several prior works have shown that non-invertibility of $A$ causes fundamental hardness, e.g. Perdomo et al. 2022 and Amortila et al. 2023, in the sense that no algorithm can succeed at linear OPE in the worst-case when $A$ is not invertible. Therefore, the assumption is not an artifact of our analysis, but accurately characterizes the behavior of the algorithm.
> >
> > The next question is whether we can tweak the LSTD algorithm to handle non-invertible $\Sigma$ (but not $A$).  Unlike linear regression, one cannot handle this by simply adding a small amount of ridge regression $\lambda I$ to the matrix being inverted (which is $\Sigma$ for linear regression but $A$ for LSTD), because $A$ is not a symmetric PSD matrix. That said, it may be possible to address non-invertible $\Sigma$ through a (non-trivial and non-standard) loss-minimization variant of LSTD that we explore in Appendix D. One can imagine that by adding $\lambda I$ to the \Sigma matrix in the estimator of Equation (20), we can mitigate this at the cost of introducing bias in the final error bound and obtaining a coverage parameter of $\phi\_0^\top A^{-1}(\Sigma + \lambda I) A^{-\top} \phi\_0$, thus potentially giving up the elegant linear-dynamical system interpretation of Section 5. Since our goal is to answer the key question “what is the right definition of coverage”, we choose to make simplification assumptions to enable “elegant” and “clear” definitions and bounds, which the reviewer also appreciated.
> >
> > ---
> >
> > **Further intuition for the feature-dynamics coverage? How it relates to state–action visitation overlap or spectral properties of the induced feature dynamics**
> >
> > First, as shown on Line 298, the definition recovers the familiar state-action density-ratio coverage in the tabular setting. Also, under Bellman-completeness, we recover the usual linear coverage ($C\_{\textup{lin}}^\pi$), which can be shown to be always bounded by state-action coverage (see page 9 of Xie & Jiang, 2025). The spectral properties of the induced feature dynamics do not directly determine coverage, but it plays a role in interpreting it: as our Proposition 1 shows, when the spectral radius of $B^\pi$, the induced feature dynamics, is less than $1/\gamma$, our feature dynamics coverage can be interpreted as the familiar linear feature coverage (i.e., analogous to how $C\_{\textup{lin}}^\pi$ is defined); the difference is that instead of looking at the feature evolution in the true MDP, we need to look at it in $B^\pi$ which is a deterministic linear dynamical system.
> >
> > Overall, the newly introduced coverage parameter is an abstract mathematical definition, and we have tried our best to provide intuitions by connecting to well-understood special cases (e.g., tabular and Bellman completeness) and provide interpretations exactly along the lines of the questions you asked. Nevertheless, “interpretable/intuitive” is a very subjective notion, and generally the research community tends to feel less comfortable with newly introduced definitions, and intuition is often built up gradually over subsequent investigation from the community instead of immediately “solved” by the very first paper proposing it.
> >
> > ---
> >
> > **References**
> >
> > Ghavamzadeh et al. NeurIPS 2010. LSTD with Random Projections.
> >
> > Wen & van Roy. NeurIPS 2013. Efficient exploration and value function generalization in deterministic systems.
> >
> > Duan & Wang. ICML 2020. Minimax-optimal Off-Policy Evaluation with Linear Function Approximation.
> >
> > Yin & Wang. NeurIPS 2021. Towards instance-optimal offline reinforcement learning with pessimism.
> >
> > Jin et al. NeurIPS 2021. Bellman eluder dimension: New rich classes of rl problems, and sample-efficient algorithms.
> >
> > Li et al. NeurIPS 2022. Understanding the eluder dimension.
> >
> > Amortila et al. ICML 2023. The optimal approximation factors in misspecified off-policy value function estimation.
> >
> > Perdomo et al. JMLR 2022. A complete characterization of linear estimators for offline policy evaluation
> >
> > Xie & Jiang. STS 2025. Offline Reinforcement Learning in Large State Spaces: Algorithms and Guarantees.
> >
> > Liu et al. 2025: Model Selection for Off-Policy Evaluation: New Algorithms and Experimental Protocol

---

> > > ### Comment · Reviewer_Jiag · 2025-11-17
> > >
> > > Thank the authors for the detailed response to my review and my concerns are addressed. Given the improved clarity of message delivered by this paper, I will raise my score.

---

### Official Review · Reviewer_4tkR · 2025-11-01

**Soundness:** 3
**Presentation:** 3
**Contribution:** 3
**Rating:** 6
**Confidence:** 3

**Summary:**

This paper proposes a novel finite-sample analysis of LSTDQ, based on a new coverage parameter $C_{\phi}^{\pi}$ (known as ``feature-dynamics coverage'') derived from an instrument variable (IV) point of view. It also provides sufficient discussion on how the new coverage parameter relate to the existing parameters by showing a good collection of equivalence results.

**Strengths:**

1. The proofs are checked to be mathematically sound.
2. The perspective of analysis looks new to me.
3. Section 5 is appreciated since it delivers very clear messages on how to make sense of the newly defined parameter, as well as providing a good collection of equivalence results with existing parameters.

**Weaknesses:**

1. The so-called ``IV perspective'' that inspires the new results confuses me a bit.
    * As far as I'm concerned, in a linear model $Y = X^{\top} \theta + \epsilon$, IV is only necessary when $X$ and $\epsilon$ are not independent. Speaking of intuitions, I don't see why it should be the case here.
    * It is also a little confusing to refer to Eq. (7) as the linear regression problem, since linear regression shouldn't come with the $\mathbb{E}$, but rather, with observable individual data points.
    * The IV perspective totally disappears in the proofs in the appendix. It appears to me that the analysis, at most, borrows tools from the IV literature rather than providing new interpretations of the LSTDQ estimator.
2. Empirical/numerical results are missing to support the quality of the analysis.
     * Is it possible to provide some evidence for the advantage of the new coverage parameter? Specifically, is it helpful to include an example where $C_{\phi}^{\pi}$ is much tighter than the existing estimators?

**Questions:**

See weakness above.

---

> ### Author Response · Authors · 2025-11-17
>
> We thank the reviewer for the thoughtful comments and appreciation of our work as “deliver[ing] very clear messages”. We respond to the questions below.
>
> ---
>
> **Is this IV?**
>
> First of all, “IV is necessary when $\epsilon$ and $X$ are not independent” is inaccurate. Think about standard linear regression where noise level changes with $X$: this violates independence but does not require IV at all, and OLS still works. Instead, IV is needed when there is **endogeneity**, $E[\epsilon|X] \ne 0$ (or $E[\epsilon X] \ne 0$).
>
> Second, “error in variable” is a very standard form of problems found in textbooks that need IV. It can be rewritten in the above endogenous form after transformation: start with a standard linear regression model $Y = (X’)^\top \theta + \epsilon$ where $E[X’ \epsilon]=0$ (no endogeneity) , but what we actually observe is $X = X’ + u$ which includes additional random error $u$ (thus “error in variable”). In this case, the new regression model is
>
> $Y = (X’)^\top \theta + \epsilon = (X + u)^\top \theta + \epsilon = X^\top \theta + u^\top \theta + \epsilon.$
>
> Here, when $X$ is treated as the independent variable, the noise is $u^\top \theta + \epsilon$. There will generally be a non-zero correlation between this noise and $X = X’ + u$ due to the shared $u$ components. For example, even when $X$ is 1-d and $u$ is independent of all other random variables, we still have $E[X (u \theta + \epsilon)] = E[u^2 \theta] \ne 0$, so this is a problem where OLS fails and IV is needed.
>
> In addition, after our submission, we notice that two earlier works (Bradtke & Barto, 1996; Chen et al. 2022) have also mentioned the IV interpretation of LSTD, though they did not engage in any kind of sample-complexity analyses or discussion of coverage. We will cite them in the revision.
>
> ---
>
> **IV perspective disappears in proof?**
>
> You are correct that our proofs use standard linear algebraic and concentration machinery, though this is the case for standard IV analyses as well (see e.g. Xia et al. 2024, which has an IV bound with dependence on the minimum singular value of A, and whose proof is similar to ours as well as standard linear regression). Indeed, we intended to directly apply concentration bounds from the IV literature, but they mostly focus on parameter recovery instead of the accuracy of prediction along a specific direction ($\phi\_0$ for us), so we had to essentially reproduce some of their analyses and tweak them to fit our needs.
>
> Furthermore, without casting the problem as IV and getting inspirations from concentration inequalities for IV estimation, **we wouldn’t even know what kind of bounds we should be proving**, as existing bounds in the literature all depend on parameters that are not interpretable and lack connections to well-established coverage definitions. What we did was to look up the term that appears in IV estimation’s guarantee, which is
>
> $\phi\_0^\top E[ZX^\top]^{-1} E[ZZ^\top] E[ZX^\top]^{-\top} \phi\_0$.
>
> Even this did not seem promising until we plugged in the expressions of $X$ and $Z$ for LSTDQ (which yields $C\_\phi^\pi$) and realized that it was the familiar density ratio in the tabular case. So in short, the IV perspective was absolutely pivotal in how we arrived at the theorem statement that needs to be proved in the first place, though the final proof itself relies less on the formulation.

---

> > ### Author Response · Authors · 2025-11-17
> >
> > (cont.)
> >
> > **Eq.(7) ... linear regression ... should come with observable individual data points**
> >
> > Strictly speaking, yes, but that is just a matter of changing $\mathbb{E}$ to $\hat{\mathbb{E}}$. We find it cleaner in notation to explain the concepts by assuming exact expectations. We can add clarification that the actual regression algorithm is run on the finite sample, though it should be pretty clear from the context.
> >
> > ---
> >
> > **Numerical results? Tighter than existing "estimators"?**
> >
> > Our paper does not propose a new algorithm or estimator, but rather provide novel theoretical analyses for an existing decades-old algorithm, LSTDQ. The coverage parameter is a term that appears in the finite-sample error bound and is not something that the algorithm needs to compute, so we will assume that the reviewer meant “tighter than existing analyses” rather than "tighter than existing estimators” ($C\_\phi^\pi$ is not an “estimator”). Regarding this point: we did explicitly compare to other analyses of LSTD as well as other estimators. As mentioned on Line 340, Perdomo et al’s work was shown to be sharp and had subsumed many prior analyses. When we compare our bound to theirs (see our Proposition 2), we are sharper by an operator norm bound, and thus our bound might represent an arbitrary improvement (in an instance-dependent fashion). For example, consider the case where the minimum singular value of the matrix $I - \gamma \Sigma^{-1/2}\Sigma\_{\textrm{cr}} \Sigma^{-1/2}$ goes to 0 (so the right-hand-side of Proposition 2 goes to infinity), but $\phi\_0$ remains in the directions spanned by the non-minimal eigenvectors, so our coverage term (the LHS of Proposition 2) stays bounded.
> >
> > ---
> >
> > **References**
> >
> > Bradtke and Barto. Machine learning 1996. Linear least-squares algorithms for temporal difference learning.
> >
> > Chen et al. JMLR 2022. On instrumental variable regression for deep offline policy evaluation.
> >
> > Xia et al. arXiv 2024. Instrumental variables:A non-asymptotic viewpoint.

---

> > > ### Comment · Reviewer_4tkR · 2025-11-17
> > >
> > > I thank the authors for their timely responses.
> > >
> > > For the question regarding the "tightness" of $C_{\phi}$, I admit that the last word should be "parameter" instead of "estimator" (sorry for the typo). What I'd like to see here is that maybe you could provide some empirical evidence that $C_{\phi}$ is a *comparable* (or even *better*) coverage parameter than other existing parameters. I think this would be helpful since $C_{\phi}$ is a new coverage parameter and such connections need to be established.

---

> > > > ### Author Response · Authors · 2025-11-17
> > > >
> > > > Dear reviewer 4tkR,
> > > >
> > > > Thanks for the clarification. Our response in the above rebuttal actually exactly addressed your question, that we can establish the improvement theoretically and show situations where the gap is large, so an empirical comparison is not necessary or meaningful. In particular, we compared to Perdomo et al.’s work who gave the tightest bound compared to what was known in the literature. Our Proposition 2 shows our coverage parameter is always tighter than theirs, and we provide a concrete scenario (see the end of the paragraph in our rebuttal) where their parameter is unbounded but ours is bounded. If you think is helpful, we can certainly include this example (with more concrete construction) in the revision.
> > > >
> > > > thanks,
> > > >
> > > > Authors

---

> ### Comment · Reviewer_4tkR · 2025-11-18
>
> Thank you for the follow-up! Up to this point I think all your arguments make sense to me, and I've decided to move up to 8.

---

### Official Review · Reviewer_i914 · 2025-11-02

**Soundness:** 3
**Presentation:** 2
**Contribution:** 3
**Rating:** 6
**Confidence:** 3

**Summary:**

This paper revisits linear off-policy evaluation under mere realizability by
casting LSTDQ in an instrumental-variable (IV) form and introducing the
feature-dynamics coverage $C^\pi_\phi$ that tightens return-estimation guarantees. In doing so, the analysis addresses three shortcomings of
prior coverage parameters: (i) Scale Invariance: stability to feature
rescaling, unlike $1/\sigma_{\min}(A)$); (ii) Off-Policy Characterization:
the dependence on off-policy error propagation rather than only on strict
on-policy boundedness; and (iii) Unification:
$C^\pi_\phi$ aligns with prior coverage concepts, including Bellman completeness, importance weighting and abstract MDP analysis.

**Strengths:**

- Use $Z=\phi(s,a)$ as an instrumental variables to solve the "error in variables", which is induced by
    $X=\phi(s,a)-\gamma\,\phi(s',a')$, yielding a
    finite-sample value bound.

- The proposed feature-dynamics coverage resolves key deficiencies of prior metrics, by ensuring scale-invariance and meaningful characterization under general off-policy distributions.

- The new definition of coverage via Proposition 1 is elegant, interpretable, and enables unification of various existing notions.

**Weaknesses:**

1. The motivation for key constructions appears late, making the early sections harder to follow.
2. The paper could better distinguish the roles of Theorem 1 and Proposition 1 to clarify the main message.

**Questions:**

1. The construction of variables Z and X in the IV-style formulation is introduced without context or intuition upfront, making the initial analysis
hard to follow. Only later is this formulation justified through IV theory.
2. Theorem 3, which avoids the drawbacks of 1/σmin(A), is mentioned in
the appendix despite directly addressing the key concerns raised in the
introduction.
3. The three issues raised at the beginning of the paper are all addressed
through the coverage form in Proposition 1. In contrast, Theorem 1 seems
to be more of an intermediate result, and the main storyline becomes
unclear.

---

> ### Author Response · Authors · 2025-11-17
>
> We thank the reviewer for the thoughtful comments and appreciation of our work as “elegant, interpretable, and enabl[ing] unification”. We respond to your questions below.
>
> ---
>
> **IV formulation is introduced without context**
>
> We would like to first clarify that we are not using the IV formulation to design a novel algorithm. Instead, we use the formulation to understand an existing algorithm, namely LSTDQ, which enables new analysis ideas. In fact, our proofs still use standard linear algebraic and concentration machinery, and retrospectively one doesn’t necessarily need the IV tools to perform the analyses. However, without casting the problem as IV and getting inspirations from existing concentration inequalities for IV estimation, it is very difficult to realize that this is a feasible path that leads to a clean and tight analysis.
>
> On the other hand, we do recognize that, while IV is a standard tool in the causal inference community (which is somewhat adjacent to RL theory, especially with overlap in OPE research), many readers in the RL community may not be familiar with the concept. Our writing has tried to address that by first introducing the bandit case which corresponds to linear regression. The IV can be viewed as a tweak to the linear regression setting, so we believe the presentation of the paper provides some scaffolding and pedagogical values. Readers who are previously unfamiliar with IV can also get some idea of what it is and how it is related to RL.
>
> ---
>
> **Theorem 3 which drops $1/\sigma\_{\min}(A)$ only appears in the appendix**
>
> While showing that $1/\sigma\_{\min}(A)$ is not the right notion of coverage is indeed one of the concerns we want to address, Theorem 1 and its corollary already address that: Theorem 1 has no dependence on  $1/\sigma\_{\min}(A)$ whatsoever (though it uses an empirical version of coverage). Note that empirical coverage is not uncommon at all in standard offline RL theory papers: for example, Jin et al. (2021), who proposed the first pessimistic algorithms and guarantees in linear MDPs, provide their guarantees that depend on a form of empirical coverage; see their Theorem 4.4 and Equation 8.8 (arXiv ver), where $\Lambda\_h$ is the empirical covariance matrix. In addition, our Corollary 1 also has no dependence on $1/\sigma\_{\min}(A)$ in the main term, and only contains it in the lower-order term. We believe these results are already quite convincing, and Theorem 3 only further adds to them by showing that a bound based on population coverage is possible without any dependence on $1/\sigma\_{\min}(A)$, though it requires modification of algorithm, more complex proof machinery, and provides looser bounds in the main term. Given the complexity, we believe it is appropriate to defer it to the appendix, leaving the main paper clean and more readable.
>
> ---
>
> **Organization of Theorem 1 and Proposition 1**
>
> Indeed, Proposition 1 plays an important role in our storyline as it provides interpretations to the coverage parameter. However, we also respectfully disagree that “Theorem 1 [is] more of an intermediate result”. As for any new definition of this kind in machine learning theory, the definition itself is not useful without a theoretical guarantee that uses the definition. Our storyline is built on two complementary results, that (1) tight theoretical guarantees for LSTDQ that depend on our new coverage parameter (Theorem 1 and its corollary), and (2) interpretations of the new coverage parameter and its connection to existing concepts (Proposition 1 and subsequent discussions). They come hand-in-hand, and Proposition 1 will be meaningless when presented alone without Theorem 1 that justifies the relevance of the new coverage parameter in the first place. If the reviewer feels that a “theorem” sounds much more important than a “proposition”, and the labels (“theorems” vs. “proposition”) may not be a great match for the actual roles these results play, we are happy to reconsider that and perhaps alleviate Proposition 1 to a theorem.
>
> ---
>
> **References**
>
> Jin et al. ICML 2021. Is pessimism provably efficient for offline RL?

---

### Meta-Review · Area_Chair_hYKS · 2026-01-05

**Summary:**

This paper works on off-policy evaluation for MDP with linear approximation (linear realizability and linear bellman completeness). After the response, the reviewers reached a consensus that this paper is theoretically solid. The introduction of the new coverage parameter $C_{\phi}$ provides a key insight, offering a unified theoretical perspective on coverage conditions in linear off-policy evaluation (OPE). As a result, my recommendation is to accept.

**Reviewer Concerns:**

Concerns addressed by the response:
1) Clarity about "IV perspective";
2) The validity of the assumption of $\Sigma$ and $A$;
3) Explanation about the motivation.

Concerns remaining after the response:
1) Lack of numerical experiments.

**Reviewer Scores:**

Reviewer 4tkR claimed he would raise his score to 8, and Reviewer Jiag had agreed to raise his score. I expect the rest two reviewers would keep their positive scores.

---

### Decision · Program_Chairs · 2026-01-26

Accept (Poster)